# Genetic dissection of the glutamatergic neuron system in cerebral cortex

Katherine S. Matho[1], Dhananjay Huilgol[1,2,11], William Galbavy[1,3,11], Miao He[1,8,11], Gukhan Kim[1,11], Xu An[1,2], Jiangteng Lu[1,9], Priscilla Wu[1], Daniela J. Di Bella[4], Ashwin S. Shetty[4], Ramesh Palaniswamy[1], Joshua Hatfield[1,2], Ricardo Raudales[1,3], Arun Narasimhan[1], Eric Gamache[1], Jesse M. Levine[1,5], Jason Tucciarone[1,5,10], Eric Szelenyi[1], Julie A. Harris[5,6], Partha P. Mitra[1], Pavel Osten[1], Paola Arlotta[4,7] & Z. Josh Huang[1,2 ✉]

Diverse types of glutamatergic pyramidal neurons mediate the myriad processing streams and output channels of the cerebral cortex[1,2], yet all derive from neural progenitors of the embryonic dorsal telencephalon[3,4]. Here we establish genetic strategies and tools for dissecting and fate-mapping subpopulations of pyramidal neurons on the basis of their developmental and molecular programs. We leverage key transcription factors and effector genes to systematically target temporal patterning programs in progenitors and differentiation programs in postmitotic neurons. We generated over a dozen temporally inducible mouse Cre and Flp knock-in driver lines to enable the combinatorial targeting of major progenitor types and projection classes. Combinatorial strategies confer viral access to subsets of pyramidal neurons defined by developmental origin, marker expression, anatomical location and projection targets. These strategies establish an experimental framework for understanding the hierarchical organization and developmental trajectory of subpopulations of pyramidal neurons that assemble cortical processing networks and output channels.

Pyramidal neurons (PyNs) constitute the large majority of nerve cells in the cerebral cortex and mediate all of the inter-areal processing streams and output channels[1,2,4]. Traditionally, PyNs have been classified into several major classes according to their laminar location and broad axon projection targets, such as intratelencephalic (IT) and extratelencephalic (ET or corticofugal), which further comprises subcerebral (including pyramidal tract; PT) and corticothalamic (CT) PyNs[1]. Within these classes, subsets of PyNs form specific local and long-range connectivity, linking discrete microcircuits to cortical subnetworks and output channels[1,5]. Single-cell transcriptome analysis suggests that there are over fifty PyN transcriptomic types[6]. However, genetic tools and strategies for experimentally accessing PyN subpopulations are limited.

All PyNs are generated from neural progenitors in the embryonic dorsal telencephalon, where regionally differentiated radial glial progenitors (RGs) undergo asymmetric divisions, giving rise to radial clones of PyNs that migrate to the cortex in an inside-out order[7]. RGs generate PyNs either directly or indirectly through intermediate progenitors (IPs), which divide symmetrically to generate pairs of PyNs[8]. A set of temporal patterning genes drive lineage progression in RGs, which unfold a conserved differentiation program in successively generated postmitotic neurons[3,4,9]. Resolving the lineage organization of diverse progenitors and their relationship to projection-defined PyN subpopulations requires fate-mapping tools with cell type and temporal resolution[2].

Here we present strategies and a genetic toolkit in the mouse for targeting PyN subpopulations and progenitors guided by knowledge of their developmental programs. We leverage gene expression patterns of the cell-type specification and differentiation programs to target biologically significant progenitor subsets, PyN subpopulations and their developmental trajectories (Fig. 1a–c, Extended Data Table 1). These tools and strategies provide a roadmap for accessing hierarchically organized PyN types at progressively finer resolution. They will facilitate the tracking of developmental trajectories of PyNs for elucidating the organization and assembly of neural circuits of the cerebral hemisphere, including the cortex, hippocampus and basolateral amygdala.

## Fate-mapping PyN progenitors
### RGs
The transcription factors LHX2 and FEZF2 act at multiple stages throughout corticogenesis[10–12]. The fate potential of and relationship between *Lhx2*+ RGs (RGs[Lhx2]) and *Fezf2*+ RGs (RGs[Fezf2]) are largely unknown. We generated *Lhx2-CreER*, *Fezf2-CreER* and *Fezf2-Flp* driver lines and performed a series of fate-mapping experiments at multiple

[1]Cold Spring Harbor Laboratory, Cold Spring Harbor, New York, NY, USA. [2]Department of Neurobiology, Duke University Medical Center, Durham, NC, USA. [3]Program in Neuroscience, Department of Neurobiology and Behavior, Stony Brook University, Stony Brook, NY, USA. [4]Department of Stem Cell and Regenerative Biology, Harvard University, Cambridge, MA, USA. [5]Program in Neuroscience and Medical Scientist Training Program, Stony Brook University, New York, NY, USA. [6]Allen Institute for Brain Science, Seattle, WA, USA. [7]Stanley Center for Psychiatric Research, Broad Institute of MIT and Harvard, Cambridge, MA, USA. [8]Present address: Institutes of Brain Science, State Key Laboratory of Medical Neurobiology and MOE Frontiers Center for Brain Science, Fudan University, Shanghai, China. [9]Present address: Shanghai Jiaotong University Medical School, Shanghai, China. [10]Present address: Department of Psychiatry, Stanford University School of Medicine, Palo Alto, CA, USA. [11]These authors contributed equally: Dhananjay Huilgol, William Galbavy, Miao He, Gukhan Kim. ✉e-mail: josh.huang@duke.edu

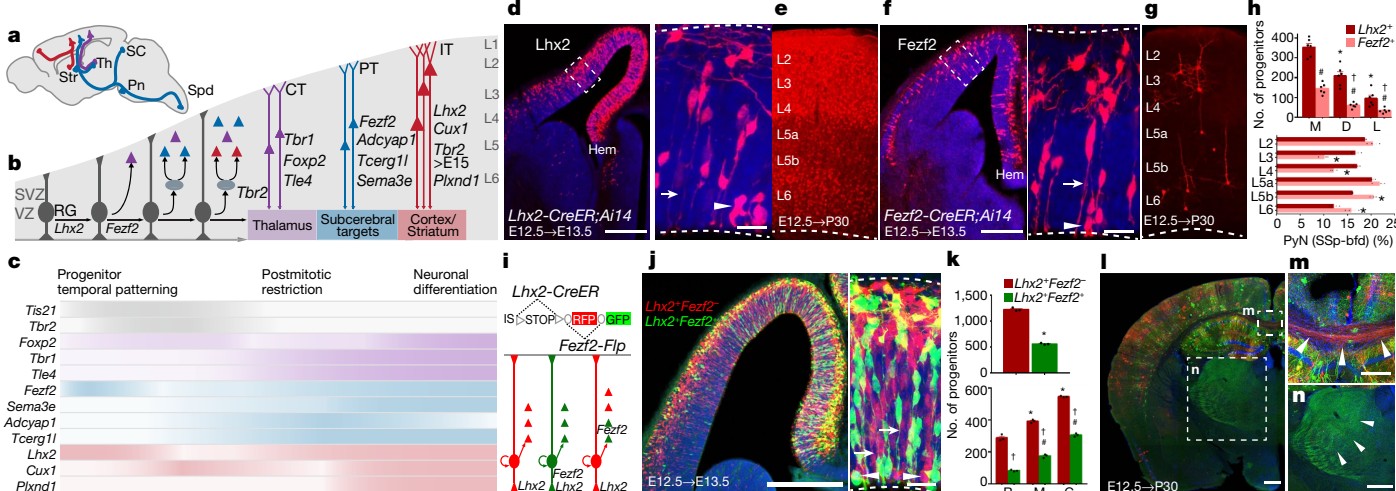

**Fig. 1 | Strategies and drivers to target PyN types and fate-map progenitors.**
**a**, Major PyN projection classes mediating intratelencephalic streams (IT, red) and cortical output channels (PT, blue; CT, purple) in a sagittal brain section. Pn, pons; SC, superior colliculus; Str, striatum; Th, thalamus; Spd, spinal cord. **b**, PyN developmental trajectory. RGs undergo direct and indirect (IP-derived) neurogenesis, producing all laminar and projection types. The listed genes are expressed in progenitor and PyN subpopulations. SVZ, subventricular zone; VZ, ventricular zone. **c**, Temporal expression patterns of genes used for generating knock-in drivers across PyN development. Colours correspond to projection class; intensity gradients depict expression levels. **d**, E12.5 tamoxifen pulse-chase in *Lhx2* embryos labelled RGs$^{Lhx2+}$ with a medial$^{high}$ to lateral$^{low}$ gradient along the dorsal neuroepithelium, ending at the cortex–hem boundary. The magnified views in **d**, **f**, **j** show RGs at multiple cell-cycle stages, with end-feet (arrows) and dividing soma (arrowheads) at the ventricle wall (dashed line). **e**, E12.5 RGs$^{Lhx2+}$ produced PyNs across cortical layers. **f**, E12.5 tamoxifen pulse-chase in *Fezf2* embryos labelled RGs$^{Fezf2+}$ with a gradient

distribution similar to that in **d** but at a lower density. **g**, RGs$^{Fezf2+}$ produced PyNs across layers. **h**, Top, distribution of RGs$^{Lhx2+}$ (red) and RGs$^{Fezf2+}$ (pink) across cortical neuroepithelium divided into medial (M), dorsal (D) and lateral (L) bins. Bottom, laminar distribution of fate-mapped PyNs. **i**, Fate-mapping scheme using an IS reporter with *Lhx2-CreER* and *Fezf2-Flp*: RGs$^{Lhx2+/Fezf2-}$ express tdTomato/RFP by 'Cre-NOT-Flp' subtraction; RGs$^{Lhx2+/Fezf2+}$ express EGFP by 'Cre-AND-Flp' intersection. **j**, E12.5 tamoxifen 24-hour pulse-chase revealed RGs$^{Lhx2+/Fezf2-}$ and RGs$^{Lhx2+/Fezf2+}$ throughout the cortical primordium. **k**, Top, the labelled number of RGs$^{Lhx2+/Fezf2+}$ is half that of RGs$^{Lhx2+/Fezf2-}$. Bottom, the number of RGs$^{Lhx2+/Fezf2-}$ versus RGs$^{Lhx2+/Fezf2+}$ at rostral (R), mid-level (M) and caudal (C) sections. Data in **h**, **k** are mean ± s.e.m.; see 'Quantification and statistics related to progenitor fate-mapping' in the Methods for statistical details. **l–n**, RG$^{Lhx2+/Fezf2-}$-derived PyNs (red) project to the corpus callosum (arrowheads, **m**) without subcortical branches; RG$^{Lhx2+/Fezf2+}$-derived PyNs project to the thalamus (arrowheads, **n**) without callosal branches. DAPI (blue). Scale bars, 20 μm (**d**, **f**, **j** insets); 100 μm (all other panels).

embryonic stages to reveal these progenitors and their lineage progression, as well as their PyN progeny in the mature cortex (Fig. 1d–h, Extended Data Figs. 1, 2).

At embryonic day (E) 10.5, a 24-hour tamoxifen pulse-chase in *Lhx2-CreER;Ai14* embryos resulted in dense labelling of neuroepithelial cells and RGs in the dorsal pallium, with a sharp border at the cortex–hem boundary (Extended Data Fig. 1a). E12.5–E13.5 pulse-chase revealed a prominent medial$^{high}$ to lateral$^{low}$ gradient of RGs$^{Lhx2}$ (Fig. 1d), suggesting differentiation of the earlier RGs. E13.5–E14.5 pulse-chase showed a similar gradient pattern at a lower cell density (Extended Data Fig. 1e). Fate-mapping from E10.5–P30, E12.5–P30 and E14.5–P30 labelled PyN progeny across cortical layers (Fig. 1e, Extended Data Fig. 1b–f, p), suggesting multipotency of RGs$^{Lhx2}$ at these stages. During postnatal development, the expression of *Lhx2* became postmitotic: pulse-chase in P5 labelled largely IT PyNs across layers and in the second postnatal week labelled more astrocytes (around 60%) than PyNs across layers (Extended Data Fig. 1q–r).

Similar fate-mapping experiments using the *Fezf2-CreER* driver yielded contrasting results. At E10.5, short pulse-chase labelled only a sparse set of pallial RGs, ending at the cortex–hem boundary (Extended Data Fig. 1g). E12.5–E13.5 pulse-chase labelled a larger set of RGs$^{Fezf2}$ with a similar medial$^{high}$ to lateral$^{low}$ gradient as RGs$^{Lhx2}$, with a notably lower density (Fig. 1f). E13.5–E14.5 pulse-chase labelled few RGs, primarily in the medial region, and otherwise postmitotic PyNs (Extended Data Fig. 1k). Fate-mapping from E10.5–P30 and E12.5–P30 labelled PyNs across cortical layers, suggesting multipotent RGs$^{Fezf2}$ (Fig. 1g, Extended Data Fig. 1h, j). After E13.5, the expression of *Fezf2* largely shifted to postmitotic layer (L) 5 and 6 (L5/6) corticofugal PyNs (Extended Data Fig. 1l). Both the *Lhx2-CreER* and the *Fezf2-CreER* drivers recapitulate endogenous expression across developmental stages

(Extended Data Fig. 2), thus providing fate-mapping tools for these progenitor pools.

To probe the relationship between RGs$^{Lhx2}$ and RGs$^{Fezf2}$, we designed an intersection–subtraction (IS) strategy. Combining *Lhx2-CreER* and *Fezf2-Flp* with an IS reporter[13], we differentially labelled RGs$^{Lhx2+/Fezf2-}$ and RGs$^{Lhx2+/Fezf2+}$ (Fig. 1i–n, Extended Data Fig. 1s, t). E11.5–E12.5 and E12.5–E13.5 pulse-chase revealed two distinct RG subpopulations intermixed across the dorsal pallium, with RGs$^{Lhx2+/Fezf2-}$ more than twice as abundant as RGs$^{Lhx2+/Fezf2+}$. Both subpopulations distributed in a medial$^{high}$ to lateral$^{low}$ and rostral$^{high}$ to caudal$^{low}$ gradient, consistent with the patterns of *Lhx2* and *Fezf2* expression (Fig. 1j, k). Although most RGs$^{Fezf2}$ expressed LHX2, approximately 10% did not (Extended Data Fig. 2e, i), suggesting that there are three distinct RG subpopulations distinguished by differential expression of *Lhx2* and *Fezf2*. Notably, long pulse-chase revealed that whereas RGs$^{Lhx2+/Fezf2-}$-derived PyNs extended callosal but no subcortical axons—the IT type—RGs$^{Lhx2+/Fezf2+}$-derived PyNs extended subcortical but no callosal axons—the ET type (Fig. 1l–n). This result suggested fate-restricted RG lineages that produce categorically distinct PyN projection classes.

## Neurogenic RGs

Early cortical progenitors comprise proliferative and neurogenic subpopulations. *Tis21* (also known as *Btg2*) is a transcription co-regulator that is expressed in both pallium-derived glutamatergic and subpallium-derived GABAergic neurogenic RGs (nRGs)[14]. E10.5 fate-mapping in the *Tis21-CreER* driver line labelled columnar clones of PyNs and astrocytes intermixed with subpallium-derived GABAergic interneurons (Extended Data Fig. 3b, f, g). We used *Tis21-CreER;Fezf2-Flp;IS* mice to restrict fate-mapping to glutamatergic neurogenic RGs (Extended Data Fig. 3h–k). E11.5–E12.5 pulse-chase

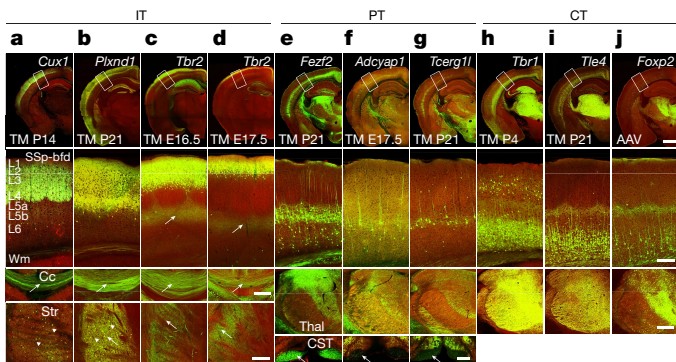

**Fig. 2 | Genetic targeting of PyN subpopulations. a–j,** Driver line recombination patterns visualized through reporter expression (green; background autofluorescence, red). First row, coronal hemisections. TM, tamoxifen. **a–d,** Second row, IT drivers targeting laminar subsets in L2–L5a of somatosensory barrel cortex (SSp-bfd), which project axons across the corpus callosum (Cc) (third row) and to the striatum (Str) (bottom row). *Cux1* and *Plxnd1* drivers also label subsets of medium spiny neurons in the striatum (arrowheads). E16.5 and E17.5 tamoxifen induction of *Tbr2-CreER* label L2/3 and L2 PyNs, respectively. **e–g,** Second row, PT drivers label L5B PyNs, which project to numerous subcortical targets, including the thalamus (Thal) (third row) and spinal cord (corticospinal tract; CST) (bottom row). **h–j,** Second row, CT drivers label L6 PyNs, sending axons mainly to different nuclei in the thalamus (third row). Tamoxifen induction times are indicated in the first row. The reporter allele was *Ai14*, except for *Plxnd1-CreER* (*Snap25-LSL-EGFP*) and *Foxp2* (systemic injection of AAV9-CAG-DIO-EGFP). White matter (Wm). Scale bars, 100 μm (bottom (CST) panel in **g**); 1 mm (hemisection in **j**, which applies to the entire row); 200 μm (all other scale bars). Cell bodies are indicated by arrowheads and axons by arrows.

demonstrated that *Tis21–Fezf2* intersection specifically labelled a set of pallial nRG$^{Fezf2+}$ with enhanced green fluorescent protein (EGFP), whereas *Tis21–Fezf2* subtraction labelled pallial and subpallial nRG$^{Fezf2-}$ with red fluorescent protein (RFP). Pallial nRGs consisted of both *Fezf2*$^+$ and *Fezf2*$^-$ subpopulations, suggesting heterogeneity. E12.5–P30 fate-mapping in these mice revealed three types of PyN clones (Extended Data Fig. 3i-k). RFP-only clones are likely to have derived from nRG$^{Fezf2-}$ in which *Tis21-CreER* activated RFP expression; they probably consisted of PyNs that did not express *Fezf2* at any stage. EGFP-only clones are likely to have derived from nRG$^{Fezf2+}$, in which *Tis21-CreER* and *Fezf2-Flp* co-expression activated EGFP in the IS reporter allele. Mixed clones containing both EGFP and RFP cells probably derived from nRG$^{Fezf2-}$ in which *Tis21-CreER* activated RFP expression followed by postmitotic activation of EGFP through *Fezf2-Flp*. Together, these results indicate the presence of nRG$^{Fezf2+}$ and nRG$^{Fezf2-}$, both multipotent in generating PyNs across all cortical layers.

## IPs

IPs and indirect neurogenesis have evolved largely in the mammalian lineage and have further expanded in primates[14,15]. Along the neural tube, IP-mediated indirect neurogenesis is restricted to the telencephalon and is thought to contribute to the expansion of cell numbers and diversity in the neocortex. The majority of PyNs in mouse cortex are produced through IPs[16,17], but the link between indirect neurogenesis and PyN types remains unclear. The T-box transcription factor *Tbr2* (also known as *Eomes*) is expressed in pallial IPs throughout indirect neurogenesis[18]. E16.5 pulse-chase in the *Tbr2-CreER* driver line specifically labelled IPs (Extended Data Fig. 3a, c). E16.5 and E17.5 fate-mapping labelled PyNs in L2/3 and upper L2, respectively (Fig. 2c, d, Extended Data Fig. 3d). Therefore, the *Tbr2-CreER* driver enables highly restricted laminar targeting of PyN subpopulations in supragranular layers. Furthermore, *Tis21-CreER* and *Tbr2-FlpER* intersection enabled specific targeting of neurogenic but not the transit-amplifying IPs (Extended

Data Fig. 3a, e). Altogether, these progenitor driver lines facilitate dissecting progenitor diversity and tracking the developmental trajectories of PyNs from their lineage origin to circuit organization.

## Targeting PyN subpopulations

We generated driver lines targeting PyN subpopulations and characterized these in comparison to existing lines where feasible (Fig. 2, Extended Data Table 1, Extended Data Fig. 4, Supplementary Tables 1, 2). These tamoxifen-inducible drivers confer temporal control and dose-dependent labelling and manipulation from individual cells to dense populations.

### IT drivers

IT PyNs constitute the largest top-level class and mediate intracortical and corticostriatal communication streams[1,19–21]. *Cux1* and *Cux2* are predominantly expressed in supragranular IT PyNs and their progenitors[22,23]. In our *Cux1-CreER;Ai14* mice, postnatal tamoxifen induction prominently labelled L2–L4 PyNs dorsal to the rhinal fissure as well as a set of hippocampal PyNs, recapitulating the endogenous pattern (Fig. 2a, Extended Data Fig. 4, Supplementary Video 1). Anterograde tracing revealed that PyNs$^{Cux1}$ in somatosensory barrel cortex (SSp-bfd or SSp) projected predominantly to the ipsi- and contralateral cortex, with only very minor branches in the striatum (Fig. 3a, d, Extended Data Fig. 7, Supplementary Video 2). Compared with existing IT drivers (Extended Data Figs. 4, 5, Supplementary Tables 1, 2), *Cux1-CreER* is unique in targeting predominantly cortex- but not striatum-projecting IT subpopulations.

The supragranular layers comprise diverse IT types[20], but only a few L2/3 drivers have been reported so far[24] and none distinguish L2 versus L3 PyNs. We used a lineage and birth dating approach to dissect L2/3 PyNs. In our *Tbr2-CreER* driver targeting IPs, tamoxifen induction at E16.5 and E17.5 specifically labelled PyNs in L2/3 and L2, respectively (Fig. 2c, d). Combined with the CreER to Flp conversion strategy that converts lineage and birth timing signals to permanent Flp expression[13], this approach enables adeno-associated virus (AAV) manipulation of L2 and L3 IT neurons.

The plexin D1–semaphorin 3E receptor–ligand system has been implicated in axon guidance and synapse specification[25,26]. In developing and mature cortex, *Plxnd1* (encoding plexin D1) is expressed in large sets of IT PyNs[10]. *Plxnd1-CreER* and *Plxnd1-Flp* driver lines recapitulated endogenous expression and labelled projection neurons in the cerebral cortex, hippocampus, amygdala and striatum (Figs. 2b, 4e, Extended Data Figs. 4, 5a, Supplementary Tables 3, 4, Supplementary Videos 3, 4); in the neocortex, L5A and L2/3 IT PyNs$^{Plxnd1}$ were labelled (Fig. 2b, Extended Data Fig. 6a–c). As *Plxnd1* is also expressed in vascular cells, we bred *Plxnd1-CreER* mice with the neuron-specific reporter *Snap25-LSL-EGFP* to selectively label *Plxnd1*$^+$PyNs (PyNs$^{Plxnd1}$; Fig. 2b). Anterograde tracing from SSp-bfd revealed that PyNs$^{Plxnd1}$ project to ipsi- and contralateral cortical and striatal regions (Fig. 3a, d, Extended Data Fig. 7a–c, f, Supplementary Tables 5–7, Supplementary Video 5). Thus, *Plxnd1* drivers confer access to this major IT subpopulation and to *Plxnd1*$^+$ subpopulations in the striatum and amygdala.

### PT drivers

After early expression in a subset of dorsal pallial progenitors, *Fezf2* becomes restricted to postmitotic L5/6 corticofugal PyNs, with higher levels in L5B PT neurons and lower levels in a subset of CT neurons[10,27]. At postnatal stages, *Fezf2* drivers labelled projection neurons in the cerebral cortex, hippocampus, amygdala, and olfactory bulb (Extended Data Fig. 4, Supplementary Table 4, Supplementary Video 6). Within the neocortex, PyNs$^{Fezf2}$ reside predominantly in L5B and to a lesser extent in L6 (Fig. 2e, 1, Extended Data Figs. 4a, d, 6d–f); PyNs$^{Fezf2}$ are absent below the rhinal fissure (Supplementary Table 4, Supplementary Video 6). Anterograde tracing of PyNs$^{Fezf2}$ in SSp-bfd revealed projections to numerous somatomotor cortical (for example, ipsilateral

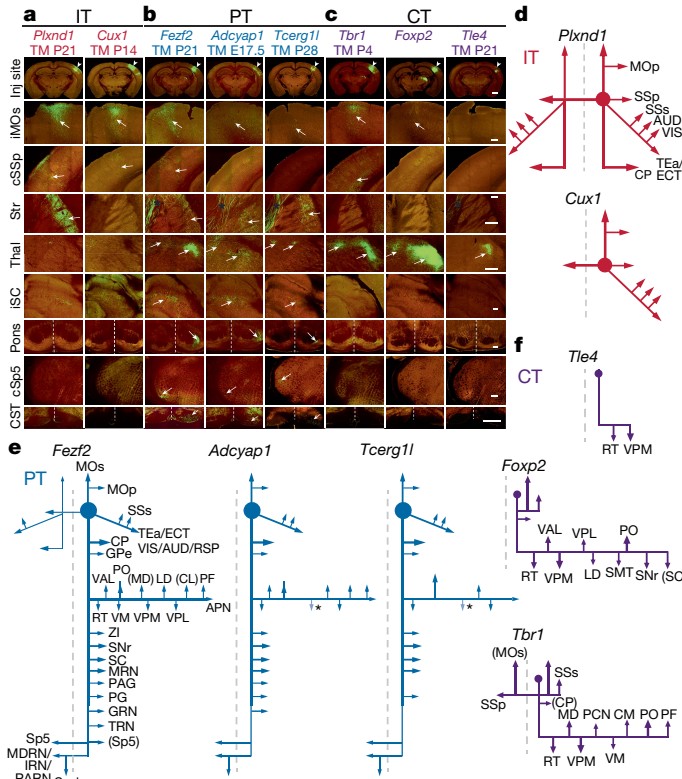

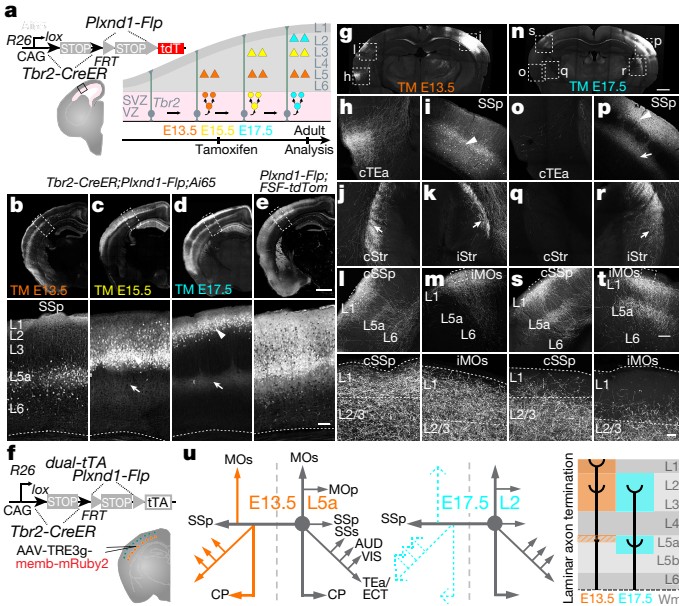

## Fig. 3 | Projection patterns of PyN subpopulations in SSp-bfd cortex.

**a**–**c**, Images at SSp-bfd injection site (inj site) (first row, arrowhead) and selected subcortical projection targets for eight driver lines: EGFP expression from Cre-activated viral vector (green) and background autofluorescence (red). Tamoxifen induction time points are indicated below each gene name. Arrows indicate axons. **a**, IT drivers project to the cortex and striatum. Note the near absence of projection to the striatum for the *Cux1* driver. **b**, PT drivers project to many corticofugal targets, including the brainstem and spinal cord. **c**, CT drivers project predominantly to the thalamus. **d**–**f**, Schematics of main projection targets for each PyN subset generated in this study. **d**, IT drivers. **e**, PT drivers. **f**, CT drivers. Ipsilateral secondary motor area (iMOs); contralateral primary somatosensory area (cSSp); secondary somatosensory area (SSs); auditory areas (AUD); visual areas (VIS); retrosplenial area (RSP); temporal association areas (TEa); ectorhinal area (ECT); reticular nucleus of the thalamus (RT); ventral anterior–lateral complex of the thalamus (VAL); ventral posteromedial complex of the thalamus (VPM); ventral posterior–lateral complex of the thalamus (VPL); submedial nucleus of the thalamus (SMT); posterior complex of the thalamus (PO); substantia nigra, reticular part (SNr); superior colliculus (SC); caudoputamen (CP); mediodorsal nucleus of the thalamus (MD); paracentral nucleus (PCN); central medial nucleus of the thalamus (CM); parafascicular nucleus (PF); globus pallidus, external segment (GPe); lateral dorsal nucleus of the thalamus (LD); central lateral nucleus of the thalamus (CL); anterior pretectal nucleus (APN); ventral medial nucleus of the thalamus (VM); zona incerta (ZI); midbrain reticular nucleus (MRN); periaqueductal gray (PAG); pontine gray (PG); gigantocellular reticular nucleus (GRN); tegmental reticular nucleus (TRN); medullary reticular nucleus (MDRN); intermediate reticular nucleus (IRN); and parvicellular reticular nucleus (PARN). Scale bars, 1 mm (first row, in **c**); 200 μm (second to eighth rows, in **c** (for each respective row); 100 μm (CST panel in **c**, which applies to the bottom row). Asterisks in **b**, **c**, **e** indicate passing fibres.

vibrissal secondary motor area (MOs)) and subcortical regions including the striatum, thalamic ventral posteromedial nucleus of the thalamus (VPM) and posterior complex of the thalamus (PO), anterior pretectal nucleus, ipsilateral superior colliculus (iSC), pontine nucleus, corticospinal tract (CST) and contralateral spinal trigeminal nucleus (cSp5) (Fig. 3b, e, Extended Data Figs. 7a, b, d, f, 8f–r, Supplementary Tables 5, 6, Supplementary Video 7).

## Fig. 4 | Combinatorial targeting of IT subtypes by lineage, birth time and anatomy.

**a**, Strategy for combinatorial labelling of PyN[Plxnd1] subtypes. In a *Tbr2-CreER;Plxnd1-Flp;Ai65* mouse, tamoxifen inductions at successive embryonic times label deep or more superficial PyN[Plxnd1] subsets born sequentially from intermediate progenitors. **b**–**d**, Laminar subsets born at E13.5 (**b**), E15.5 (**c**) and E17.5 (**d**). **e**, The overall population is labelled in a *Plxnd1-Flp;R26-FSF-tdTom* mouse for comparison. The bottom panels in **b**–**e** show high-magnification images of the boxed regions in the top panels. **f**, In *Tbr2-CreER;Plxnd1-Flp;dual-tTA* mice, E13.5 or E17.5 tamoxifen inductions activate tTA expression in L5A or L2 PyNs[Plxnd1], respectively, and AAV-TRE3g-memb-mRuby2 anterograde injection in SSp-bfd reveals the projection pattern of each laminar subset. **g**, **n**, Coronal sections display the injection site and several major projection targets. **g**–**m**, Anterograde tracing from E13.5-born L5A PyNs[Plxnd1] in SSp-bfd, with images (**h**–**m**) of projection targets in several ipsi- and contralateral regions. **n**–**t**, Anterograde tracing from E17.5-born L2 PyNs[Plxnd1] in SSp-bfd, with images (**o**–**s**) of projection targets in several ipsi- and contralateral regions. The bottom panels in **l**, **m**, **s** and **t** show high-magnification images of the boxed regions in the top panels. The higher magnification of cSSp (**l**, **s**; bottom) and iMOs (**m**, **t**; bottom) display laminar axon termination differences between L5A and L2 PyNs[Plxnd1]. **u**, Schematics comparing E13.5-born L5A (left) and E17.5-born L2 (middle) PyN[Plxnd1] projection patterns; note differences in the strength of several contralateral targets and in the laminar pattern of axon termination (right). Arrowheads indicate cell body positions; arrows indicate axons. Contralateral temporal association area (cTEa); ipsilateral striatum (iStr); contralateral striatum (cStr); white matter (Wm). Scale bars: 1 mm (**b**–**e** (top panels), **g**, **n**); 200 μm (**b**–**e** (bottom panels)); 50 μm (**h**–**t** (including top panels in **l**, **m**, **s**, **t**)); 5 μm (**l**, **m**, **s**, **t** (bottom panels)).

We further generated several lines targeting finer PT subpopulations. In *Adcyap1-* and *Tcerg1l-CreER* drivers, late embryonic induction labelled L5B subpopulations that project only to ipsilateral cortical targets and to a subset of targets innervated by PyN[Fezf2] (Figs. 2f, g, 3b, e, Extended Data Figs. 4, 6d–f, 7a, b, d, f, Supplementary Tables 4–6, Supplementary Videos 8–11). In *Sema3e-CreER* (ref. [28]), postnatal induction labelled a subset of L5B PyNs that project to more-restricted subcortical areas, namely higher-order thalamic nucleus POm and pontine nucleus (Extended Data Figs. 4d, 8a–e, Supplementary Tables 4, 5, Supplementary Video 12). Together, the new set of PT drivers will enable a finer hierarchical dissection of molecularly and anatomically defined PT types.

## CT drivers

*Tbr1* is expressed in postmitotic L6 CT neurons and represses the expression of *Fezf2* and *Ctip2* (also known as *Bcl11b*) to suppress the PT fate[3,29]. In *Tbr1-CreER;Ai14* mice, tamoxifen induction at P4 marked

L6 CT neurons densely, with sparse labelling in L2/3, cerebral nuclei, hippocampus, piriform cortex and amygdala (Fig. 2h, Extended Data Fig. 4a–c, Supplementary Table 4, Supplementary Video 13). PyNs$^{Tbr1}$ from SSp-bfd projected to multiple thalamic targets, including primary and higher order nuclei, as well as reticular nucleus of the thalamus (Fig. 3c, f, Supplementary Tables 5, 6, Supplementary Video 14). Consistent with a study showing that some PyNs$^{Tbr1}$ project to the contralateral cortex[30], we found labelling in the corpus callosum (Extended Data Fig. 6). It remains to be determined whether PyNs$^{Tbr1}$ with contralateral projections (Fig. 3f) represent a distinct type.

*Tle4* is a transcription corepressor that is expressed in a subset of CT PyNs[31,32]. Our *Tle4-CreER* driver specifically labelled L6 CT PyNs across the cortex (Fig. 2i, Extended Data Figs. 4, 6d–f, Supplementary Table 4, Supplementary Video 15). *Tle4* is also expressed in medium spiny neurons of the striatum, olfactory bulb, hypothalamus, iSC, cerebellum and septum (Extended Data Fig. 4c, Supplementary Table 4, Supplementary Video 15). PyNs$^{Tle4}$ in SSp-bfd specifically projected to first-order thalamic VPM and reticular nucleus of the thalamus (Fig. 3c, f, Extended Data Fig. 7a, b, e, f, Supplementary Table 4, Supplementary Video 16).

*Foxp2* is expressed in many CT neurons from the postmitotic stage to the mature cortex[33–35]. In adult *Foxp2-IRES-Cre* mice[36], systemic injection of Cre-dependent AAV9-DIO-GFP specifically labelled L6 PyNs; *Foxp2*$^+$ cells were also found in the striatum, thalamus, hypothalamus, midbrain, cerebellum and inferior olive (Fig. 2j, Extended Data Fig. 4a–c, Supplementary Table 4, Supplementary Video 17). PyN$^{Foxp2}$ in SSp-bfd projected to thalamus, tectum and some ipsilateral cortical areas (Fig. 3c, f, Supplementary Tables 4, 5, Supplementary Video 18). Compared to PyNs$^{Tle4}$, PyNs$^{Foxp2}$ projected more broadly to the thalamus, largely overlapping with PyN$^{Tbr1}$ axons.

To further characterize several PyN driver lines, we performed a set of histochemical analyses (Extended Data Fig. 6). PyNs targeted in *Fezf2*, *Tcerg1l* and *Adcyap1* drivers extensively co-labelled with PT markers. PyNs targeted in *Tle4* and *Tbr1* drivers co-labelled with CT markers. The laminar patterns and class-specific marker expression in these driver lines precisely recapitulated endogenous patterns (in situ hybridization data in the Allen Brain Map: Mouse Brain Atlas; https://mouse.brain-map.org/search/index), providing further evidence of the reliability and specificity of these driver lines.

## Combinatorial targeting of projection types

To further dissect driver-line-defined subpopulations according to projection targets, we first used retrograde tracing. Within the PT population, retroAAV and fluorogold injections in the spinal cord of *Fezf2-CreER* mice specifically labelled L5B corticospinal PyNs in the sensorimotor cortex (Extend Data Fig. 10a–c, Supplementary Table 6). To explore PyNs$^{Fezf2}$ subpopulations jointly defined by projection targets and sublaminar position, we used the IS reporter[13]. Consistent with previous findings[37], PyNs$^{Fezf2}$ that project to the thalamus and medulla resided in the upper and lower sublamina of L5B in the primary motor area (MOp), respectively (Extended Data Fig. 9d–f). In SSp-bfd, PyNs$^{Fezf2}$ with collaterals to the striatum resided in upper L5, those with collaterals to the superior colliculus or cSp5 resided in the middle and lower portion of L5B, and those projecting to thalamic POm resided both in middle to lower L5B and in L6 (Extended Data Fig. 9g–h, l–o). We then distinguished subsets of L5B PyNs$^{Fezf2}$ according to their expression of the calcium-binding protein parvalbumin using *Fezf2-CreER;Pv-Flp;IS* mice that differentially labelled PyNs$^{Fezf2+/PV-}$ and PyNs$^{Fezf2+/PV+}$ (in which *PV* represents parvalbumin; this gene is also known as *Pvalb*) (Extended Data Fig. 9i, j). Compared to PyNs$^{Fezf2+/PV-}$, PyNs$^{Fezf2+/PV+}$ exhibited more depolarized resting membrane potentials. In addition, we designed a strategy (triple trigger) to target PyNs$^{Fezf2}$ jointly defined by a driver line, a projection target and a cortical location (Extended Data Fig. 10).

We also used retroAAV to dissect the CT and IT populations. In *Tle4-CreER;IS* mice, retrograde tracing from the thalamic VPM revealed

two subpopulations of L6 PyNs$^{Tle4}$, one extending apical dendrites to the L4/5 border, the other to L1 (Extended Data Fig. 9q), suggesting differential inputs. In *Plxnd1-CreER* mice (Extended Data Fig. 9p, r–w), whereas L5A PyNs$^{Plxnd1}$ projected to both the ipsi- and the contralateral striatum, L2/3 PyNs$^{Plxnd1}$ projected mostly to the ipsilateral striatum.

In addition, consistent with the finding that some PyNs$^{Fezf2}$ extend contralateral cortical and striatal projections (Fig. 3e), retrograde cholera toxin subunit B (CTB) tracing from the striatum labelled a set of contralateral PyNs$^{Fezf2+}$ at the L5A–L5B border (Extended Data Fig. 11a–e), a characteristic IT feature. Indeed, a small set of PyNs at the L5A–L5B border co-expressed *Fezf2* and *Plxnd1* mRNAs; these PyNs$^{Fezf2/Plxnd1}$ occupied the very top sublayer of the PyN$^{Fezf2}$ population (Extended Data Fig. 11f–h), and thus probably contributed to their contralateral cortical and striatal projections (Fig. 3a, b, d, e). Single-cell reconstruction may reveal whether PyNs$^{Fezf2/Plxnd1}$ are typical IT cells or also project subcortically and represent an 'intermediate PT-IT' type.

Finally, we show highly specific targeting of PyN subtypes by combining their developmental, molecular and anatomical attributes. PyNs$^{Plxnd1}$ localize to L5A, L3 and L2 and project to numerous ipsilateral and contralateral cortical and striatal targets (Figs. 2b, 3a, d, Extended Data Figs. 4, 7). We developed a method (Fig. 4a) to dissect PyN$^{Plxnd1}$ subtypes on the basis of the developmental principle that PyN birth order correlates with laminar position and the observation that the majority of IT PyNs are generated from IPs[17]. In *Plxnd1-Flp;Tbr2-CreER;Ai65* mice, the constitutive *Plxnd1-Flp* allele marks the whole population (Fig. 4e) and the inducible *Tbr2-CreER* allele enables birth dating (Fig. 4a). Notably, tamoxifen induction at E13.5, 15.5 and 17.5 selectively labelled L5A and progressively more superficial PyNs$^{Plxnd1}$ (Fig. 4b–d). We then bred *Plxnd1-Flp;Tbr2-CreER;dual-tTA* mice for anterograde tracing of projection patterns (Fig. 4f). AAV-TRE3g-mRuby injection into SSp-bfd in E13.5- and 17.5-induced mice labelled distinct subtypes of PyNs$^{Plxnd1}$ with different projection patterns. E13.5-born PyNs$^{Plxnd1(E13.5)}$ resided in L5A and projected ipsilaterally to multiple cortical areas, contralaterally to homotypic SSp-bfd cortex and heterotypic cortical areas, and bilaterally to the striatum (Fig. 4g–m, u). By contrast, E17.5-born PyNs$^{Plxnd1(E17.5)}$ resided in L2; although they also extended strong projections to ipsilateral cortical and striatal targets and to the homotypic contralateral cortex, they had minimal projections to the heterotypic contralateral cortex and striatum (Fig. 4n–u). These birth-dated PyN$^{Plxnd1}$ subsets further differed in their axon termination patterns within a cortical target area. Whereas PyN$^{Plxnd1(E13.5)}$ axons terminated throughout the thickness of L1 and L2/3, with few axon branches in L5A, PyN$^{Plxnd1(E17.5)}$ axons terminated strongly in L2/3 and L5A, with few branches in L1 (Fig. 4l–m, s–u). Thus, even within the same target areas, birth-dated PyNs$^{Plxnd1}$ may preferentially select different postsynaptic cell types and/or subcellular compartments.

## Discussion

Together with previous resources[24,38,39], the PyN driver lines we present here provide much improved specificity, coverage and robustness for a systematic dissection of PyN organization from broad subclasses to finer types. By focusing on driver lines that recapitulate the expression of key transcription factors and effector genes that are implicated in specification and differentiation, these tools will enable the dissection and fate-mapping of biologically significant subpopulations of PyNs through their inherent developmental, anatomical and physiological properties; that is, 'carving nature at its joints'. The precision and reliability of these drivers also allows the combinatorial targeting of finer projection types through the intersection of molecular, developmental and anatomical properties. The inducibility of driver lines enhances the specificity and flexibility of cell targeting, manipulation and fate-mapping. Inducibility also allows control over the density of labelling and manipulation, from dense coverage to single-cell analysis—the ultimate resolution for clarifying the stereotypical and

variable features of neurons within marker-defined subpopulations[40–42]. Temporal control allows gene manipulations at different developmental stages to discover the cellular and molecular mechanisms of circuit development and function. Together, these tools and strategies establish a roadmap for dissecting the hierarchical organization of PyN types on the basis of their inherent biology. The incorporation of recently developed enhancer AAVs[43] with these driver lines may further increase the specificity, ease and throughput of cell-type access.

Several transcription factors used in this study (for example, *Cux1*, *Fezf2*, *Tbr1*, *Tbr2* and *Foxp2*) continue to evolve and diverge in primates[44] and are implicated in developmental disorders such as autism[23,35]. Our transcription factor driver lines provide handles to track the developmental trajectories of PyN subpopulations in cortical circuit assembly, with implications in the cross-species evolution of PyNs and for deciphering the genetic architecture of neurodevelopmental disorders.

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

## Methods

### Data reporting

No statistical methods were used to predetermine sample size. The experiments were not randomized and the investigators were not blinded to allocation during experiments and outcome assessment.

### Generation of knock-in mouse lines

Driver and reporter mouse lines listed in Supplementary Table 1 were generated using a PCR-based cloning, as described before and below[13,45]. All experimental procedures were approved by the Institutional Animal Care and Use Committee (IACUC) of the Cold Spring Harbor Laboratory (CSHL) in accordance with NIH guidelines. Mouse knock-in driver lines are deposited at The Jackson Laboratory for wide distribution. Knock-in mouse lines were generated by inserting a 2A-CreER or 2A-Flp cassette in-frame before the STOP codon of the targeted gene. Targeting vectors were generated using a PCR-based cloning approach as described before. In brief, for each gene of interest, two partially overlapping BAC clones from the RPCI-23&24 library (made from C57BL/b mice) were chosen from the Mouse Genome Browser. The 5′ and 3′ homology arms were PCR-amplified (2–5 kb upstream and downstream, respectively) using the BAC DNA as template and cloned into a building vector to flank the 2A-CreERT2 or 2A-Flp expressing cassette as described[47]. These targeting vectors were purified and tested for integrity by enzyme restriction and PCR sequencing. Linearized targeting vectors were electroporated into a 129SVj/B6 hybrid embryonic stem (ES) cell line (v6.5). ES clones were first screened by PCR and then confirmed by Southern blotting using appropriate probes. DIG-labelled Southern probes were generated by PCR, subcloned and tested on wild-type genomic DNA to verify that they give clear and expected results. Positive v6.5 ES cell clones were used for tetraploid complementation to obtain male heterozygous mice following standard procedures. The $F_0$ males were bred with reporter lines (Supplementary Tables 1, 3, 4) and induced with tamoxifen at the appropriate ages to characterize the resulting genetically targeted recombination patterns.

### Tamoxifen induction

Tamoxifen (T5648, Sigma) was prepared by dissolving in corn oil (20 mg ml$^{-1}$) and applying a sonication pulse for 60 s, followed by constant rotation overnight at 37 °C. Embryonic inductions for most knock-in lines were done in the Swiss-Webster background; inductions for *Tis21-CreER*, *Fezf2-Flp* intersection experiments were done in the C57BL6 background. E0.5 was established as noon on the day of vaginal plug and tamoxifen was administered to pregnant mothers by gavage at a dose varying from 2–100 mg kg$^{-1}$ at the appropriate age. For embryonic collection (12–24 h pulse-chase experiments), a dose of 2 mg kg$^{-1}$ was administered to pregnant dams via oral gavage. For postnatal induction, a 100–200 mg kg$^{-1}$ dose was administered by intraperitoneal injection at the appropriate age.

### Immunohistochemistry

Postnatal and adult mice were anaesthetized (using Avertin) and intracardially perfused with saline followed by 4% paraformaldehyde (PFA) in 0.1 M PB. After overnight post-fixation at 4 °C, brains were rinsed three times and sectioned at a 50–75-µm thickness with a Leica 1000s vibratome. Embryonic brains were collected in PBS and fixed in 4% PFA for 4 h at room temperature, rinsed three times with PBS, dehydrated in 30% sucrose-PBS, frozen in OCT compound and cut by cryostat (Leica, CM3050S) in 20–50-µm coronal sections. Early postnatal pups were anaesthetized using cold shock on ice and intracardially perfused with 4% PFA in PBS. Post-fixation was performed similarly to older mice. Postnatal mice aged 1–2 months were anaesthetized using Avertin and intracardially perfused with saline followed by 4% PFA in PBS; brains were post-fixed in 4% PFA overnight at 4 °C and subsequently rinsed

three times, embedded in 3% agarose-PBS and cut to a 50–100-µm thickness using a vibrating microtome (Leica, VT100S). Sections were placed in blocking solution containing 10% normal goat serum (NGS) and 0.1% Triton-X100 in PBS1X for 1 h, then incubated overnight at 4 °C with primary antibodies diluted in blocking solution. Sections were rinsed three times in PBS and incubated for 1 h at room temperature with corresponding secondary antibodies (1:500, Life Technologies). Sections were washed three times with PBS and incubated with DAPI for 5 min (1:5,000 in PBS, Life Technologies, 33342) to stain nuclei. Sections were dry-mounted on slides using Vectashield (Vector Labs, H1000) or Fluoromount (Sigma, F4680) mounting medium.

To perform molecular characterization of GeneX-CreER mouse lines, we stained 40-µm vibratome sections for CUX1 and CTIP2, that were imaged in a Nikon Eclipse 90i fluorescence microscope. Focusing on the somatosensory cortex, we counted tdTomato$^+$ cells in a column of around 300-µm width and determined their relative position along the dorso-ventral axis that goes from the ventricular surface (0) to the pia (100%). As a reference, CTIP2$^+$ and CUX1$^+$ regions were plotted as green and blue bars, where the upper limits correspond to the mean relative position of the dorsal-most positive cells, and the lower limits correspond to the mean relative position of the ventral-most positive cells. Grey areas in histograms correspond to the s.d. of those limits. The frequency of tdTomato$^+$ cells along the dorso-ventral axis was plotted in a histogram with a bin width of 5%. Number of cells: *Fezf2-CreER*, 2,781 cells; *Tcerg1l-CreER*, 185 cells; *Adcyap1-CreER*, 54 cells; *Tle4-CreER*, 2,737 cells; *Lhx2-CreER*, 1,380 cells; *Plexind1-CreER*, 809 cells; *Cux1-CreER*, 2,296 cells; *Tbr1-CreER*, 3,572 cells; *Tbr2-CreER* tamoxifen E16.5, 1,273 cells; *Tbr2-CreER* tamoxifen E17.5, 1,871 cells. For each line we quantified at least four sections from two embryos. Differences in cell numbers are due to differences in labelling density.

For colocalization determination, we obtained confocal z-stacks centred in layer 5 or 6 of the somatosensory cortex, of 320 × 320 × 40 µm$^3$ volumes. For all tdTomato$^+$ cells in the volume, we manually determined whether they were also positive for the desired markers by looking in individual z-planes. The percentage of positive cells was calculated for each area. Average number of tdTomato$^+$ cells quantified per staining: *Fezf2-CreER*, 314 cells in layer 5 and 472 in layer 6; *Tcerg1l-CreER*, 162 cells; *Adcyap-CreER*, 20 cells; *Tle4-CreER*, 157 cells in layer 5 and 1,081 in layer 6; *Lhx2-CreER*, 294 cells; *Plexind1-CreER*, 468 cells in layers 4 and 5a; *Cux1-CreER*, 761 cells; *Tbr1-CreER*, 858 cells; *Tbr2-CreER*, 1,380 cells. For each line we quantified at least four sections from at least two embryos. Differences in cell numbers are due to differences in labelling density.

### Antibodies

Anti-GFP (1:1,000, Aves, GFP-1020), anti-RFP (1:1,000, Rockland Pharmaceuticals, 600-401-379), anti-mCherry (1:500, OriGene AB0081-500), anti-mKATE2 for Brainbow 3.0 (a gift from D. Cai), anti-SATB2 (1:20, Abcam ab51502), anti-CTIP2 (1:100, Abcam 18465), anti-CUX1 (1:100, SantaCruz 13024), anti-LDB2 (1:200, Proteintech 118731-AP), anti-FOG2 (1:500, SantaCruz m-247), anti-LHX2 (1:250, Millipore-Sigma ABE1402) and anti-TLE4 (1:300, Santa Cruz sc-365406) were used.

### Validation of PyN driver lines

ViewRNA tissue Assay (Thermo Fisher Scientific) fluorescent in situ hybridization (FISH) was carried out as per the manufacturer's instructions on genetically identified PyNs expressing H2bGFP nuclear reporter (GeneX-CreER;LSL-H2bGFP) to validate the expression of PyN mRNA within Cre-recombinase dependent H2bGFP expressing cells in adult tissue (p24). Antibody validation with Cre-recombinase dependent reporter (GeneX-CreER;Ai14) was also used as it was available for use in adult tissue. For both FISH and antibody validation experiments, the percentage of total recombinase-dependent reporter-positive cells co-expressing PyN driver transcript or antibody was quantified.

## Viral injection and analysis

**Stereotaxic viral injection.** Adult mice were anaesthetized by inhalation of 2% isoflurane delivered with a constant air flow (0.4 l min$^{-1}$). Ketoprofen (5 mg kg$^{-1}$) and dexamethasone (0.5 mg kg$^{-1}$) were administered subcutaneously as preemptive analgesia and to prevent brain oedema, respectively, before surgery, and lidocaine (2–4 mg kg$^{-1}$) was applied intra-incisionally. Mice were mounted in a stereotaxic head-frame (Kopf Instruments, 940 series or Leica Biosystems, Angle Two). Stereotactic coordinates were identified (Supplementary Table 5). An incision was made over the scalp, a small burr hole drilled in the skull and brain surface exposed. Injections were performed according to the strategies delineated in Supplementary Table 5. A pulled glass pipette tip of 20–30 µm containing the viral suspension was lowered into the brain; a 300–400 nl volume was delivered at a rate of 30 nl min$^{-1}$ using a Picospritzer (General Valve Corp); the pipette remained in place for 10 min preventing backflow, prior to retraction, after which the incision was closed with 5/0 nylon suture thread (Ethilon Nylon Suture, Ethicon) or Tissueglue (3M Vetbond), and mice were kept warm on a heating pad until complete recovery.

**Systemic AAV injection.** *Foxp2-IRES-Cre* mice were injected through the lateral tail vein at 4 weeks of age with a 100 µl total volume of AAV9-CAG-DIO-EGFP (UNC Viral Core) diluted in PBS ($5 \times 10^{11}$ vg per mouse). Three weeks after injection, mice were transcardially perfused with 0.9% saline, followed by ice-cold 4% PFA in PBS, and processed for serial two-photon (STP) tomography.

## Viruses

AAVs serotype 8, 9, DJ PHP.eB or rAAV2-retro (retroAAV) packaged by commercial vector core facilities (UNC Vector Core, ETH Zurich, Biohippo, Penn, Addgene) were used as listed in Supplementary Table 5. In brief, for cell-type-specific anterograde tracing, we used either Cre- or Flp-dependent or tTA-activated AAVs combined with the appropriate reporter mouse lines[28] (Supplementary Table 7), or dual-tTA (Fig. 4 and Extended Data Fig. 10) to express EGFP, EYFP or mRuby2 in labelled axons. retroAAV-Flp was used to infect axons at their terminals[46] in target brain structures to label PyNs retrogradely according to the experiments detailed in Supplementary Table 5.

## Microscopy and image analysis

Imaging was performed using Zeiss LSM 780 or 710 confocal microscopes, Nikon Eclipse 90i or Zeiss Axioimager M2 fluorescence microscopes, or whole-brain STP tomography (detailed below). Imaging from serially mounted sections was performed on a Zeiss LSM 780 or 710 confocal microscope (CSHL St. Giles Advanced Microscopy Center) and Nikon Eclipse 90i fluorescence microscope, using objectives ×63 and ×5 for embryonic tissue, and ×20 for adult tissue, as well as ×5 on a Zeiss Axioimager M2 System equipped with MBF Neurolucida Software (MBF). Quantification and image analysis was performed using ImageJ/FIJI software. Statistics and plotting of graphs were done using GraphPad Prism 7 and Microsoft Excel 2010.

**Twenty-four-hour pulse-chase embryonic experiments.** For 24-hour pulse-chase embryonic experiments (Fig. 1, Extended Data Fig. 1), high-magnification insets are not maximum intensity projections. To observe the morphology of RGs, only a few sections from the $z$-plane in low-magnification images have been projected in the high-magnification images.

**Quantification and statistics related to progenitor fate-mapping.** Quantification for top panels in Fig. 1h, Extended Data Fig. 2m–o ($n$ = 5–6 from two litters): mean values, number of progenitors ± s.e.m. *$P$ < 0.05 (compared with bin M, RGs$^{Lhx2+}$), #$P$ < 0.05 (compared with bin M, RGs$^{Fezf2+}$), one-way ANOVA, Tukey's post-hoc test. †$P$ < 0.05 (compared

with RGs$^{Lhx2+}$ for corresponding bins), unpaired Student's $t$-test. Quantification for bottom panels in Fig. 1h, Extended Data Fig. 2m–o ($n$ = 3 from two litters): mean values for percentage total PyNs (S1) ± s.e.m. *$P$ < 0.05 (compared with PyNs$^{Lhx2+}$). Quantification for Fig. 1k, Extended Data Fig 2t: top panel: ($n$ = 3 from two litters): mean values, number of progenitors ± s.e.m. *$P$ < 0.0001 (compared with RGs$^{Lhx2+Fezf2-}$), unpaired Student's $t$-test. Bottom panel ($n$ = 3 from two litters): mean values, number of progenitors ± s.e.m. *$P$ < 0.05 (compared with rostral RGs$^{Lhx2+Fezf2-}$), #$P$ < 0.05 (compared with rostral RGs$^{Lhx2+Fezf2+}$), one-way ANOVA, Tukey's post-hoc test. †P < 0.05 (compared with RGs$^{Lhx2+/Fezf2-}$ for corresponding regions), unpaired Student's $t$-test.

**Target-specific cell depth measurement.** Cell depth analysis for retrogradely labelled projection-specific genetically identified PyNs (GeneX-CreER) were obtained using 5× MBF fluorescent widefield images of 65-µm thick coronal sections in MO and SSp-bfd. MO cell depths are presented in micrometres owing to the absence of a defined white matter border in frontal cortical areas and SSp-bfd depth ratio measurements were normalized to the distance from pia to white matter. For each condition we quantified at least four sections taken from two mice.

## Whole-brain STP tomography and image analysis

Perfused and post-fixed brains from adult mice were embedded in oxidized agarose and imaged with TissueCyte 1000 (Tissuevision) as described[48,49]. We used the whole-brain STP tomography pipeline previously described[48,49]. Perfused and post-fixed brains from adult mice, prepared as described above, were embedded in 4% oxidized-agarose in 0.05 M PB, cross-linked in 0.2% sodium borohydrate solution (in 0.05 M sodium borate buffer, pH 9.0–9.5). The entire brain was imaged in coronal sections with a 20× Olympus XLUMPLFLN20XW lens (NA 1.0) on a TissueCyte 1000 (Tissuevision) with a Chameleon Ultrafast-2 Ti:Sapphire laser (Coherent). EGFP/EYFP or tdTomato signals were excited at 910 nm or 920 nm, respectively. Whole-brain image sets were acquired as series of 12 ($x$) × 16 ($y$) tiles with 1 µm × 1 µm sampling for 230–270 $z$ sections with a 50-µm $z$-step size. Images were collected by two PMTs (PMT, Hamamatsu, R3896), for signal and autofluorescent background, using a 560-nm dichroic mirror (Chroma, T560LPXR) and band-pass filters (Semrock FF01-680/SP-25). The image tiles were corrected to remove illumination artifacts along the edges and stitched as a grid sequence[47,49]. Image processing was completed using ImageJ/FIJI and Adobe/Photoshop software with linear level and nonlinear curve adjustments applied only to entire images.

**Cell body detection from whole-brain STP data.** PyN somata were automatically detected from cell-type specific reporter lines (R26-LSL-GFP or Ai14) by a convolutional network trained as described previously[48]. Detected PyN soma coordinates were overlaid on a mask for cortical depth, as described[48].

**Axon detection from whole-brain STP data.** For axon projection mapping, PyN axon signal based on cell-type-specific viral expression of EGFP or EYFP was filtered by applying a square root transformation, histogram matching to the original image, and median and Gaussian filtering using Fiji/ImageJ software[50] so as to maximize signal detection while minimizing background auto-fluorescence, as described before[51]. A normalized subtraction of the autofluorescent background channel was applied and the resulting thresholded images were converted to binary maps. Three-dimensional rendering was performed on the basis of binarized axon projections and surfaces were determined based on the binary images using Imaris software (Bitplane). Projections were quantified as the fraction of pixels in each brain structure relative to each whole projection.

**Axon projection cartoon diagrams from whole-brain STP data.** To generate cartoons of axon projections for a given driver line, axon detection

outputs from all individual experiments were compared (sorting the values from high to low), and analysed side-by-side with low-resolution image stacks (and the CCFv3 registered to the low-resolution dataset for brain area definition) to get a general picture of the injection, as well as high-resolution images for specific brain areas.

**Registration of whole-brain STP image datasets.** Registration of brain-wide datasets to the Allen reference Common Coordinate Framework (CCF) version 3 was performed by 3D affine registration followed by a 3D B-spline registration using Elastix software[52], according to established parameters[52]. For cortical depth and axon projection analysis, we registered the CCFv3 to each dataset so as to report cells detected and pixels from axon segmentation in each brain structure without warping the imaging channel.

## In vitro electrophysiology

**Brain slice preparation.** Mice (>P30) were anaesthetized with isoflurane, decapitated, brains dissected out and rapidly immersed in ice-cold, oxygenated, artificial cerebrospinal fluid (section ACSF: 110 mM choline-Cl, 2.5 mM KCl, 4 mM $MgSO_4$, 1 mM $CaCl_2$, 1.25 mM $NaH_2PO_4$, 26 mM $NaHCO_3$, 11 mM D-glucose, 10 mM Na ascorbate, 3.1 Na pyruvate, pH 7.35, 300 mOsm) for 1 min. Coronal cortical slices containing somatomotor cortex were sectioned at a 300-μm thickness using a vibratome (HM 650 V; Microm) at 1–2 °C and incubated with oxygenated ACSF (working ACSF; 124 mM NaCl, 2.5 mM KCl, 2 mM $MgSO_4$, 2 mM $CaCl_2$, 1.25 mM $NaH_2PO_4$, 26 mM $NaHCO_3$, 11 mM D-glucose, pH 7.35, 300 mOsm) at 34 °C for 30 min, and subsequently transferred to ACSF at room temperature (25 °C) for more than 30 min before use. Whole-cell patch recordings were directed to the somatosensory and motor cortex, and the subcortical whiter matter and corpus callosum served as primary landmarks according to the atlas (Paxinos and Watson Mouse Brain in Stereotaxic Coordinates, 3rd edition).

**Patch-clamp recording in brain slices.** Patch pipettes were pulled from borosilicate glass capillaries with filament (1.2 mm outer diameter and 0.69 mm inner diameter; Warner Instruments) with a resistance of 3–6 MΩ. The pipette recording solution consisted of 130 mM potassium gluconate, 15 mM KCl, 10 mM sodium phosphocreatine, 10 mM HEPES, 4 mM ATP·Mg, 0.3 mM GTP and 0.3 mM EGTA (pH 7.3 adjusted with KOH, 300 mOsm). Dual or triple whole-cell recordings from tdTomato[+] and EGFP[+] PyNs were made with Axopatch 700B amplifiers (Molecular Devices) using an upright microscope (Olympus, BX51) equipped with infrared-differential interference contrast optics (IR-DIC) and a fluorescence excitation source. Both IR-DIC and fluorescence images were captured with a digital camera (Microfire, Optronics). All recordings were performed at 33–34 °C with the chamber perfused with oxygenated working ACSF.

Recordings were made with two MultiClamp 700B amplifiers (Molecular Devices). The membrane potential was maintained at −75 mV in the voltage clamping mode and zero holding current in the current clamping mode, without the correction of junction potential. Signals were recorded and filtered at 2 kHz, digitalized at 20 kHz (DIGIDATA 1322A, Molecular Devices) and further analysed using the pClamp 10.3 software (Molecular Devices) for intrinsic properties.

## Reporting summary

Further information on research design is available in the Nature Research Reporting Summary linked to this paper.

## Data availability

Raw and stitched whole-brain STP imaging data are available from the BICCN Brain Image Library (BIL) (http://www.brainimagelibrary.org/download.html) at the Pittsburgh Supercomputing Center. Antero-grade projection and cell distribution datasets can be visualized on the Mouse Brain architecture website (http://brainarchitecture.org/cell-type/projection and http://brainarchitecture.org/cell-type/density, respectively) as detailed in Supplementary Tables 4, 5.

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

**Acknowledgements** We thank R. Palmiter for providing the *FoxP2-IRES-Cre* mouse line; Y. Yoshida for providing the *Sema3E-CreER* mouse line; L. Li at CSHL for help with generation of knock-in mice; S. M. Kelly for contributing to the generation and characterization of the *Tis21-2A-CreER* mouse line; Y. Kim for providing the digital flatmap of the mouse cortex; F. Boato for help in mouse breeding and brain tissue preparation; K. Kelly for project management; H. Kondo for providing anatomical datasets; S. Srivas for help with *Tbr1* driver data; and B. Huo, X. Li, F. Xu and K. Ram for depositing data to the repository and administering and uploading data to the Brain Architecture portal. We thank CSHL Animal Resources for mouse husbandry. This work was supported in part by the NIH grants 5R01MH101268-05 and 5U19MH114821-03 to Z.J.H. and P.A.; 1S10OD021759-01 to Z.J.H.; and the CSHL Robertson Neuroscience Fund to Z.J.H. P.O. is supported by NIH U01 MH114824-01. D.H. was supported by the Human Frontier Science Program long-term fellowship LT000075/2014-L and NARSAD Young Investigator grant no. 26327. R.R. was supported by the NRSA F31 Predoctoral Fellowship 5F31MH114529-03. J.T. and J.L. were supported by the NRSA F30 Medical Scientist Predoctoral Fellowships 5F30MH097425-03 and 5F30MH108333.

**Author contributions** Z.J.H. and P.A. conceived the study and obtained funding. Z.J.H. coordinated the study. Z.J.H., P.A., K.S.M. and D.H. designed the experiments. M.H., P.W. and E.G. designed and generated all new knock-in mouse lines. K.S.M., X.A., D.H., R.P., J.H., D.J.D.B., J.T., J.M.L., R.R., W.G., E.G., P.W., M.H., J.A.H. and E.S. conducted mouse breeding, anatomy, immunohistochemistry, imaging and quantification. K.S.M., W.G., X.A., G.K., J.H. and R.P.

conducted virus injection experiments. J.L. performed the patch-clamp recording in brain slice. K.S.M., J.H., R.P., X.A., W.G., A.N., P.P.M. and P.O. performed whole-brain STP tomography and analysis of cell distribution and axon projection patterns. Z.J.H. wrote the manuscript with contributions and edits from P.A., K.S.M., D.H., W.G., M.H., J.L. and X.A.

**Competing interests** J.A.H. is currently employed by Cajal Neuroscience. The remaining authors declare no competing interests.

**Additional information**
**Correspondence and requests for materials** should be addressed to Z. Josh Huang.

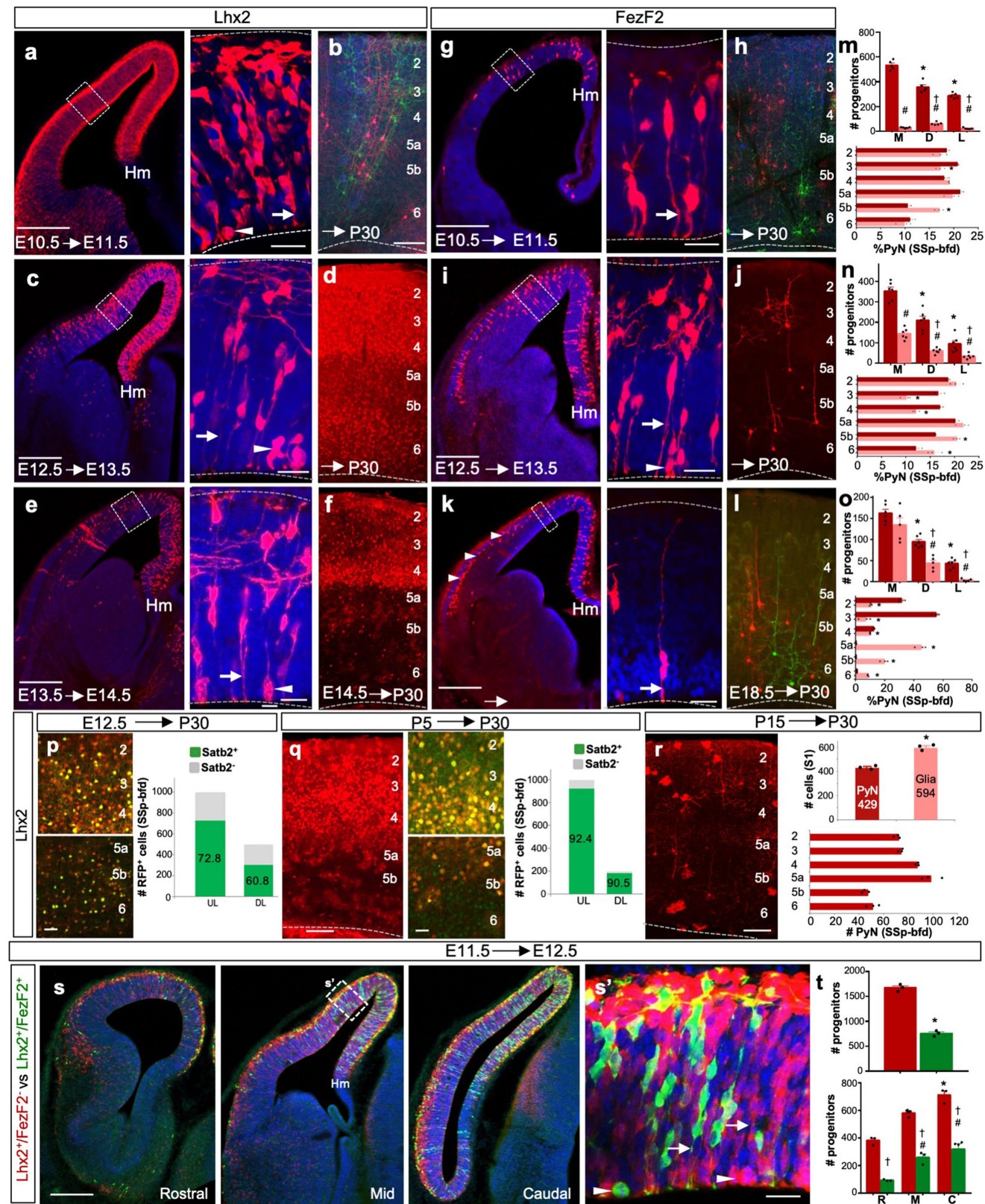

**Extended Data Fig.1** | See next page for caption.

**Extended Data Fig. 1 | Fate-mapping using _Lhx2_ and _Fezf2_ driver lines.**
**a**, 24-hour pulse-chase in a E10.5 _Lhx2-CreER;Ai14_ embryo densely labelled RGs$^{Lhx2+}$ throughout the dorsal neuroepithelium, sharply ending at the cortex–hem boundary (Hm); magnified view shows RGs at different stages of the cell cycle, with endfeet (arrow) or dividing somata (arrowhead) at the ventricle wall (dashed lines). **b**, E10.5 RGs generate PyNs across layers in P30 neocortex in _Lhx2-CreER;RGBow_ mouse[13] (Supplementary Table 1). **c, d** are duplicated from Fig. 1c, d for comparison. **e**, Same experiment as in **a** done at E13.5. RGs$^{Lhx2+}$ remain distributed in a medial$^{high}$-lateral$^{low}$ gradient along the dorsal pallium but at a reduced density compared to earlier stages. **f**, E14.5 RGs$^{Lhx2+}$ generated PyNs with lower density in L5/6, higher density in L2-4, and highest density in L4, suggesting more restricted fate potential. **g**, 24-hour pulse-chase in a E10.5 _Fezf2-CreER;Ai14_ embryo. The spatial extent of RGs$^{Fezf2+}$ in neuroepithelium is more restricted compared to RGs$^{Lhx2+}$ at this stage; magnified view shows RG endfoot (arrow) at the ventricle (dashed lines). Note the sparsity of dividing RGs. **h**, Fate-mapping of E10.5 RGs$^{Fezf2+}$ in a _Fezf2-CreER;RGBow_ mouse labelled PyN progeny across all cortical layers. **i, j** are duplicated from Fig. 1e, f for comparison. **k**, Same experiment as in **g** done at E13.5. Only sparse RGs$^{Fezf2+}$ remain in medial pallium at this time, when _Fezf2_ expression shifts to postmitotic PyNs (arrowheads); magnified view shows a remaining RG (arrow) at the ventricle wall (dashed line). **l**, Fate-mapping at E18.5 in _Fezf2-CreER;RGBow_ labelled only L5b/L6 PyNs in mature cortex. **m**, Upper panel, quantification of E10.5 RGs$^{Lhx2+}$ (red) (in **a**) versus RGs$^{Fezf2+}$ (pink) (in **g**) distributed across the cortical primordium, divided into medial (M), dorsal (D) and lateral (L) bins of equal length. Lower panel, laminar distribution of PyNs generated by RGs$^{Lhx2+}$ (red) or RGs$^{Fezf2+}$ (pink) at E10.5, shown as percentage of total PyNs in S1 barrel cortex. **n**, Same quantification of RGs$^{Lhx2+}$ and RGs$^{Fezf2+}$ as in **m** except done at E12.5. **o**, Same quantification of

RGs$^{Lhx2+}$ and RGs$^{Fezf2+}$ as in **m** except done at E13.5 (upper panel), and the laminar distribution of their PyN progeny fate-mapped at E14.5 (for RGs$^{Lhx2+}$) and E18.5 (for RGs$^{Fezf2+}$) (lower panel). **p**, Fate-mapping E12.5 RGs$^{Lhx2+}$ to mature cortex labelled PyN progeny that are SATB2$^+$ (IT class, 66.8%) as well as SATB2$^-$ (non-IT class, 33.2%). Of these, 72.8% of L2-4 (UL; 728 of 1,000 cells) and 60.8% of L5-6 (DL; 304 of 500 cells; n=3 brains from 2 litters) PyNs are of IT-type. **q**, P5 tamoxifen induction in _Lhx2-CreER;Ai14_ shows dense labelling of L2-4 PyNs and sparse labelling in L5/6 in P28 cortex (Left panel) Most labelled PyNs are of the IT class expressing SATB2 (middle and right panels; UL, 92.4% - 924 of 1,000 cells; DL, 90.5%-181 of 200 cells; n=3 brains from 2 litters). **r**, P15 tamoxifen induction in _Lhx2-CreER;Ai14_. (Upper) Quantification in SSp-bfd shows 58% cells are glia (Mean values for number of cells in SSp-bfd ± SEM. *P < 0.05 compared to PyN number, unpaired Student's _t_-test). (Lower) Labelled PyNs are distributed across cortical layers with more labelling in L5a and L4 (n=3 brains from 2 litters). **s**, Presence of RGs$^{Lhx2+Fezf2-}$ (RFP) and RGs$^{Lhx2+Fezf2+}$ (EGFP) at E11.5 throughout cortical primordium revealed by intersection/subtraction fate-mapping with 24-hour pulse-chase in _Lhx2-CreER;Fezf2-Flp;IS_ mice, schematized in Fig. 1i (Also see Fig. 1j–n). These progenitors distribute in a medial high-lateral low gradient along the dorsal neuroepithelium, ending at the cortex–hem boundary (Hm). Rostral, Mid and Caudal sectioning levels show a caudal high-rostral low distribution of RGs$^{Lhx2+Fezf2-}$ and RGs$^{Lhx2+Fezf2+}$. **s'**, Magnified view shows RGs$^{Lhx2+Fezf2-}$ and RGs$^{Lhx2+Fezf2+}$ at multiple cell cycle stages with endfeet (arrow) and dividing soma (arrowhead) at ventricle wall (dashed line). **t**, Upper, total number of RGs$^{Lhx2+Fezf2+}$ (green) is approximately half that of RGs$^{Lhx2+Fezf2-}$ (red). Lower, quantification of RGs$^{Lhx2+Fezf2-}$ versus RGs$^{Lhx2+Fezf2+}$ at rostral (R), mid-level (M) and caudal (C) sections revealed their caudal$^{high}$-rostral$^{low}$ distribution. Scale bars: 20μm (high magnification) in **a, c, e, g, i, k, s'**; 100μm for all other panels.

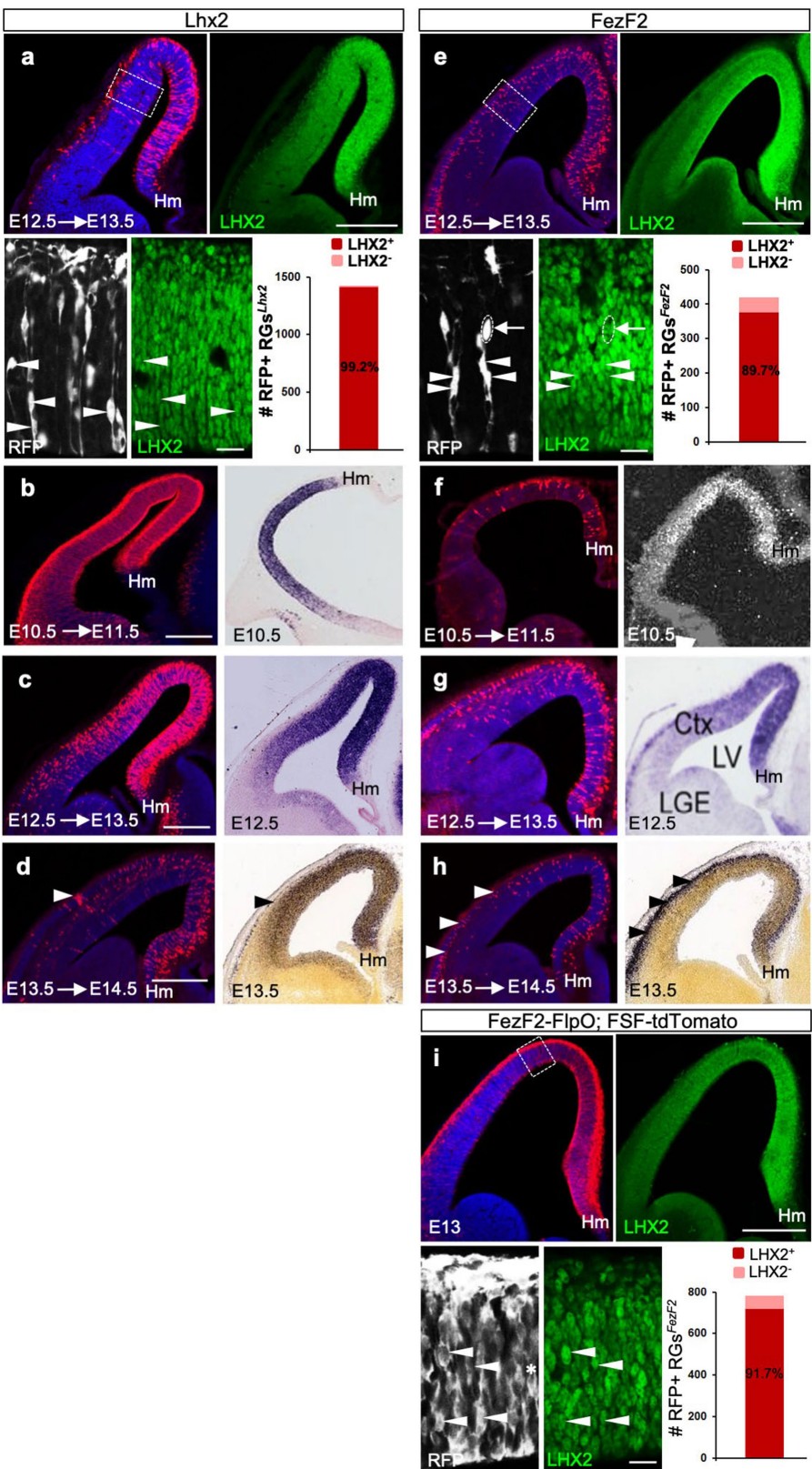

**Extended Data Fig. 2** | See next page for caption.

**Extended Data Fig. 2 | *Lhx2* and *Fezf2* driver lines precisely recapitulate endogenous developmental expression patterns. a**, Anti-LHX2 immunohistochemistry on a 24-hour pulse-chase in a E12.5 *Lhx2-CreER;Ai14* embryo revealed that 99.2% RGs$^{Lhx2+}$ express LHX2 (1410 of 1422 cells; n = 4 embryos from 2 litters). High magnification images show colocalization (arrowheads). Note the medial$^{high}$-lateral$^{low}$ gradient. **b**, 24-hour pulse-chase in a E10.5 *Lhx2-CreER;Ai14* embryo densely labelled RGs throughout the dorsal neuroepithelium, ending medially at the cortex–hem boundary (Hm). This expression recapitulated *Lhx2* mRNA in-situ hybridization at E10.5 (reproduced/adapted with permission from *Development*)[68]. **c**, Same experiment as in **b** done at E12.5. The medial$^{high}$-lateral$^{low}$ gradient of RGs$^{Lhx2+}$ is highly similar to *Lhx2* mRNA expression (reproduced/adapted with permission from *Development*)[69]. **d**, Same experiment as in **b** done at E13.5. Note the reduction in RGs$^{Lhx2+}$ and concomitant increase in post-mitotic cells (arrowheads), in both the *Lhx2* driver line and mRNA expression (ISH Data: Allen Brain Atlas: Developing Mouse Brain). **e**, anti-LHX2 immunohistochemistry on a 24-hour pulse-chase in a E12.5 *Fezf2-CreER;Ai14* embryo reveals that 89.7% RGs$^{Fezf2+}$ express LHX2 (376 of 419 cells; n = 4 embryos from 2 litters); high magnification images show colocalization (arrowheads) and non-colocalization (arrow and encircled). **f**, 24-hour pulse-chase in a E10.5 *Fezf2-CreER;Ai14* embryo. The spatial extent of RGs$^{Fef2+}$ is restricted compared to RGs$^{Lhx2+}$ at this stage. This sparse labelling recapitulated *Fezf2* mRNA expression at E10.5 (reproduced/adapted with permission from *J Comp Neurol*)[70]. **g**, Same experiment as in **f** done at E12.5. The medial$^{high}$-lateral$^{low}$ distribution gradient of RGs$^{Fezf2+}$ is also seen with *Fezf2* expression at this stage. Note RGs$^{Fezf2+}$ are sparsely labelled compared to RGs$^{Lhx2+}$ using in-situ hybridization at E12.5 (reproduced/adapted via Open Access from *Neural Dev*)[71]. **h**, Same experiment as in **f** done at E13.5. The drastic decrease of *Fezf2* expression in RGs$^{Fezf2+}$ accompanied by an increase in post-mitotic PyNs (arrowheads) is comparable in both *Fezf2-CreER; Ai14* and *Fezf2* in-situ data (ISH Data: Allen Brain Atlas: Developing Mouse Brain). **i**, anti-LHX2 immunohistochemistry on E13 *Fezf2-FlpO;FSF-tdTomato* embryo reveals that 91.7% RGs$^{Fezf2+}$ express LHX2 (718 of 783 cells; n = 4 embryos from 2 litters). Similar to **e**. high magnification images show colocalization (arrowheads) and non-colocalization (asterisk). Note the medial$^{high}$-lateral$^{low}$ gradient. Scale bars for low mag images = 100μm in **a-i**. Scale bars for high mag images = 20μm in **a**, **e**, **i**.

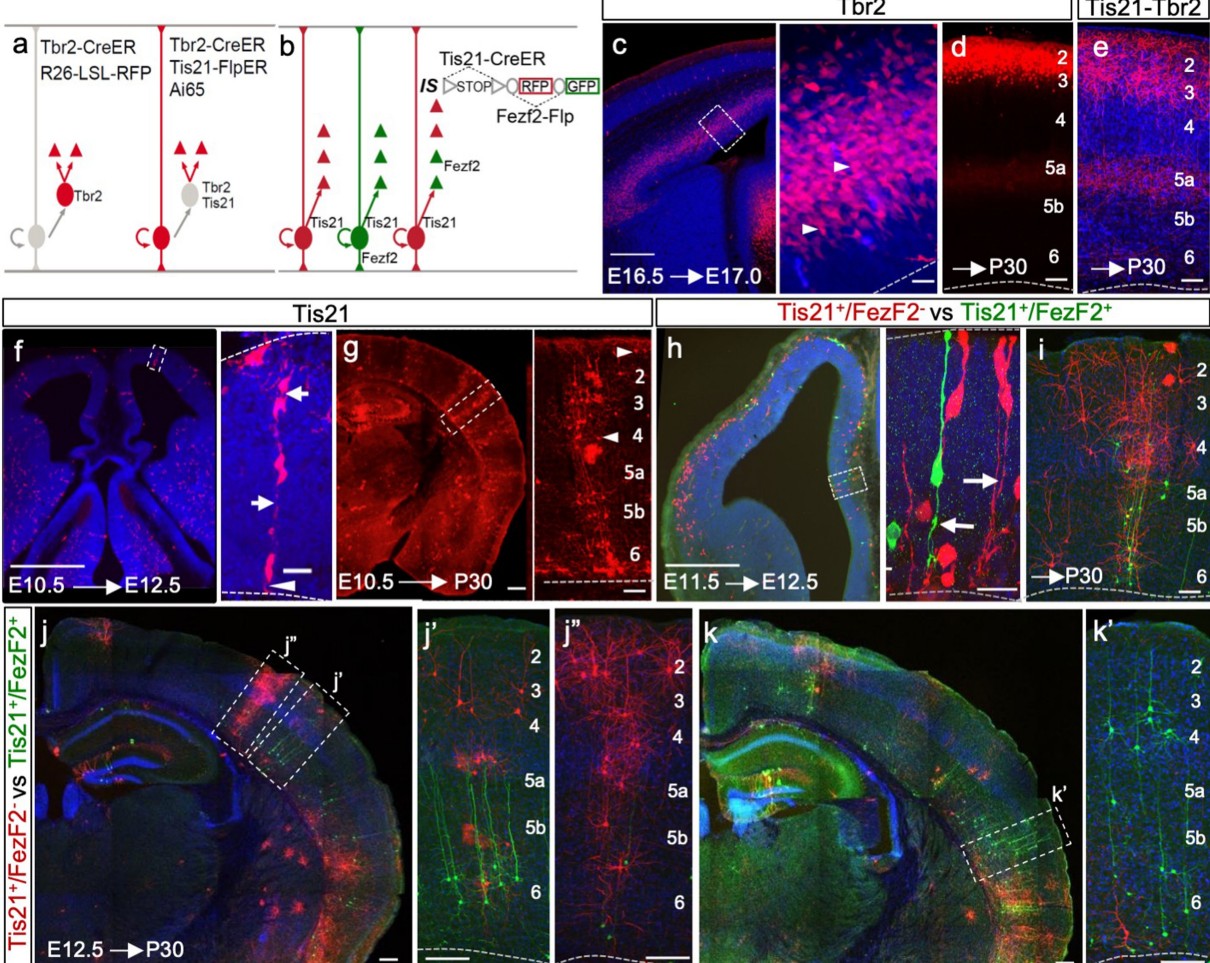

**Extended Data Fig. 3 | Fate-mapping neurogenic and intermediate progenitors. a**, Fate-mapping strategy for intermediate progenitors (IPs) and indirect neurogenesis, used in **c-e**. *Tbr2*-CreER labels an IP and all of its progeny with a fluorescent marker when combined with Ai14 (Left). The intersection of *Tis21-CreER* and *Tbr2-FlpER* specifically targets neurogenic IPs when combined with the Ai65 intersectional reporter (Right). **b**, Simultaneous fate-mapping of different molecularly defined neurogenic RGs using an intersection/ subtraction reporter (*IS*) combined with *Tis21-CreER* and *Fezf2-Flp* drivers. This scheme is used in **h-k**. In RGs$^{Tis21+Fezf2-}$, Cre activates RFP expression only. In RGs$^{Tis21+Fezf2+}$, Cre and Flp recombinations remove the RFP cassette and activate EGFP expression. At a later stage when *Fezf2* is only expressed in postmitotic deep layer PyNs, *Tis21-CreER* in RGs activates RFP expression in all of its progeny, but RFP is then switched to EGFP only in *Fezf2*+ PyNs expressing Flp. **c**, E16.5 IPs densely labelled by 12-hour pulse-chase in *Tbr2-CreER;Ai14* mice; magnified view shows IP somata (arrowhead) away from the lateral ventricle (dashed line) lacking radial fibers and endfeet. **d**, Fate-mapping E16.5 IPs in *Tbr2-CreER;Ai14* mice labels PyNs in L2-3 cortex at P28. **e**, Intersectional fate-mapping of neurogenic IPs at E16.5 in *Tis21-CreER;Tbr2-FlpER;Ai65* mice, as depicted in **a**, labelled L2-3 PyN progeny in P28 cortex. **f**, 48-hr pulse-chase in E10.5 *Tis21-CreER;Ai14* embryo labels *Tis21*+ neurogenic progenitors (nRGs) and their postmitotic progeny throughout the neural tube, including dorsal pallium (high magnification). Self-renewing RGs are identified by their endfeet at the ventricular surface (arrowheads) and radial fibers (arrow). **g**, Fate-mapping of E10.5 nRGs to mature cortex reveals PyNs are distributed throughout cortical layers. Note that multipolar GABAergic interneurons (some in layer 1) derived from subpallium nRGs are also labelled (arrowheads). **h**, The presence of nRGs$^{Fezf2-}$ and nRGs$^{Fezf2+}$ at E11.5 is revealed by intersection/ subtraction fate-mapping with 24-hour pulse-chase in *Tis21-CreER;Fezf2-Flp;IS* mice, schematized in **b**; magnified view shows RFP-labelled nRGs$^{Fezf2-}$ and EGFP-labelled nRGs$^{Fezf2+}$. **i**, Fate-mapping E11.5 nRGs using *Tis21-CreER; Fezf2-Flp;IS* mice. The mixed RFP and EGFP clone is likely to have derived from a nRG$^{Fezf2-}$, which activated RFP expression in all progeny and EGFP expression was then switched on only in *Fezf2*+ postmitotic deep layer PyNs expressing Flp. **j, k**, More examples of differential fate-mapping of nRGs$^{Fezf2-}$ and nRGs$^{Fezf2+}$ from E12.5 to the mature cortex using the scheme in **b**. The majority of clones consist of mixed RFP and EGFP PyNs (j'), and rarely RFP-only (j") or EGFP-only PyNs (k). RFP-only clones (1 of 29) probably derive from nRG$^{Fezf2-}$ whose progeny were all *Fezf2*– (j"). EGFP-only clones (2 of 29) are derived from nRGs$^{Fezf2+}$, suggesting multipotency of RGs$^{Fezf2+}$ (k). Mixed RFP/EGFP clones are most prominent and are likely to result from Cre activation of RFP in nRGs$^{Fezf2-}$ and subsequent Flp activation of EGFP in *Fezf2*+ L5/6 postmitotic PyNs (i,j'). Scale bars: 500μm in **d**, **j**, **k**; 100μm in **d** (high mag), **e**, **f**, **g**, **h**, **i**, **j'**,**j"**, **k** (high mag); 20μm in **c**, **e**, **h**.

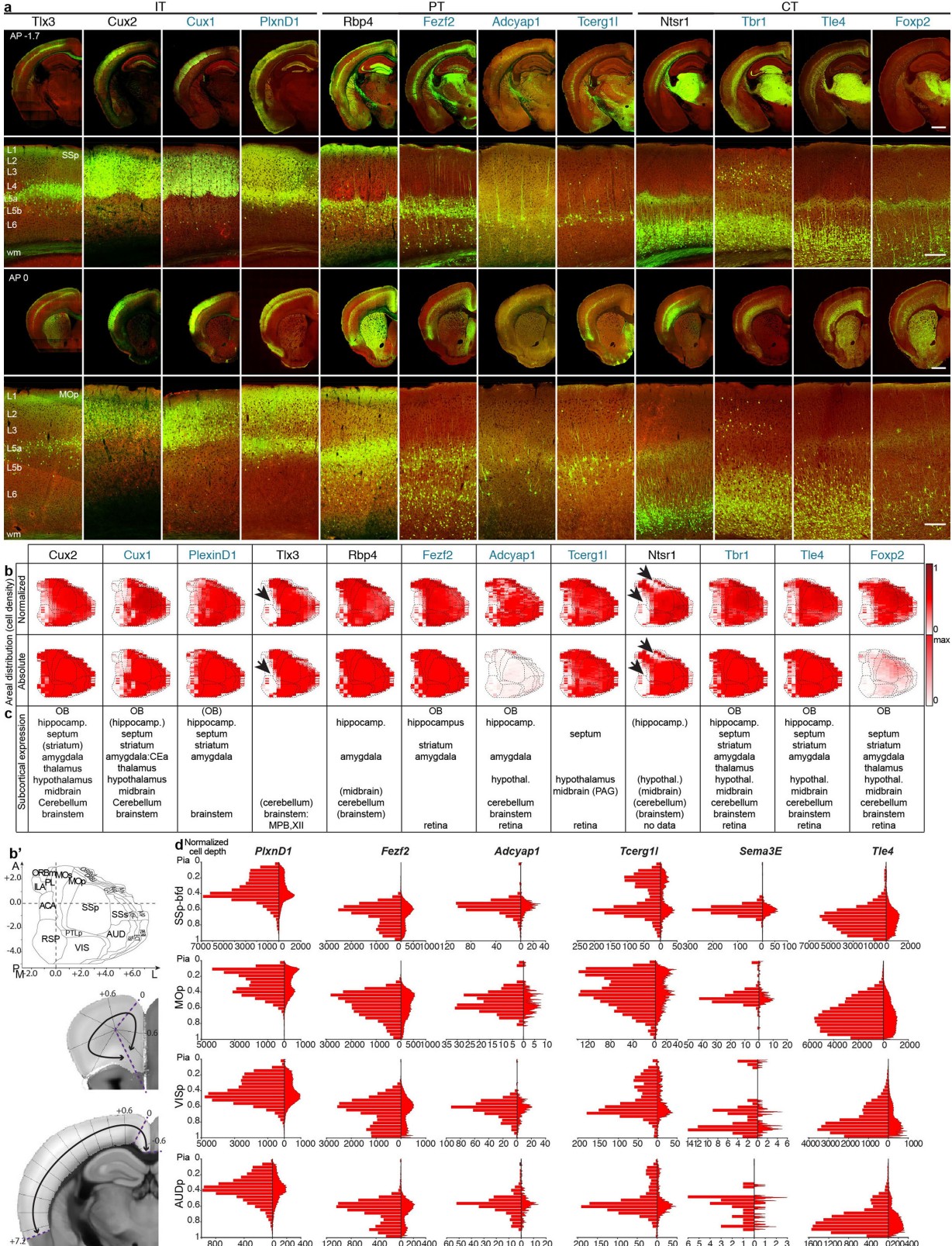

**Extended Data Fig. 4** | See next page for caption.

**Extended Data Fig. 4 | Comparison of new with existing driver lines in terms of areal and laminar patterns. a**, Side-by-side comparison of Cre recombination patterns from 8 mouse driver lines characterized in this study (blue font) and 4 existing driver lines (black font) visualized through reporter expression (green; background autofluorescence in red), grouped according to IT, PT and CT projection classes. First row: coronal hemisections at Bregma -1.7 mm. Second row: Image panel showing cortical depth detailing cell body distribution pattern of PyN subpopulations within SSp-bfd taken from the hemisection above at level Bregma -1.7 mm. Third row: coronal hemisections at Bregma 0 mm. Image panel showing cortical depth detailing cell body distribution pattern of PyN subpopulations within MOp taken from the hemisection above at level Bregma 0 mm. For comparison to the PT driver *Sim1-Cre* transgenic line, see reference 37. **b**, Cortex-wide distribution patterns of PyN subpopulations viewed as cortical flatmaps in a side-by-side comparison of 8 newly generated (blue font) and 4 existing driver lines (black font): first row, normalized for each dataset's total number of cells detected; second row, absolute scale per flatmap grid area, with maximum number of cells for any PyN subpopulation. Arrowheads indicate gaps in expression and labelling. **b'** shows cortical flat-mapping coordinate space and two exemplary coronal hemisections describing the demarcations used to generate the cortical grid for flatmapping[48]. **c**, Overview of brain-wide cell body distribution patterns for each driver line. This table provides an overall impression of the recombination patterns in major adult brain regions in selected lines. **d**, Histograms showing normalized laminar distribution for six genetically targeted PyN subpopulations by cortical area. Brain-wide cortical depth quantification was performed based on cell detection by convolutional networks from GeneX-CreER driver lines crossed to Ai14 (R26-LSL-tdTomato), R26-LSL-h2b-GFP or Snap25-LSL-EGFP reporters and induced at the ages specified in Fig. 2, and P7 for Sema3E. The normalized cortical depth (0-1) was divided into 24 bins for the left histogram and 124 bins for the right plot in each panel. Abbreviations explained in the Supplementary Information. Scale bars: Last panel of first and third rows applies to all hemisections, 1mm; last panel of second and fourth rows applies to all cortical depth image panels, 200µm.

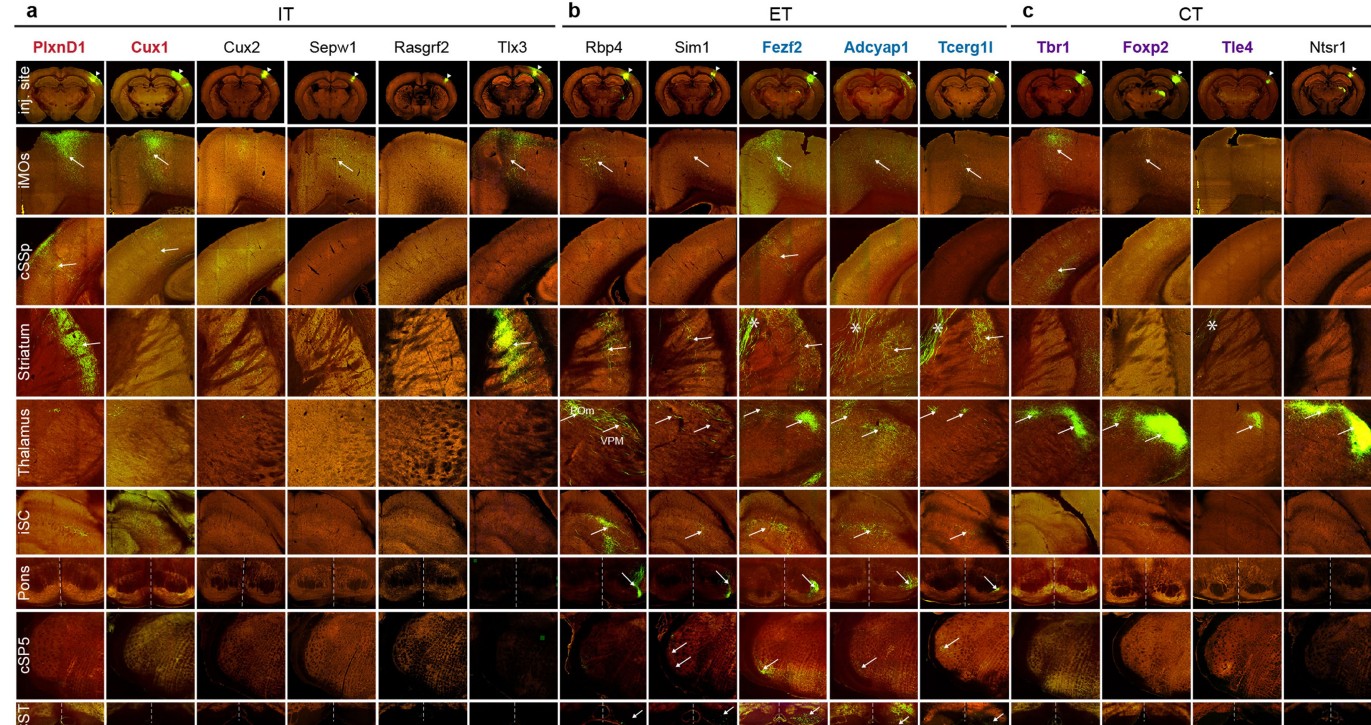

**Extended Data Fig. 5 | Comparison of new with existing driver lines in terms of axon projections from SSp-bfd somatosensory cortex. a-c**, STP images at the SSp injection site (first row, arrow head) and at selected subcortical projection targets for eight driver lines characterized in this study (coloured gene names code for IT-red, PT-blue and CT-purple) compared to seven existing driver lines (black gene names), with EGFP or EYFP expression from Cre-activated viral vector (green) and background autofluorescence (red). Arrows point to axons. **a**, IT drivers project to cortical and striatal targets. PyNs$^{Plxnd1}$ project bilaterally to cortex and striatum; PyNs$^{Cux1}$ project bilaterally to cortex but not to striatum. **b**, PT drivers project to many corticofugal targets including brainstem and spinal cord. PyNs$^{Fezf2}$, PyNs$^{Adcyap1}$ and PyNs$^{Tcerg1l}$ project to multiple ipsilateral targets and to the contralateral brainstem (arrows). **c**, CT drivers project predominantly to the thalamus. PyNs$^{Tbr1}$ project bilaterally to cortex and to ipsilateral thalamus, PyNs$^{Foxp2}$ and PyNs$^{Tle4}$ project to the ipsilateral cortex and thalamus. Scale bars: first row in c (applies to first row), 1 mm; second to eighth rows in c (applies to each respective row), 200 μm; CST panel (bottom row) in c applies to entire row, 100 μm. Asterisks in b & c indicate presence of passing fibers. A side-by-side list of axon projection matrix for all these lines is presented in Fig. 3d.

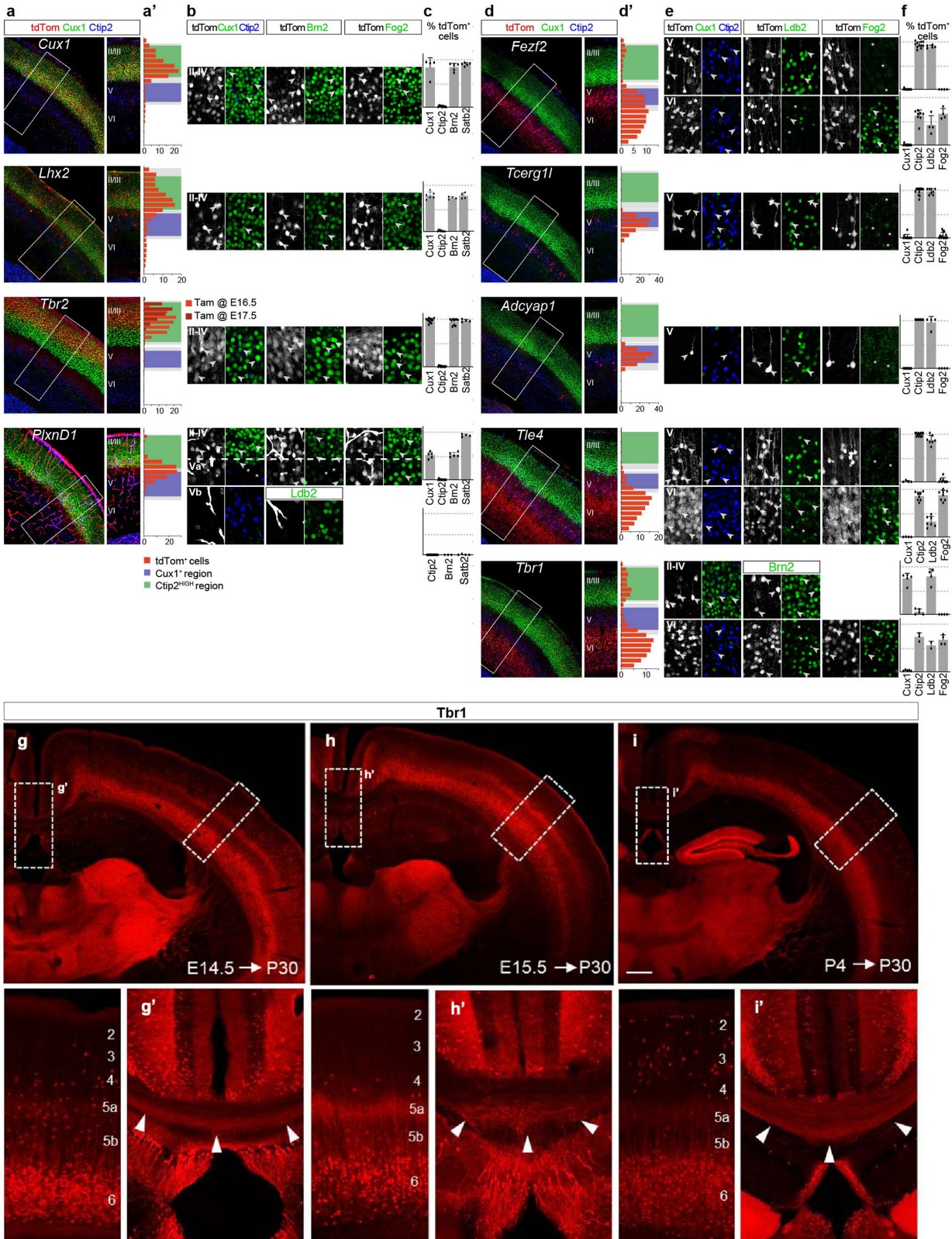

**Extended Data Fig. 6** | See next page for caption.

**Extended Data Fig. 6 | Molecular validation and developmental characterization of PyN driver lines. a, d**, Low magnification images of sections of somatosensory cortex (SSp) at P7 stained with antibodies against CTIP2, CUX1, and tdTomato from *PyN-CreER;Ai14* mice induced with tamoxifen. Inset to the right shows markers and Tomato⁺ cell distributions across layers. *Fezf2-*, *Tcerg1l-*, *Adcyap1-* and *Tle4*-CreER;Ai14 were induced at E16.5 and collected at P7. *Tbr1-*, *Cux1-* and *Plxnd1*-CreER;Ai14 were induced at P4 and collected at P7. *Tbr2*-CreER;Ai14 was induced at E16.5 (not shown) or E17.5 and collected at P7. *Lhx2*-CreER;Ai14 was induced at P3 and collected at P7. **a', d'**, Histograms showing radial distribution of Tomato⁺ cells in the cortical plate, in the region corresponding to SSp. In brief, in CUX1- and CTIP2-stained sections, Tomato⁺ cell depths relative to the thickness of the cortex were measured, as well as the limits of the areas occupied by CUX1⁺ or CTIP2^HIGH cells, shown in green (layers 2-4) and blue (layer 5b) bars, respectively (average relative values for the same sections, gray shading corresponds to 1 SD). For *Tbr2*-CreER;Ai14, mice induced at E16.5 or E17.5 were quantified separately, showing the later induction (darker red) results in more superficial labelling. Quantifications were made from 4-10 sections from 2-3 different mice for each line. **b, e**, Magnification of Tomato⁺ cells in sections co-stained against CUX1 and CTIP2, LDB2 (enriched in PT), FOG2 (expressed in CT), BRN2, or SATB2 (expressed in IT). Arrowheads show double-positive cells; asterisks show Tomato⁺ cells not expressing the marker. **c, f**, Percentage of Tomato⁺ cells stained with each antibody. Quantifications were done in equivalent areas (320 μm by 320 μm) within the SSp centered in the specified layers. Each dot is an area from a different section, for which the percentage of double positive cells was calculated. Bars are mean+SD. Quantifications were made from 4-8 sections from 2-3 different mice for each line. *Tbr2*-CreER;Ai14 labelled CUX1⁺, SATB2⁺, BRN2⁺ IT in the most superficial layers 2-3, irrespective of their induction time. *Lhx2*-CreER;Ai14 and *Cux1*-CreER;Ai14 labelled CUX1⁺, SATB2⁺, BRN2⁺ IT deeper in layers 2-3. *Plxnd1*-CreER;Ai14 labelled SATB2⁺, BRN2⁺ cells in layer 5A, as well as CUX1⁺, SATB2⁺, BRN2⁺ cells in layer 4. No cells were found in layer 5B. *Tcerg1l*-CreER;Ai14 and *Adcyap1*-CreER;Ai14 labelled sparse LDB2⁺, CTIP2⁺ PT in layer 5. *Fezf2*-CreER;Ai14 extensively labelled PT in layer 5 that were LDB2⁺, CTIP2⁺, as well as some CT in layer 6 expressing CTIP2 and FOG2. *Tle4*-CreER;Ai14 and *Tbr1*-CreER;Ai14 labelled CT expressing FOG2 and CTIP2 (and lower levels of LDB2) in layer 6. *Tle4*-CreER;Ai14 also labelled some LDB2⁺, CTIP2⁺ cells in layer 5 (PT), whereas *Tbr1*-CreER;Ai14 also labelled some CUX1⁺, BRN2⁺ IT in layer 2/3. **g**, Fate-mapping of PyNs^Tbr1 using *Tbr1*-CreER; Ai14 mice. Tamoxifen induction at E14.5 densely labelled L6 CT cells with minor labelling of cells in layers 3-5. **h**, Tamoxifen induction at E15.5 labelled L6 CT cells. **i**, Tamoxifen induction at P4 labelled L6 CT cells and also a subset of L2/3 cells. A subset of adult PyNs^Tbr1 labelled from E14.5, E15.5 and P4 induction project to contralateral cortex via the corpus callosum (arrowheads, **g', h', i'**). Scale bars: **g-i**, low magnification, 500 μm; high magnification, 100 μm.

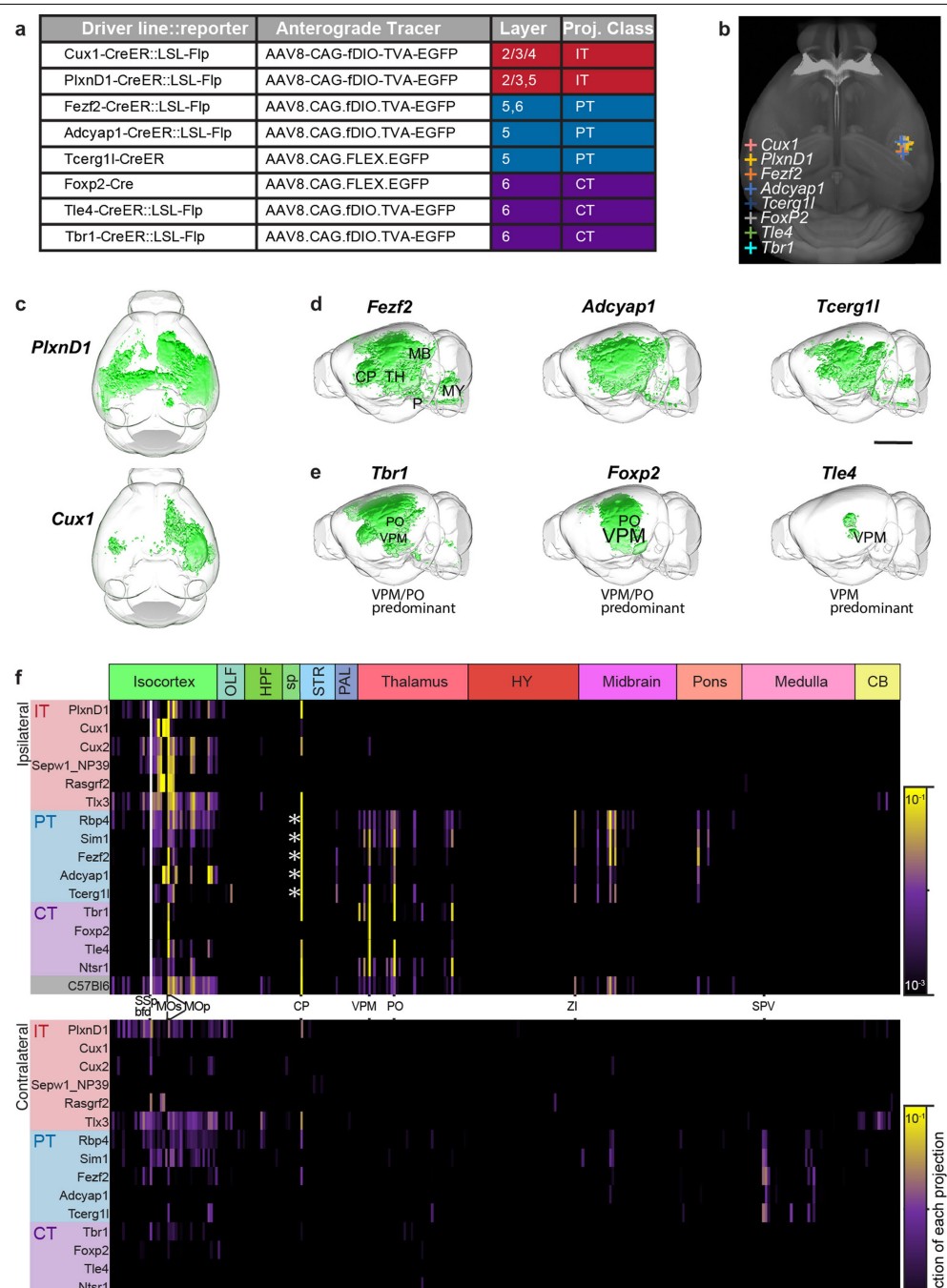

**Extended Data Fig. 7 | Anterograde tracing, registration to CCFv3 and analysis of PyN projections from SSp-bfd. a**, Summary table of driver lines and viral vectors used for anterograde tracing from PyNs in primary somatosensory cortex, related to Fig. 3. **b**, Virus injection centroid coordinates across single driver experiments in CCFv3 space on a dorsal whole-brain view. **c-e**, Whole-brain 3D renderings of axon projections registered to CCFv3 and main projection targets for each PyN subpopulation in the SSp-bfd. **f**, Axon projection matrix from SSp-bfd to 321 ipsilateral and 321 contralateral targets (in columns), each grouped under 12 major categories (top row) for each of the driver lines generated in this study highlighted in Fig. 3, and presented alongside several previously published driver lines (IT lines: Cux2, Sepw1,

Rasgrf2, Tlx3; PT lines: Rbp4, Sim1; CT lines: Ntsr1) for comparison (see Extended Data Fig. 5 for images). Colour shades in each row represent fraction of total axon signal measured from a single experiment per brain area; signal in the inj. site (white) was subtracted from total axon signal to show the fraction of projections outside the inj. site. PyNs^Plxnd1 project bilaterally to CTX and Str; PyNs^Cux1 project bilaterally to CTX but minimally to Str; PyNs^Fezf2, PyNs^Adcyap1 and PyNs^Tcerg1l project to multiple ipsilateral targets, and contralateral brainstem (arrows); PyNs^Tbr1 project bilaterally to CTX and ipsilaterally to thalamus, PyNs^Foxp2 and PyNs^Tle4 project to the ipsilateral CTX and thalamus. Scale bar in **d** (for all 3D renderings in **c-e**), 2 mm. Asterisks in **f** indicate passing fibers. Abbreviations explained in the Supplementary Information.

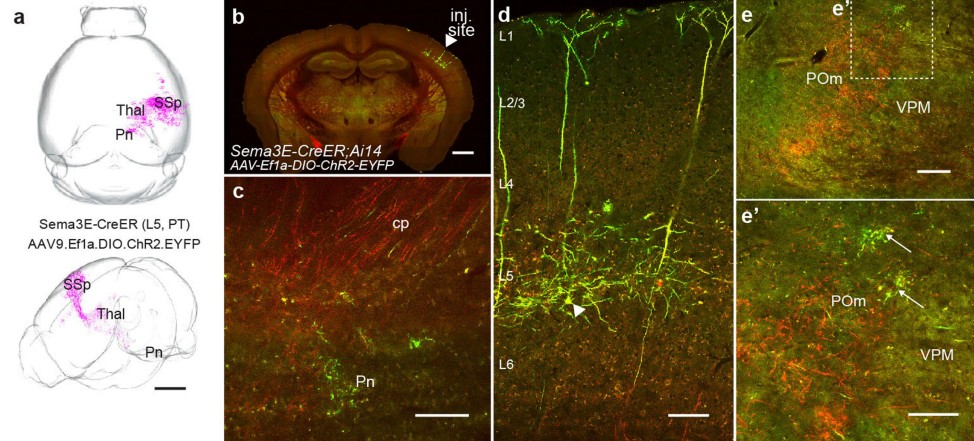

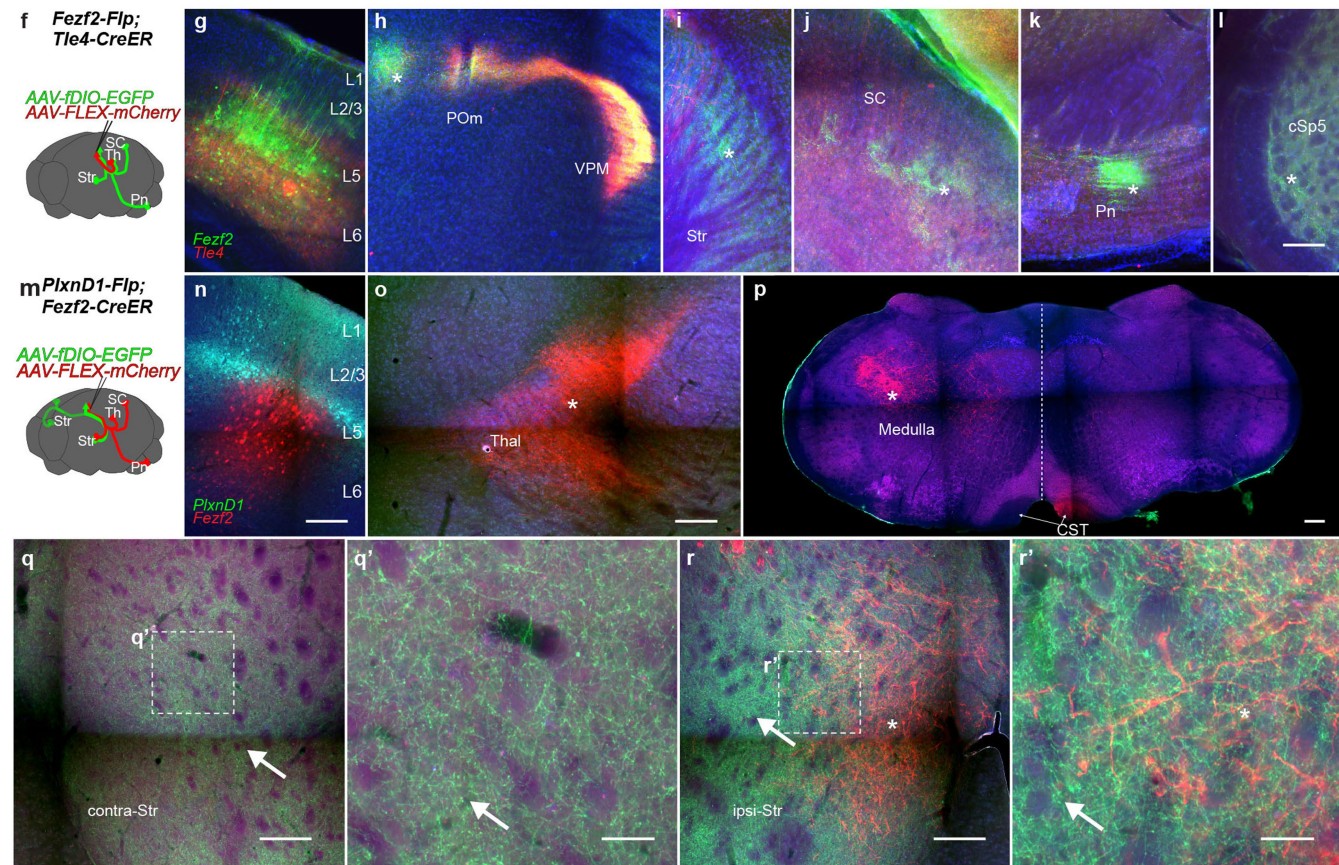

**Extended Data Fig. 8 | Anterograde tracing of axon projections from one or two PyN populations using single- or double-allele driver mice, respectively. a**, 3D rendering from STP imaging of PyNs[Sema3E] projection pattern based on injection of AAV9-Ef1a-DIO-ChR2-EYFP followed by tamoxifen induction (2 days post-injection): dorsal view (left) and parasagittal view (right). **b**, Coronal section containing injection site in a *Sema3E-CreER;Ai14* mouse. **c**, PyNs[Sema3E] project via the cerebral peduncle (cp) to the pons (Pn). PyNs[Sema3E] infected with AAV9-Ef1a-DIO-ChR2-EYFP express EYFP/tdTomato; all other PyNs[Sema3E] express tdTomato by tamoxifen induction. **d**, PyNs[Sema3E] at injection site, showing somata in layer 5 with slender tufted apical dendrites. **e**, PyNs[Sema3E] axons in thalamus with large boutons in POm (**e'**). **f-r**, Simultaneous anterograde tracing from two driver allele-defined PyN populations. **f**, Schematic showing simultaneous anterograde tracing from PyNs targeted by *Fezf2*-Flp (green) and *Tle4*-CreER (red) with co-injection of Flp- and Cre- dependent AAVs expressing EGFP and mCherry, respectively (**g-l**). **g**, PyNs[Fezf2] and PyNs[Tle4] at the injection site occupying mainly L5B and L6,

respectively. **h**, PyN[Fezf2] and PyN[Tle4] projection patterns converge in primary thalamus, VPM, whereas PyNs[Fezf2] collaterals (asterisk) extend medially to higher order thalamic nuclei. **i**, PyNs[Fezf2] (green) extend axon collaterals in Str, whereas PyNs[Tle4] (red) pass through en route to thalamus. **j-l**, PyNs[Fezf2] but not PyNs[Tle4] project to multiple other corticofugal targets, including SC, Pn and cSp5. **m**, Schematic showing simultaneous anterograde tracing from PyNs targeted by *Plxnd1-Flp* (green) and *Fezf2-CreER* (red) with co-injection of Flp- and Cre-dependent AAVs expressing EGFP and mCherry, respectively (**n-r**). **n**, PyNs[Plxnd1] and PyNs[Fezf2] at the injection site in motor cortex occupying mainly L5A and L5B/L6, respectively. **o-p**, PyNs[Fezf2] but not PyNs[Plxnd1] project to Thal (**o**) and medulla (**p**). **q-r**, PyNs[Plxnd1] and PyNs[Fezf2] project to ipsilateral Str with overlapping terminals (**r**), whereas PyNs[Plxnd1] but not PyNs[Fezf2] project to contralateral Str (**q**). Asterisks indicate PyN[Fezf2] collaterals and arrows indicate PyN[Plxnd1] collaterals. Scale bars: **a**, 2 mm; **b**, 1 mm; **c-e**, 200 μm; **e'**, 100 μm; **l** (applies to g-l), 200 μm; **n-r**, 200 μm; **q'** & **r'**, 100 μm.

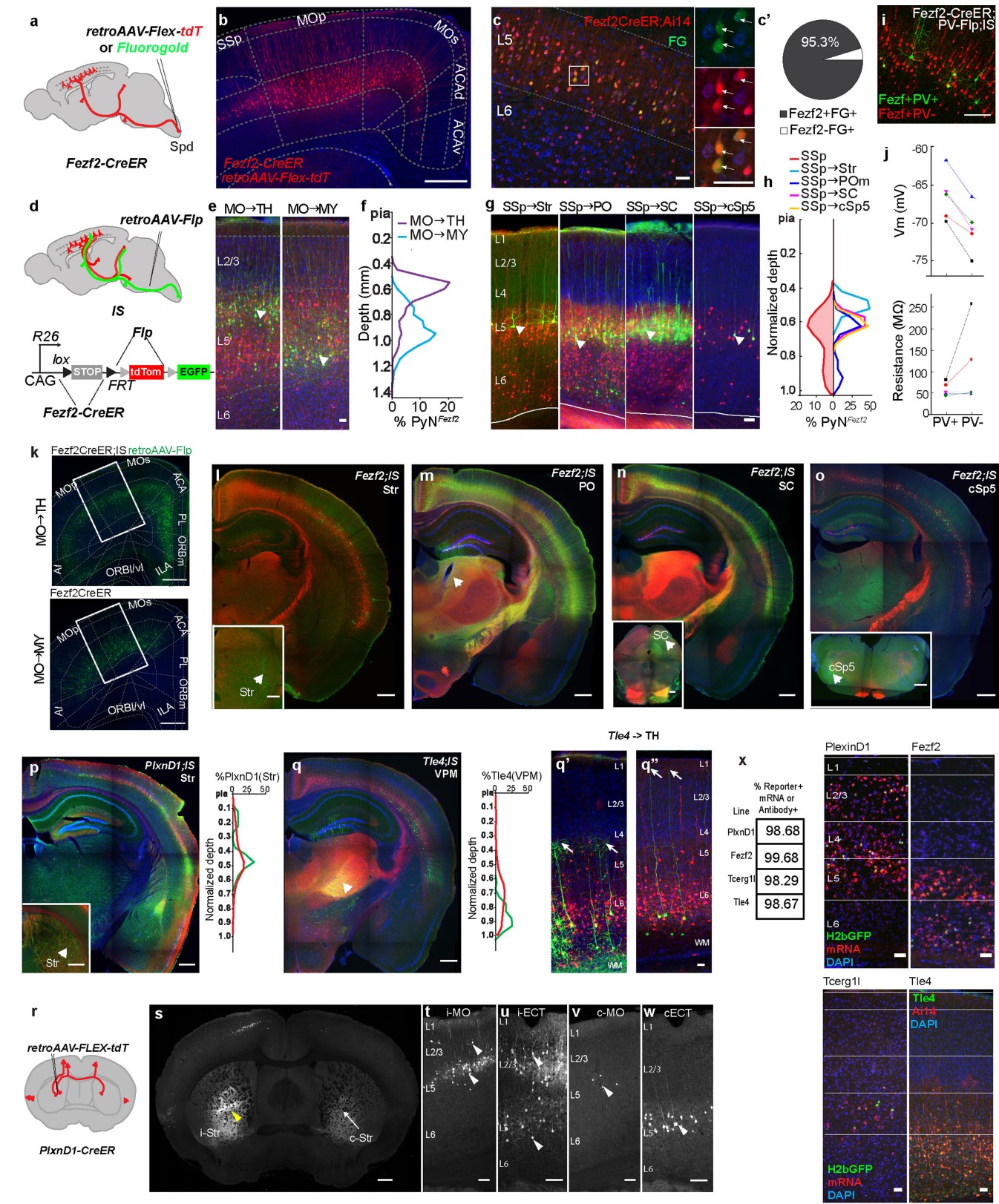

**Extended Data Fig. 9** | See next page for caption.

**Extended Data Fig. 9 | Intersectional dissection of PyN subpopulations.**
**a**, Strategy for retrograde labelling of PyN$^{Fezf2}$ subpopulations by viral or tracer injections from the spinal cord of *Fezf2-CreER* mice. **b**, PyN$^{Fezf2}$ corticospinal neurons in L5B of sensorimotor cortex labelled by retroAAV-flex-tdT. **c**, *Fezf2CreER;Ai14* captures >95% of retrogradely labelled Fluorogold+ corticospinal neurons. Inset arrows indicate *Fezf2* and Fluorogold co-labelling. **d**, An intersection-subtraction (IS) reporter strategy to label projection-defined PyN$^{Fezf2}$ subpopulations (retroAAV-Flp, EGFP) within the overall population (*Fezf2-CreER*, RFP). **e**, **f**, In motor cortex (MO), thalamus-projecting PyNs$^{Fezf2}$ are located in upper L5B, whereas medulla-projecting PyNs$^{Fezf2}$ are located in lower L5B. **g**, In SSp-bfd, PyNs$^{Fezf2}$ labelled from a defined projection target (EGFP) show more restricted sublaminar position in L5 compared to overall population (RFP). **h**, Normalized cortical depth distributions of overall PyN$^{Fezf2}$ population (leftward curve) and of each target-defined subpopulation (rightward curves) in SSp-bfd. **i**, In *Fezf2-CreER;Pv-Flp;IS* triple allele mice, PV$^-$ and PV$^+$ PyNs$^{Fezf2}$ are distinguished by their expression of RFP and EGFP, respectively, in SSp-bfd. **j**, Sample voltage responses induced by current injection from a pair of PV$^+$ (EGFP) and PV$^-$ (RFP) PyNs$^{Fezf2}$ by whole-cell patch recording in a cortical slice., Electrophysiological differences between 5 pairs of PV$^+$ and PV$^-$ PyNs$^{Fezf2}$: resting membrane potential (Vm, -66.5 ± 1.6 vs. −70.7 ± 1.5 mV, mean ± s.e.m.; p = 0.0014, Student's paired t-test); input resistance (MΩ, 60.1 ± 7.8 vs. 108.7 ± 45.4 MΩ, mean ± s.e.m.; p = 0.23, Student's paired t-test). **k**, PyNs$^{Fezf2}$ retrogradely labelled from thalamus and medulla are distributed in the upper or lower L5B, respectively, in the motor cortex (related to **e**, **f**). In a *Fezf2-CreER;IS* mouse (upper panels), retroAAV-Flp was injected in thalamus. In a *Fezf2-CreER* mouse (lower panels), retroAAV-Flex-GFP was injected into the medulla. **l-q**, Representative hemi-sections containing the SS-bfd showing the labelling patterns of PyNs$^{Fezf2}$, PyNs$^{Plxnd1}$, PyNs$^{Tle4}$ subsets by retroAAV-Flp injections at subcortical targets (arrows) in *PyN-CreER;IS* mice. In each panel, the overall *PyN-CreER* population was labelled by RFP, whereas the target-specific subset expressed EGFP. Corresponding cortical soma depth distribution is shown for Fezf2 hemisections in **g**, **h** and to the right for Plxnd1 (**p**) and Tle4 (**q**) hemisections (n = 2 for each target). PyNs$^{Tle4}$ project to VPM and consist of two subpopulations with apical dendrites in L4/5 (**q'**) and L1 (**q''**), respectively, indicated by arrows. **r**, Retrograde targeting of striatum-projecting PyNs$^{Plxnd1}$ by injection of retroAAV-FLEX-tdTomato in striatum. **s**, Coronal section displays injection site (arrowhead) and collaterals of retrogradely labelled PyNs$^{Plxnd1}$ in contralateral striatum (arrow). **t-w**, Laminar patterns of retrogradely labelled PyNs$^{Plxnd1}$ reveal that L5A PyNs$^{Plxnd1}$ project to both ipsi- and contralateral striatum, whereas L2/3 PyNs$^{Plxnd1}$ project to ipsilateral striatum (**t**, **u**, **v**, **w**). **x**, Validation of four PyN driver lines by fluorescence in situ hybridization (*Plxnd1, Fezf2, Tcerg1l*) and antibody (TLE4) using *Plxnd1-, Fezf2-, Tcerg1l-CreER* driver mice bred with a Rosa26-loxpSTOPloxp-H2bGFP reporter. H2bGFP signal colocalized with mRNA in situ signals of *Plxnd1, Fezf2* and *Tcerg1l*. In *Tle4-Cre;Ai14* mice, RFP signals colocalized with immunofluorescence of the TLE4 antibody. Scale bars: **b**, **l-q**, 500µm; **c**, **e**, **g**, **q'**, **q''**, **x**, 50µm; **i**, **s-w**, 100µm. Abbreviations explained in the Supplementary Information.

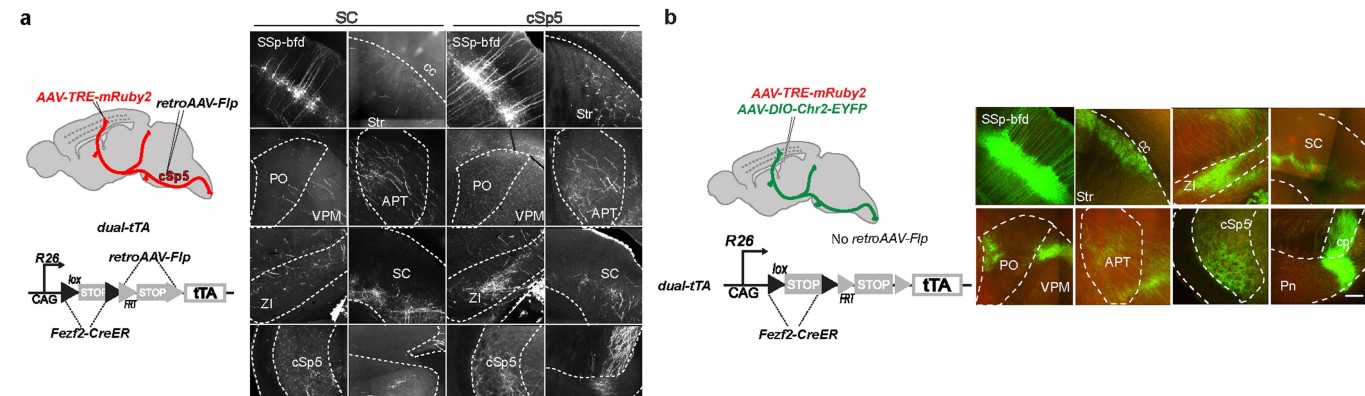

**Extended Data Fig. 10 | Strategy to target PyN subpopulations combining driver line, projection target and cortical location. a**, Left schematic shows the 'triple trigger' strategy for combinatorial targeting by marker, axon target and soma location. Upon Cre and Flp recombination, the dual-tTA reporter expresses the transcription activator, tTA. In a *Fezf2-CreER;dual-tTA* mouse, tamoxifen induction combined with retrograde retroAAV-Flp injection at the target, cSp5 or SC, activate tTA expression in cSp5- or SC-projection PyNs$^{Fezf2}$ across cortical areas, and AAV-TRE-mRuby anterograde injection at SSp-bfd then labels projection-defined PyNs$^{Fezf2}$ in the SSp-bfd. Coronal images of PyNs$^{Fezf2}$ in SSp-bfd with projection to SC or cSp5, displaying axon collaterals at various subcortical targets, including Str, ZI, thalamus, pons. **b**, Left schematic depicts the control experiment for use of Cre- and Flp-dependent *dual-tTA* reporter for target-defined axon projection mapping of PyNs$^{Fezf2}$. Co-injection of a Cre-dependent AAV-DIO-ChR2-EYFP (green, positive control) and tTA-activated AAV-pHB-TRE-mRuby2 (red, negative control), followed by tamoxifen induction, in absence of Flp confirms dependence of reporter on both Cre and Flp recombination. Example images of anterograde injection site (SSp-bfd) and axon projection targets of several indicated ipsi- and contralateral sites displaying EGFP+PyNs$^{Fezf2}$ axons from Cre-dependent AAV, but no mRuby2+ axons (in absence of Flp), demonstrating the dependence of 'triple trigger' strategy on intersection of Cre and Flp. Scale bars: a-b, 200 um.

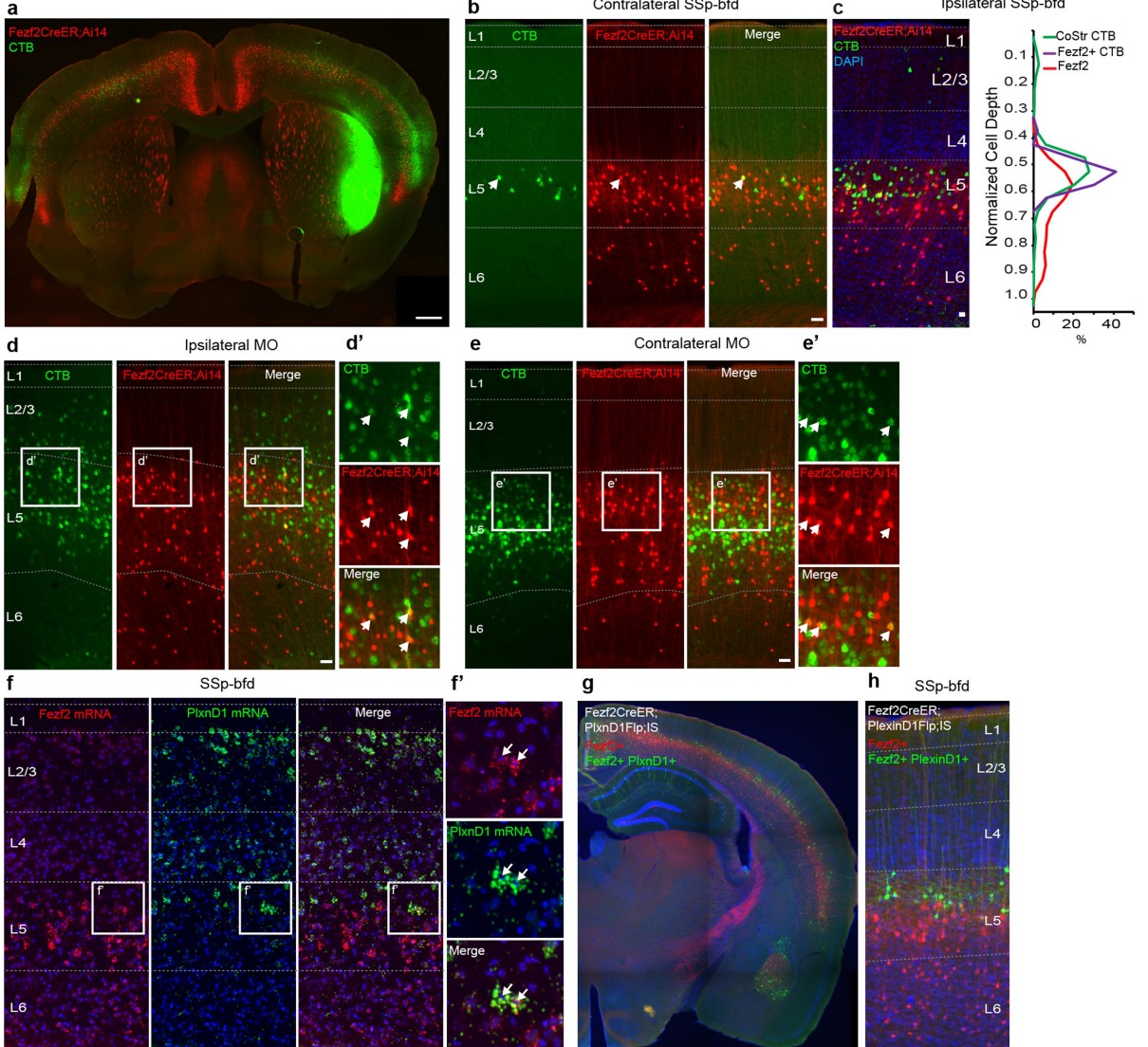

**Extended Data Fig. 11 | A subset of PyNs^Fezf2 manifest IT features.**
**a-c**, Retrograde injection of CTB (green) in the lateral striatum labelled a set of contralateral (**a**, **b**) and ipsilateral (**c**) PyNs^Fezf2 at the top of L5B in SSp-bfd. The whole population of PyNs^Fezf2 labelled in *Fezf2-CreER;Ai14*. Depth distributions in **c** compares total CTB-labelled PyNs (green), CTB+ Fezf2+ (magenta), and the whole population of PyNs^Fezf2 labelled in *Fezf2-CreER;Ai14*. **d**, **e**), Retrograde injection of CTB in the lateral striatum labelled a set of ipsilateral (**d**, **d'**) and

contralateral (**e**, **e'**) PyNs^Fezf2 at the top of L5B in MO. **f**, *Fezf2* and *Plxnd1* mRNAs are co-expressed (**f'**, arrows) in a subset of L5 PyNs in the L5B/L5A border region. **g**, **h**, Intersection/subtraction mapping in *Fezf2-CreER;Plxnd1-Flp;IS* mice revealed a small set of PyNs^Fezf2/Plxnd1 (EGFP) located at the top of L5B among other PyNs^Fezf2 (RFP). Magnified insets are from the boxed regions in each panel. DAPI (blue) counterstain in c, **f-h**. Scale bars: **a**, **g**, 500μm; **b**, **c**, **d**, **e**, **f**, **h**, 50μm.

**Extended Data Table 1 | Summary of information on the expression and molecular function of genes selected to generate new driver lines and the targeted progenitor and PyN types**

| Gene | Function | Expression | Driver | Progenitor type | Projection type | Laminar pattern |
|---|---|---|---|---|---|---|
| *Lhx2*[10,32,53-57] TF; homeodomain | Cortical primordium formation; NE proliferation and differentiation; RG fate restriction; upper layer PyN differentiation; timing of gliogenesis | E8-E10: NE; E10-E17: RGs; Perinatal: L2-5A PyNs; Postnatal: astrocytes | CreER | NE; RGs TM E10-E16 | IT | L2/3/4/5a >E14.5 TM |
| *Fezf2*[11,12,32] TF; zinc finger | Master regulator of corticofugal PyN identity | Embryonic: RGs, Postmitotic: L5/6 PyNs | CreER Flp | RGs TM E10-E13 | PT | L5b/6 Postnatal TM |
| *Tis21*[14] Transcription co-regulator | Anti-proliferative, control cell cycle checkpoint | Across neurogenesis: nRGs and IPs | CreER | nRGs, nIPs TM E10-E18 | All | All |
| *Tbr2*[14,15,58] TF; T-box | Anti-proliferative, control cell cycle checkpoint | Across neurogenesis: nRGs and IPs | CreER | nRGs, nIPs TM E10-E18 | All | All |
| *Cux1*[22,23,32,59-61] TF; homeodomain | Initiate and maintain IT identity, dendrite and synapse formation, activity-dependent connectivity | Embryonic: RGs, Postmitotic: L2-4 PyNs | CreER | RGs, IPs | IT | L2-4 Postnatal TM |
| *PlxnD1*[25,26,32] CAM | Migration, axon guidance, synapse specificity | Postmitotic: L2/3, L5A IT PyNs | CreER Flp | | IT | L2/3/5A Postnatal TM |
| *Adcyap1*[32,62] Peptide | Stimulates adenylate cyclase, Fezf2 transcriptional target | Postmitotic: L5B PT | CreER | | PT | L5b E16-17 TM |
| *Tcerg1l*[10,32,63] TF; elongation | PT-differentiation | Postmitotic: L5B PT | CreER | | PT | L5b E17 TM |
| *Sema3E*[26,27,64,65] CAM | Migration, axon guidance, synapse specificity | Postmitotic: L5B PT | CreER | | PT | L5b Postnatal TM |
| *Tbr1*[29,66] TF; T-box | Repress PT fate (Fezf2, Ctip2) | Postmitotic: L6 CT, L2-4 PyNs | CreER | | CT | L5/6 Postnatal TM |
| *Tle4*[31,32] TF; co-repressor | Regulates CT characteristics | Postmitotic: L6 CT, L5B subset | CreER | | CT | L6 Postnatal TM |
| *Foxp2*[32,33,35,67] TF; winged helix | Multiple roles in neuronal development, implicated in speech and language | Embryonic: RGs, Postmitotic: L6 CT | Cre | RGs | CT | L6 Systemic AAV |

NE: neuroepithelium; RG: radial glia; IP: intermediate progenitors; nRG: neurogenic RG; TF: transcription factor; CAM: cell adhesion molecule

This table includes citations of refs. [53–67].

# nature research

# Reporting Summary

Nature Research wishes to improve the reproducibility of the work that we publish. This form provides structure for consistency and transparency in reporting. For further information on Nature Research policies, see our Editorial Policies and the Editorial Policy Checklist.

## Statistics

For all statistical analyses, confirm that the following items are present in the figure legend, table legend, main text, or Methods section.

| n/a | Confirmed | |
|---|---|---|
| ☐ | ☒ | The exact sample size (*n*) for each experimental group/condition, given as a discrete number and unit of measurement |
| ☐ | ☒ | A statement on whether measurements were taken from distinct samples or whether the same sample was measured repeatedly |
| ☐ | ☒ | The statistical test(s) used AND whether they are one- or two-sided<br>*Only common tests should be described solely by name; describe more complex techniques in the Methods section.* |
| ☒ | ☐ | A description of all covariates tested |
| ☒ | ☐ | A description of any assumptions or corrections, such as tests of normality and adjustment for multiple comparisons |
| ☐ | ☒ | A full description of the statistical parameters including central tendency (e.g. means) or other basic estimates (e.g. regression coefficient) AND variation (e.g. standard deviation) or associated estimates of uncertainty (e.g. confidence intervals) |
| ☐ | ☒ | For null hypothesis testing, the test statistic (e.g. *F*, *t*, *r*) with confidence intervals, effect sizes, degrees of freedom and *P* value noted<br>*Give P values as exact values whenever suitable.* |
| ☒ | ☐ | For Bayesian analysis, information on the choice of priors and Markov chain Monte Carlo settings |
| ☒ | ☐ | For hierarchical and complex designs, identification of the appropriate level for tests and full reporting of outcomes |
| ☒ | ☐ | Estimates of effect sizes (e.g. Cohen's *d*, Pearson's *r*), indicating how they were calculated |

*Our web collection on statistics for biologists contains articles on many of the points above.*

## Software and code

Policy information about availability of computer code

| | |
|---|---|
| Data collection | 1. Serial two photon tomography (STP) datasets were collected using the commercial set-up by TissueVision (Cambridge, MA), followed by Fiji-based stitching, Elastix-based registration, segmentation, quantification and data presentation, as described in Ragan et al 2012 and Mandelbaum et al 2019, doi: 10.1016/j.neuron.2019.02.035, and publicly distributed in Kim et al., 2017, doi: 10.1016/j.cell.2017.09.020).<br>2. For non-STP data, imagesets were acquired (every other section) across the whole brain using a Zeiss Axioimager M2 System equipped with MBF Neurolucida Software (MBF) and x5 objective. Representative fields of view were selected and then confocal image stacks were acquired on a Zeiss LSM 780 or 710 microscope (CSHL St. Giles Advanced Microscopy Center) using objectives x20, x40 and x63.<br>3. Molecular characterization of driver lines with embryonic induction and fixation at P7 was performed on a Nikon Eclipse 90i fluorescence microscope with a x20 objective.<br>4. Electrophysiology data was collected using the commercially available pCLAMP 10.3 software (Molecular devices). |
| Data analysis | 1. Elastix was used within the established pipeline to register brains with an average reference brain, either the Allen Institute's CCFv3 and its associated grayscale brain structure annotations (as in Harris et al 2019) or the Osten reference brain published in Ragan et al 2012 for areal distributions and cortical flatmapping as in Kim et al 2015.<br>2. Code to plot cell distribution in cortical flatmaps is publicly distributed in Kim et al 2017, doi: 10.1016/j.cell.2017.09.020).<br>3. Custom Matlab (Matlab_R2018a) code for both cortical depth distributions and anterograde tracing projection matrix is available upon request.<br>4. Other software includes: MS Excel, GraphPad Prism 7, Fiji/ImageJ Version 2.0.0-rc-68/1.52g; Imaris software (Bitplane) version 9.3 and 9.5; Adobe Photoshop CS6. |

For manuscripts utilizing custom algorithms or software that are central to the research but not yet described in published literature, software must be made available to editors and reviewers. We strongly encourage code deposition in a community repository (e.g. GitHub). See the Nature Research guidelines for submitting code & software for further information.

## Data

Policy information about availability of data

All manuscripts must include a data availability statement. This statement should provide the following information, where applicable:

- Accession codes, unique identifiers, or web links for publicly available datasets
- A list of figures that have associated raw data
- A description of any restrictions on data availability

Raw and stitched whole-brain STP imaging data is available from the BICCN Brain Image Library (BIL) (http://www.brainimagelibrary.org/download.html) at the Pittsburgh Supercomputing Center, based on the accession codes detailed in Supplementary Tables 1 & 2. Anterograde projection datasets can be visualized on the Mouse Brain architecture website (http://brainarchitecture.org/cell-type/projection) as detailed in Supplementary Tables 1 & 2. All accession codes will be included within Supplementary Tables 1 & 2 prior to publication. All other datasets generated during this study are available from the corresponding author upon request.

# Field-specific reporting

Please select the one below that is the best fit for your research. If you are not sure, read the appropriate sections before making your selection.

☒ Life sciences ☐ Behavioural & social sciences ☐ Ecological, evolutionary & environmental sciences

For a reference copy of the document with all sections, see nature.com/documents/nr-reporting-summary-flat.pdf

# Life sciences study design

All studies must disclose on these points even when the disclosure is negative.

| | |
|---|---|
| Sample size | Sample sizes were estimated on the basis of previous studies using similar methods and analyses (Oh et al 2014, doi: 10.1038/nature13186.; Kim et al 2015; Harris et al 2019, doi: 10.1038/s41586-019-1716-z). A full list of data acquired is provided in Tables S1 and S2. Based on consistency of results between litters and between individuals within a given litter, a sample size of 2 litters was considered sufficient for all embryonic characterization. |
| Data exclusions | All analyzed data was included in the study. Prior to STP analysis, datasets were screened for standard quality control by two independent reviewers according to pre-established criteria. |
| Replication | -The labeling patterns achieved from the strategies described here reveal a high degree of replicability across animals. A full list of data acquired for cell distribution and virus injections is provided in Supplementary Tables 4 and 5, respectively. These tables detail experiments based on knock in Cre/Flp driver lines bred with appropriate reporters for cell distribution analysis and injected with virus for axon projection mapping.<br>-Oh et al 2014 and Harris et al 2019 demonstrated a high degree of replicability across animals for anterograde virus tracing. Based on this rationale, they confidently and comprehensively sampled with n=1 experiment per source area and driver line.<br>-Here we utilized a minimum n=2-3 for each cell distribution and virus tracing experiment in adult animals, and spanning a minimum of 2 litters for short-pulse embryonic experiments. For Sema3E and Tcerg1l, due to the high replicability mentioned above, we analyzed a single successful injection for each driver line. |
| Randomization | Not relevant because there was no group allocation. |
| Blinding | Not relevant to this study because there was no group allocation. |

# Reporting for specific materials, systems and methods

We require information from authors about some types of materials, experimental systems and methods used in many studies. Here, indicate whether each material, system or method listed is relevant to your study. If you are not sure if a list item applies to your research, read the appropriate section before selecting a response.

## Materials & experimental systems

| n/a | Involved in the study |
|---|---|
| ☐ | ☒ Antibodies |
| ☒ | ☐ Eukaryotic cell lines |
| ☒ | ☐ Palaeontology and archaeology |
| ☐ | ☒ Animals and other organisms |
| ☒ | ☐ Human research participants |
| ☒ | ☐ Clinical data |
| ☒ | ☐ Dual use research of concern |

## Methods

| n/a | Involved in the study |
|---|---|
| ☒ | ☐ ChIP-seq |
| ☒ | ☐ Flow cytometry |
| ☒ | ☐ MRI-based neuroimaging |

# Antibodies

| Antibodies used | Anti-GFP (1:1000, Aves, GFP-1020); anti-RFP (1:1000, Rockland Pharmaceuticals, 600-401-379); anti-mCherry (1:500, OriGene AB0081-500); anti-mKate2 for Brainbow 3.0 (gift of Dr. Dawen Cai, U Michigan); anti-SATB2 (1:20, Abcam ab51502); anti-CTIP2 (1:100, Abcam 18465); anti-CUX1 (1:100, SantaCruz 13024); anti-LDB2 (1:200, Proteintech 118731-AP); anti-Fog2 (1:500, SantaCruz m-247), anti-LHX2 (1:250, Millipore-Sigma ABE1402) and anti-Tle4 (1:300, Santa Cruz sc-365406). |
|---|---|
| Validation | All antibodies are commonly used in the field and have been validated in previous publications/by the manufacturer, as detailed here:<br>-anti-GFP (Aves, GFP-1020): validated by manufacturer by immunohistochemistry (1:500) using transgenic mice expressing GFP<br>-anti-RFP (Rockland Pharmaceuticals, 600-401-379): validated by the manufacturer by immunoelectrophoresis resulting in a single precipitin arc against anti-Rabbit Serum and purified and partially purified Red Fluorescent Protein (Discosoma). No reaction was observed against Human, Mouse or Rat serum proteins. https://rockland-inc.com/store/Antibodies-to-GFP-and-Antibodies-to-RFP-600-401-379-O4L_24299.aspx<br>-anti-mCherry (OriGene AB0081-500): https://www.origene.com/catalog/antibodies/primary-antibodies/ab0081-500/mcherry-goat-polyclonal-antibody<br>-anti-mKate2 for Brainbow 3.0 (gift of Dr. Dawen Cai, U Michigan): validated in Cai et al 2013 (doi: 10.1038/nmeth.2450)<br>-anti-SATB2 (Abcam ab51502): https://www.abcam.com/satb2-antibody-satba4b10-c-terminal-ab51502.html<br>-anti-CTIP2 (Abcam 18465): https://www.abcam.com/ctip2-antibody-25b6-chip-grade-ab18465.html<br>-anti-CUX1 (SantaCruz 13024): https://www.scbt.com/scbt/product/cdp-antibody-m-222<br>-anti-LDB2 (Proteintech 118731-AP): https://www.ptglab.com/Products/LDB2-Antibody-11873-1-AP.htm#datasheet<br>-anti-FOG2 (SantaCruz m-247): widely used in the field (e.g. Alfano et al 2014, doi: 10.1038/ncomms6632)<br>-anti-LHX2 (Millipore-Sigma ABE1402): https://www.emdmillipore.com/US/en/product/Anti-LHX2-Antibody,MM_NF-ABE1402<br>-anti-Tle4 (SantaCruz sc-365406): https://www.scbt.com/p/tle4-antibody-e-10 |

# Animals and other organisms

Policy information about studies involving animals; ARRIVE guidelines recommended for reporting animal research

| Laboratory animals | -Species/strain: Mus musculus, C57Bl/6J or Swiss Webster<br>Cre/Flp driver knock in and reporter lines used: Lhx2-2A-CreER (JAX stock # 036293), PlexinD1-2A-CreER (JAX stock # 036294), PlexinD1-2A-flpO (JAX stock # 036295), Fezf2-2A-CreER (JAX stock # 036296), Fezf2-2A-flpO (JAX stock # 036297), Tcerg1l-2A-CreER (JAX stock # 034000), Adcyap1-2A-CreER (JAX stock # 033999), Sema3E-CreER (Y. Yoshida; Pecho-Vrieseling et al., 2009, doi: 10.1038/nature08000); Tle4-2A-CreER (JAX stock # ), FoxP2-IRES-Cre (R. Palmiter, Rousso et al., 2016, doi: 10.1016/j.celrep.2016.04.069), Tbr1-2A-CreER (JAX stock # 036299), Cux1-2A-CreER (JAX stock # 036300), Tbr2-2A-CreER (JAX stock # 036301), Tbr2-2A-FlpER, Tis21-2A-CreER (JAX stock # 036303), dual-tTA (JAX stock # 036304).<br>Other Cre/Flp driver knockin and reporter lines used: PV-2A-FlpO (JAX stock # 022730), Ai14 (JAX stock # 007908), Ai65 (JAX stock # 021875), LSL-h2b-GFP (He et al 2016), LSL-Flp (JAX stock # 028584), IS (JAX stock # 028582), Snap25-LSL-EGFP (JAX stock # 021879), RGBbow (JAX stock # 028583), Cux2-Cre (Franco et al., Science 2012, RRID:MMRRC_031778-MU), Cux2-CreERT2 (Franco et al., Science 2012, RRID:MMRRC_032779-MU), Ntsr1-Cre_GN220 (Gerfen et al., Neuron 2013), Rasgrf2-T2A-dgFlpO (JAX stock# 029589), Rbp4-Cre_KL100 (JAX stock # 031125), Sepw1-Cre_NP39 (JAX stock # 037622), Sim1-Cre_KJ18 (JAX stock # 031742), Tlx3-Cre_PL56 (JAX stock # 036547)<br>New driver and reporter lines have been deposited to the Jackson Laboratory for wide distribution.<br>-Age: embryonic E11.5, E12.5, E13, E13.5, E14.5, E17; postnatal stages P5, P7, 1-6 months<br>-Sex: males and females<br>All details appear in Supplementary Tables 1, 3, 4, 5, 7 |
|---|---|
| Wild animals | The study did not involve wild animals. |
| Field-collected samples | The study did not involve field-collected samples. |
| Ethics oversight | All experimental procedures were carried out in accordance with NIH guidelines and approved by the Institutional Animal Care and Use Committees of Cold Spring Harbor Laboratory and Harvard University. |

Note that full information on the approval of the study protocol must also be provided in the manuscript.

