## [Peer Review File · Nature]

Manuscript Title: Genetic dissection of the glutamatergic neuron system in cerebral cortex

Editorial Notes:

Redactions – unpublished data

Reviewer Comments & Author Rebuttals

Reviewer Reports on the Initial Version:

Referee #1 (Remarks to the Author):

Summary of the key results

This paper introduces 16 mouse lines for genetically labeling defined subsets of cortical pyramidal neurons using a combination of recombinases (11x CreER, 1xCre, 2xFlpO, 1x FlpER, and a new Dual tTA reporter, Table 1). These lines are designed as knockin mice to specific transcription factor sites, which, along with the temporal control offered by tamoxifen induction of the CreER/FlpER alleles, are intended to offer superior control of the targeting of transgene expression to genetically defined cell types. In particular, the tools are designed to study different subtypes of excitatory pyramidal neurons in neocortex for developmental or circuit studies. Retrograde viral injections (to target cells based on axonal projection patterns) are added to refine subtypes based on projection pattern.

Genetic access to defined pyramidal cell types is of interest to neuroscientists dissecting cortical circuits. The field begins to understand of the range of cortical cell types based on expression patterns (Zeisel et al., 2015, Tasic et al., 2018, and Hodge et al, 2019, etc.) or projection patterns (Economo et al., 2018). Though some lines exist, these are predominantly inhibitory interneuron lines (such as Taniguchi et al., 2011 from the senior author's group). There are some useful lines that do address this issue (from the GENSAT project, Gerfen et al., 2013, among others – Harris et al., 2014). These are not sufficient for all cell types in all cortical areas, and thus the questions these tools are intended to address are highly relevant.

Originality and significance: if not novel, please include reference

The work is original and significant. I would like to see the authors make a more direct comparison of the cell types to which their methods permit access and the existing tools to specify (a) which type/types can we reach that we could not reach before, (b) how do these types correspond to what we think the existing genetically defined or projectionally defined cell types are. My excitement for these tools would be improved if such a comparison were presented in an easy-to-understand way. This is further detailed below.

Data & methodology: validity of approach, quality of data, quality of presentation

These are generally strong. There are some minor comments on the presentation of the paper's figures (in minor comments). The main data that would ideally be shared is the image libraries of different mouse lines (which of course are not possible to put in the paper). In the movies, these are visible at low resolution. I found the Pittsburgh Supercomputing Site difficult to access/navigate/obtain the images, perhaps due to lack of familiarity. The supplemental tables to provide some ID numbers that might be helpful in getting access to the image stacks. This is

further detailed below.

Appropriate use of statistics and treatment of uncertainties

Appropriate. There is very little use of statistics. Where the images do not seem to support the point being made, I put a comment but without some quantification of the images, this is a qualitative and not quantitative assessment.

Conclusions: robustness, validity, reliability

See comments.

Suggested improvements: experiments, data for possible revision

There are two main suggestions for improvement:

(a)

-Discussion/Text to address which lines/strategy are comparable or better than existing tools. I did not find this done persuasively. An example of what I mean is this: Suppose I am interested in studying the range of subtypes of "PT-type" L5B projection neurons (Shepherd 2013 Nat Rev Neurosci). The literature has led me to believe that there are likely 2-3 subsets of PT-type neurons (Economo et al 2018), those that project to thalamus (possibly two types, Winnubst et al., 2019) and those that project to medulla. If I want access to these, I can:

Use a Cre-driver like KJ18/Sim1_Cre (Gerfen et al., 2013) which gets some of all of these for many cortical areas, including S1. To limit this to just S1, I must inject some Cre-dependent AAV into S1.

Or

Use a Cre-driver like KJ18/Sim1_Cre (Gerfen et al., 2013) which gets some of all of these for many cortical areas, including S1. Then to further limit this to just medulla or thalamus projecting subset, I use a retrograde injection of some Flp-dependent construct. To then limit this to just S1, I must inject some Cre and Flp dependent AAV into S1 (Fenno et al., 2014), with a few options for Con/Fon, Con/Foff, etc.

Or

For targets in superior colliculus, brainstem, spinal cord, a single retrograde injection in these areas may suffice (or a retrograde injection of Cre in this target and a cortical injection of Cre-dependent virus.)

Your approach may be superior to this in some ways, but I need to be persuaded how and why (because after reading the paper I am still thinking this way). The summary in Fig 4e suggested that the *Fezf2*, *Adcyap1*, and *Tcerg1l*, for example, might include a mix of cell types that project to both thalamus and medulla, and so also require additional manipulation to subdivide.

(b)

-General housekeeping (see Clarity and context). I think the figures would benefit from covering different projectionally defined classes together (e.g. all the figures pertaining to tools for PT-type neurons, then all the figures pertaining to tools for CT-type, then IT-type; currently parts of these are all in Fig 3 and 4, while 5 and 6 are more accessible to me since they go into detail). Clearing up the references to supplemental material will help too.

References: appropriate credit to previous work?

This is good. I try to include in my comments the papers that influenced my thinking (which largely overlap with the citations), but don't mean to force changes to citations. The references are definitely appropriate.

Clarity and context: lucidity of abstract/summary, appropriateness of abstract, introduction and conclusions

The clarity of the paper is (to me) the main problem. The scope of the work is obviously complex, encompassing a range of new mouse tools for addressing several different classes of cell types. Following the text is thus difficult, since the figure panels are discussed somewhat out of order

(text moves between figures and then back and forth). The supplemental material was difficult to assess since it was presented in many formats: excel files, pdf table, and movies. The numbering of the movies does not seem to always correspond to the text (once this is figured out, they are nice) and several are not even referenced in the main text or the supplement. As such, I think the paper could be improved by editing (even in the absence of more data).

Major Comments:

How do lines compare to the cell types we know?

The main obstacle to my understanding the advance here is trying to make the connection between existing resources and approaches and how the new ones move beyond that. Specifically, I want to know:

How do the cell types able to be targeted in the present paper compare with the genetically (such as Zeisel et al 2015, Tasic et al 2018, Hodge et al, 2019) or projectionally defined pyramidal cell types (Chen, Helmchen 2015 Nat Neurosci, Economo et al, 2018, etc.) in other studies? This is discussed transiently: Line 354-355: "Together, this new set of driver lines facilitates experimental access to a set of genetically-defined PT PyNs." This is a great finding and seems likely to be the case. Line 488 "The relationship between current and previously reported driver lines targeting similar PyN classes remains to be further characterized at the cellular resolution." Achieving an understanding of this is necessary. It is not as useful to the field to present the new tool without characterizing how it is comparable to the previously reported one. The statement (Line 88: "Among the challenges of establishing genetic access to PyN types, the first and foremost is specificity (with an appropriate granularity) and the second is comprehensiveness.") gave me the hope that there would be a range of lines for different IT-type neurons (something similar to M1 or S2 projecting L2/3 neurons in S1, or V1 neurons projecting to LM or PM or other higher order visual areas). I was concerned that the results here don't quite reach that level of granularity. For example, it might be possible to indicate whether the new lines cover the same projection classes as existing lines and/or any additional specificity achieved? (e.g. a Table similar to Fig 3a or Table 2 which lists all the PT-type together, categorizing as ... 'PT-type, thalamus projecting' or 'PT-type, thalamus and medulla projecting', and something similar for IT-type and CT-type?).

How effective do you assess the transcription factor strategy to be?

I would like to believe that the hierarchical transcription factor strategy would minimize this, but some of the images suggest that genetically defined cells in a single line might contribute to multiple projectionally-defined populations. I would have hypothesized that a genetically homogenous population would project to a well-defined set of targets. Do you feel that these mouse lines support or reject that? Do some lines have multiple populations mixed together (PT-type and CT-type or PT-type and IT-type in the same line)? If we must reject this hypothesis, it suggests that maybe retrograde injections will be necessary for the foreseeable future to target projectionally-defined types, unless the enhancer-based techniques give an alternative effective method (Blankvoort et al; Allen Institute). A potential problem for this approach different cell types might be generated at different times with some probability (potentially overlapping developmental periods when multiple cell types are being generated) as well as if cell types are specified not only by whether a given transcription factor is on/off, but if these genes could be expressed at different levels (a low enough level to mediate recombinase activity but not drive high levels of all the genetic targets needed for differentiation into a given cell type). Are there some labeled neurons using these approaches that don't fit the expected pattern? To what degree is this a problem?

Over what cortical areas (m/l or a/p or granular/agranular) do these lines work best?

In characterizing these lines, it would be helpful to understand how broadly (different cortical areas from medial to lateral, granular/agranular) these new lines work? Medial-to-lateral, Extended Data Fig 3 is the closest which is very nice for the lines that are presented. But this isn't available for all lines. But it seems like the images exist for all the lines. In other pyramidal cell lines (GENSAT, for example), the number of neurons labeled varies across the tangential surface

of cortex. This opens the possibility that either neurons of potentially the same type are missed (for example, pyramidal tract type cells can be ID'd by retrograde approaches in many cortical areas, such as medial wall areas, but seemingly not well by the GENSAT transgenics or the current methods). Thus, there is something about this that suggests that neither approach does a comprehensive job of targeting projectionally defined cell types. Isn't it a concern for the logic of the approach (hierarchical use of transcription factors) that some brain areas don't obey these rules (e.g. the transcriptional control of PT-type neuron generation in medial sites may differ from lateral sites)?

What about the fraction of neurons labeled in each of these approaches?

Is there some idea of the fraction of total PT-type cells each line might label? Is the change in % labeled consistent with the hierarchical level? (e.g. successively fewer % labeled for later-acting TFs?) Fig 3c suggests that *Fezf2* labels far more neurons in L5B than *Adcyap1* and *Tcerg11*, but what fraction of the overall PT population is this targeting?

Supplemental Info and Tables need some detailed proofreading/examination.

The text reads "Adeno-associated viruses (AAVs) serotype 8, 9, DJ PHP.eB or retro2 packaged by commercial vector core facilities (UNC Vector Core, ETH Zurich, Biohippo, Penn, Addgene, ...) were used as listed in Supplementary Table 2." where the ellipsis might be omitted. The table I think is one of the Excel files, which lists all injections for all figures, but contains many blanks and isn't titled. This is in contrast to other Tables that are titled or are in a different format (S4 versus S1, say). Furthermore, some movies are referred to incorrectly and not all movies are mentioned in the text (Movies 14, 15, 16 never mentioned). It was frustrating to try to be thorough and review the data as the text directs me to, and find it difficult to connect which movie went with which statement.

Here are some examples of where this arose: Movies 6 and 7 are referenced (lines 339-341) but the actual movies are likely to be 7 and 8. The movie files don't contain text to differentiate in the movie itself, so I went to the titles for the uploads (see below). But even then, the same Movie 7 is referred to in the text as pertaining to *Adcyap1* (Line 341) and *Tcerg11* (Line 345). The imaging is beautiful and deserves to be examined, but the way the files are presented and referenced makes it frustrating and puts the onus on the reader to figure out what's going on. This could be improved by potentially including a caption in the video itself to disambiguate. I have not yet figured out where the higher resolution images on which the movies are based are stored, but these are the sort of thing I would want to peruse at leisure if there were a site/viewer with which to do it. (<http://www.brainimagelibrary.org/download.html> or <http://www.brainimagelibrary.org/datasets.html>) from the methods seems to be the site at which to do it, but this is not as user friendly as other sites (I don't see how to browse images) and requires me to figure out some identifying number for the specific brain I want to see. Possibly some of this information is in the supplemental tables. To test this, I tried using Image ID from Table 2 to search, but didn't find it and the website for most of the images in that table are blank. Brains 180804 and 180820 are listed in both the PSC website and the Table, but others are not. I imagine others would want to browse these images too!

Line specific comments:

Cux1-CreER seems to label layers 2-4. Issues include: (1) a difference in the connectivity of L2 and L3, and (2) different subtypes with L3 depending on the projection target (Chen, Helmchen et al *Nat Neurosci* 2015; Sato, Svoboda et al *J Neurosci* 2010, etc.). The line is also not uniform across layers (as few of these lines are), but it suggests that there are some neurons in these layers that aren't accessible under these strategies. This line requires work with an additional strategy (retrograde label or other) in order to achieve some specificity beyond layers 2/3/4. How this might be superior to some of the layer 2/3 Cre lines that already exist? (The *Sepw1-Cre* line (Gerfen et al., 2013), the *Cux2-CreERT2* (Franco et al., 2012) and *Rasgrf2-2A-dCre* (Harris et al., 2014)).

PlexinD1 also not uniform across layers (assumes the images shown are the *PlexinD1-CreER*

instead of the Flp line but the legend could be more explicit). Is the expression pattern the same in PlexinD1-CreER and PlexinD1-Flp? The example images in 3b (Cre) versus 6e (Flp) look slightly different - the Flp version is more specific for L5 or L5A and also the Flp version is brighter in S1 (some lateral area) than other areas, while perhaps this difference is not as pronounced in the CreER. Is there a figure like the Extended Data Fig 3 "flat map" to compare? PlexinD1 in your images (Fig 3b) has seemingly two clear peaks ~L2 and L5, but the histogram across S1 looks flat in Ext Data Fig 3, why the difference? (Also, I don't think you need both histograms with different binning if you edit that figure.)

The treatment of the *Adcyap1* expression needs improvement. In Fig 3c, *Adcyap1* expression looks relatively specifically restricted to L5B. It is sparse and not uniform from medial to lateral. (If Video 7 is typical, that brain had much more widespread expression from medial to lateral). But then the text directs me to the Supplementary Info Fig 3 where the histogram seems to suggest expression across many layers (Present in 2/3 in the histogram, a dip but not absent in 4, more in 5 ... maybe the histogram is just mislabeled). But then example images are back to sparse in L5 targeted in the Supplementary Info Fig 4.

I didn't understand the comparison of *Tcerg11* to *Fezf2* (Lines 345-347): "It is currently unclear how these *PyNsTcerg11* relate to *PyNsFezf2*. Anterograde AAV labeling of *PyNsTcerg11* revealed overlap with *PyNsFezf2* but absence of the collateral to first-order thalamic nucleus VPM". I didn't see (or expect per prior work) much collateralization of any PT-type projection in VPM from L5B cells, and this seems consistent with neither of these lines showing much arborization there (Fig 3c, the VPM and POm row). Compare Fig 3c and Fig 4b. They seem to tell the opposite story for *Fezf2* (Fig 3c has POm >> VPM; Fig 4b has VPM >> POm nearly invisible). Why? I hypothesize (but can't tell) that the sparse *Fezf2* population visible in L6 might contribute corticothalamic output to VPM (from the S1 viral injection). It is possible that the relative absence of VPM in the S1 injection (4b) is because these axons visible in 3c originate from other cortical areas.

Line 217-218: "There was no apparent morphological difference between *Fezf2+* and *Fezf2-nRGs*." What are you intending to convey here? Prior to this, you're arguing for heterogeneity in this population. But this suggests you don't see it (in whatever feature was surveyed for morphology). Seems contradictory. I imagine many people will think that some of the heterogeneity hypothesized might come in the projection targets, and this was not easy to assess (in that figure). So either say what the morphological feature measured is (or leave out)?

It is interesting in the CT-projection population that *Tle4* but not *Tbr1* has a more pronounced intracortical axonal projection to L5A (guess of lamination from images in 3d middle row), suggesting these neuron subsets might not be overlapping.

PT-type retrograde label:

I found it interesting that there was no laminar structure to the different L5B cells based on their projection targets (Fig 5d). The graph suggests similar laminar pattern for the three groups (except striatal), though the text argues that *cSp5* differs. Is this mostly due to inclusion of some very deep layer cells in layer 6? Are there L5A (presumed IT-type) included in the striatal injection? The sample image seems sparse, though I would have anticipated the opposite since so many PT-axons to other subcortical targets have collaterals in striatum (MouseLight has many examples, but I think this is also described as far back as Cajal ... and Charles Wilson? See Kita and Kita *J Neurosci* 2012; more recently the MouseLight papers *Economo et al* 2018, *Morita et al* 2019 *Front Neural Circuits*, *Winnubst et al* *Cell* 2019). Is this retrograde approach susceptible to any problem of tropism (e.g. AAV2retro infecting all PT-types equally)? This finding seems to slightly contrast to the *Economo et al* (2018) paper (upper vs lower neurons projecting to thalamus and medulla, respectively). This was also shown in rat by *Rojas-Piloni* (2017 *Nat Communications*), with different laminar distributions for ~4 targets. The data here are especially different since the one group that differs is the retrograde experiment from striatum. If all/most PT-type neurons have a striatal collateral, and this would be the broadest group (e.g. that the other

subsets might all be included here).

Complex genetic strategy:

Line 467: "using just three alleles" For many groups this approach will be expensive and complex. The number of mice/transgenes required to pursue some of these approaches (potentially three transgenes plus viral injections) will be a challenge for some smaller labs to adopt due to cost/space. Thus, it is likely that adoption will not be uniform for the field. So, can the authors argue for the cases where a more well-defined subset of cell types is targeted that can't be achieved other ways, so as to justify the cost?

Minor Points (Including typos and presentation):

Intro Line 56: Some recent evidence for (rare) GABAergic projection neurons include:

<https://www.ncbi.nlm.nih.gov/pubmed/25164650>

<https://pubmed.ncbi.nlm.nih.gov/27159237/>

In many cases, an acronym is used ('IS mice', line 213) prior to begin defined in the text (line 403). Other places include Line 95: TF-defined RGs? Transcription factor? It wasn't defined yet. Line 154: Define NEs before using acronym.

Items listed as Tables (Table 1, Table 2) are presented to the reviewers in the Supplemental Information pdf and not with the manuscript (makes it harder to find when reviewing). It's not clear if the idea is to present in the paper or in the supplement once formatted. This info (Tables 1+2) is quite useful but does take up space (the list of lines is partially duplicative with the table in Fig 3).

Line 141: "In a similar yet contrasting way", probably not ideal phrase.

Fig 1e: Cartoon looks like a more defined trajectory than the inset in 1c (where IT's seemingly continue to be produced at later times).

The order of the panels for presentation/discussion could be improved (e.g. present figures and panels in the order they are discussed in the text? Fig 2q-s discussed after 2t-u).

Fig 2: For panels 2e-u, where are the scale bars? These don't seem to be in the legend. For these same panels, are the authors sure the inset is the same as the boxed area in the left images (2e, 2g 2i, etc.)? This is not obvious and in the case of 2e, as the intense fluorescence of the low magnification image in the top and bottom layers is not as apparent in the magnified image. Why is the RGBow approach used for 2f but not h and j? This is in some panels and not others (also in the Fezf2 part of the figure). The blue channel is visible in some but not others.

The use of the IS mouse has some minor inconsistency. I thought this mouse is supposed to express either red or green. "Express tdTomato for "Cre NOT Flp" subtraction and GFP for "Cre AND Flp" intersection" (Miao et al 2016). But the double expressing cells (Fig. 2) are not explored in great detail other than the explanation that "The mixed GFP-RFP clones are likely derived from Fezf2- nRGs in which Tis21-CreER first activated RFP expression followed by postmitotic activation of GFP through Fezf2-Flp (Fig. 2u, Extended Data Fig. 2d, d', g)." Maybe this is a reasonable explanation but is not directly tested. Since the cell was born embryonically, but the dual expression persists at P30, doesn't this suggest the possibility that some cells in the IS reporter line are capable of stable expression of both red and green?

Line 333: "anteropretectal nucleus" I think means anterior pretectal nucleus (References I looked at did not call it that.)

Line 346: Calls the gene Tcerg11 instead of Tcerg 1l in one place.

Electrophysiology: Why 2/2 mM Ca and Mg?

Fig 4c: Scale bar typo fraction not 'faction'.

Fig 5c: What are the inset cartoons in 5c used for? I see a few areas identified but then no corresponding images (or discussion in the figure legend). Maybe this is where the retrograde AAV was injected (and 5b shows only one case)? Why do some images have DAPI, others do not? Why is the level of RFP expression high for most cases, but so low in the cSpV case? Is this region in the same cortical area as the others?

Fig 6jk: The cStr and iStr labels not ideally placed (white on white).

References: Ref #11 missing year

Referee #2 (Remarks to the Author):

Matho et al. generate and describe new mouse models to dissect cortical circuits, by enabling to target specific subpopulations of pyramidal neurons. The models rely on the combinatorial use of inducible CreER or Flp, knocked in the endogenous locus of about 10 genes displaying specific patterns of expression in the developing or adult cortex.

They first describe two drivers (FexFf2 and Lhx2) labeling radial glial cells that may constitute two distinct but partially overlapping pools, that seemingly generate most pyramidal neurons, and drivers labeling committed progenitors (Tis21 and Tbr2 genes), enabling to target distinct, mostly layer-specific, subtypes using timed induction.

These are complemented by about 10 more specific lines that enable collectively (typically using intersectional strategies) to label in a specific way a specific repertoire of pyramidal neuron subsets with distinct patterns of projections (intratelencephalic, subcerebral, thalamic) and/or layer identity. They provide some examples of how these can be used to identify anatomically some subtypes, characterized by specific axonal or dendritic patterns that would otherwise be indistinguishable by existing genetic or anatomical labeling techniques.

Overall this study provides a comprehensive resource that may be useful for a broad range of applications, from development to functional analyses of defined populations of neurons. It goes beyond the state of the art by the degree of specificity achieved, that results from the genetic targeting strategy as well as combinatorial aspects relying on regional and temporal patterns of expression.

One weakness is that there are no major novel insights on cell diversity of the cortex, as the intriguing observations that suggest unexpected cell diversity among some of the labeled progenitor or neuron subpopulations, are not investigated in depth. This study should thus be considered mostly as the description of a resource to the community for future studies, more than a direct approach to tackle cortical cell diversity in a systematic way.

In this context one limitation is that the targeted neuronal types are not linked in any way to the emerging cortical neuronal identities recently identified by single cell transcriptomic analyses. This would greatly increase the impact and applicability of the models described. It could be done by profiling the labeled cells in at least a few of the described lines, followed by direct comparison with the existing transcriptome datasets.

The authors show data suggesting that fezf2 and Lhx2- CreER strains label different populations of progenitors, based on cell distribution and outcome. The authors should provide more evidence, in particular quantification (more cells, different litters,..) to determine if the qualitative differences that they observe are indeed robust and reproducible and not the mere result of variability in

labeling efficiency. Transcriptome profiling of each cell population would be another complementary way.

Similarly, the authors describe distinct profiles of Tbr2 / Tis21 labeled cells that indicate that “most Tbr2+ progenitors are neurogenic instead of transit-amplifying”. It is not clear how the data presented lead unequivocally to this conclusion. Clearly more should be done to support this conclusion, in particular by comparing Tbr2/Tis21 and Tbr2 cell labelling at shorter time points, and using clonal (low TAM induction) analyses following the induction.

The authors state that all the lines described allow expression that is faithful to the endogenous gene, but only reporter gene expression are shown. This is particularly important since the pattern of reporter genes also reflects lineage and not just expression. Importantly, they should also demonstrate that the expression pattern of the endogenous gene is unaltered by the targeting. The authors should provide more data on these important aspects, for instance using in situ hybridization of Cre / Flpe compared with endogenous gene expression, in the various knock-in and control lines.

The authors provide information on layer distribution of the reporters in various cortical areas. They should also provide a similar information for axonal projection patterns. This would be indeed extremely useful information as these are likely to differ depending on the area considered.

The authors should describe in much more detail the mode of administration of Tamoxifen (timing, method, concentration etc) that is given to the mice for timely inductions to achieve each given profile of cell specificity. This is not trivial given the known variability of this method to achieve reliable and safe induction of gene expression. Similarly, the mouse strain background should be specified and any known strain difference in specificity or efficiency of the induction of the reporters should be explicated.

The various crossings and reporters used should be schematized more systematically in each figure : It would make the study much easier to follow.

Referee #3 (Remarks to the Author):

Advances in sequencing technologies have greatly improved our understanding of the genetic identity of neuronal subtypes in the mammalian brain. However, in part due to a lack of genetic access, much less is known about the functional and anatomical diversity of pyramidal neurons and their underlying developmental lineage. In this manuscript Matho et al. describe novel genetic knockin lines that make use of developmentally-timed induction to target specific neuronal progenitor pools. By combining these lines with intersectional reporters they were able to target sets of neurons defined by birth time, marker expression, anatomical location and projection targets. The resources and techniques described in this manuscript are innovative and are combined in highly creative ways. This will likely open up new possibilities for researchers studying cell-type specific neuronal functioning, connectivity, and developmental neurogenesis. However, the impact of this resource is undercut by a lack of detailed characterization of the presented lines. Furthermore, the potential of combining this approach with retrograde tracing to characterize PyNs subtypes based on projection targets is only marginally explored. As a result, there is a lack of novel biological insights that would make it valuable to a larger audience.

Major Points

1. How specific, and non-overlapping, are the populations labeled with the described approach? This is not made explicit in several parts of the paper. For example the Lhx2 and Fezf2-CreER

driver lines are described as giving 'partially overlapping as well as possibly distinct RG pools.' (190-191). Are Lhx2 and Fezf2 chosen because they are non-overlapping during development? Could the authors perform stainings for the opposite TF with each driver line to investigate this? Later on it is mentioned that 'their progeny might differ in terms of projection pattern and connectivity when analyzed at cellular and clonal resolution' (192-193). Unfortunately, this is not further investigated later in the paper which prevents any conclusive statements on 'whether different RG subpopulations ..contribute, ultimately, to projection-defined PyN subpopulations' (line 81-83). In general, it is often ambiguous if parts of this section serve as validation or aim to present novel biological findings on corticogenesis as the authors often state that results capture known endogenous expression patterns.

2. It is unclear how the described driver lines for IT, PT, and CT neurons fit with our current understanding of known projection subtypes. A previous study has shown that in the motor cortex the PT neurons can be roughly divided into two anatomical and genetic subtypes: medulla and thalamic projecting PT neurons (Economo et al. 2018). It would be very informative to know what proportion of cells of the Fezf2-CreER line are labeled by injections into the thalamus and/or medulla. Single cell reconstruction data has suggested that the actual diversity of PT cells is likely to be even larger (Winnubst et al. 2019). Can the authors provide any evidence that the Adcyap1, Tcerg1l, Sema3a lines capture different parts of this diversity?

3. Similarly, L6 CT projecting neurons in the barrel cortex are often separated based on their projections to primary or non-primary sensory nuclei. According to this scheme, primary projecting cells are located in upper L6, have dendrites terminating in L4, and project mostly to the VPM and reticular nucleus; non-primary cells are located deeper in L6, have dendrites terminating in L5, and project mostly to POM as well as VPM but not the reticular nucleus (Zhang and Deschênes 1997; Thomson 2010). The authors describe that "Anterograde tracing reveals that PyNsTle4 in S1 project specifically to first order thalamic nucleus VPM". Because of the previously described literature it's quite surprising that these neurons are located throughout L6 and have dendrites in L4 and 5. Even more surprising is that in Extended Data Figure 5f it seems that cells in deeper L6 are labeled by the retrograde injection in VPM. Can the authors comment on this? The authors should be better able to identify these subtypes based on a retrograde injection in the reticular nucleus. Do the Tbr1, Tle4, Foxp2 label both types equally or is there a bias? Finally, in line 373 and 413-415, why is it specifically the retrograde label that confirms the two populations of cells with dendrites in L4 and L1? Both cell types are labeled and it's unclear what the retrograde injection adds here. I found this quite confusing because I initially assumed that this meant only L1 neurons projected to VPM.

4. An extension of Extended Data Figure 4 to include more of the reported cell lines would be very helpful for getting an insight into the identity of the targeted cell populations. A similar analysis as in (c), but with a calculation of the fraction of antibody labeled cells that are tdTom+ would also be informative.

5. In the Tis21-CreER;Fezf2-Flp long-duration experiments, isn't it also possible that Gfp-only cells were Fezf2 negative at one point but the RFP protein has since degraded? Are there therefore three distinct groups or could there be a gradient? Also, why does it say on line 222 that RFP-only cells likely consisted of PT cells? I assume it can only be said that these cells probably originated from nRGs that never expressed Fezf2.

6. Can the authors say anything about how precise the tamoxifen induction window likely is during the embryonic stage? Are the short-duration pulse-chase experiments (sacrifice after 24 hours) directly relatable to the long-term expression observed in the adult or can the induction of expression take place over a longer timescale?

Minor Points

1. The figure panels are on many occasions cited out of order with some of the skipped panels being cited till much later in the text. I would suggest the authors consider reordering the figures to be more in line with the text.
2. Please note that several figures lack scale bars (for example: Fig. 2, Fig 5f, most of Fig 6.)
3. The barrel cortex is sometimes referred to as SSp other times as SSp-bfd, according to the Allen atlas these abbreviations refer to more the general and specific primary sensory areas and should be used consistently.
4. Line 179-180. What characteristics are referred to here? the medial-lateral gradient? Because this was already mentioned in the previous sentence.
5. Line 372. Reference to Extended Data Figure 5h-n does not seem relevant.
6. Line 455. Contralateral striatum mentioned twice.
7. Line 74 in extended data. '...'
8. Figure 1
 - 1a. Order of blue, purple red cells doesn't seem to make sense compared to c.
 - Why does only Tbr2 have an associated embryonic age (>15)?
9. Figure 2
 - In many cases the zoomed-in panels don't seem to match the marked boxes (e.g. 2e. does not appear to have dense labeling near the ventricle wall).
 - 2b. The dating with tamoxifen induction is not mentioned in this schematic.
 - 2f,h,j Density of the reporters look very different (RGBow vs Ai14) making them hard to compare. Would it be to add a quantification of the layer distribution?
 - 2i,o. Can't really see the PyNs clones generated from individual RGs. Separate inset would be useful (similar to Extended Data Figure 1) with an explanation in the legend why these must come from single RGs. Also why is the asterisk yellow?
10. Figure 3c. According to text Adcyap PyNs should be in upper L5B but in picture it looks like they are mostly in L5a.
11. Figure 4
 - 4c. Not very helpful since lines are so thin and many areas are zero for all. Better to group by larger anatomical areas. Text states 'faction' should be 'fraction'. Finally, the panel is referred to as 4b in the legend.
 - 4e. Are the cartoons for Adcyap and Tcerg1l switched? Shouldn't Tcerg1l have no VPM projections?
12. Figure 6. Usage of colors for induction days is slightly misleading since they are not linked to the expressed fluorophore.
13. Table 2. Unclear what the meaning of +?, +-, and what the difference is between - and no entry?
14. Extended Data Figure 1f. No quantification, going by eye seems to be roughly half/half glia cells and PyNs and more in layer 5? Doesn't seem to match text (170-171).
15. Extended Data Figure 2a. Mislabeled insets.
16. Extended Data Figure 4.

- It's not clear to me why some apparently double labeled cells do not have arrows or tdTomato cells without labeling have no asterisks, are these supposed to be examples only? Why are there two different kinds of arrows?
- I would add somewhere that *ctip2* and *cux1* are used as markers for L5-6 and L2-4

17. Extended Data Figure 6. Would be useful to see the quantification of the red cells as well for comparison

References

Economo, Michael N., Sarada Viswanathan, Bosiljka Tasic, Erhan Bas, Johan Winnubst, Vilas Menon, Lucas T. Graybuck, et al. 2018. "Distinct Descending Motor Cortex Pathways and Their Roles in Movement." *Nature* 563 (7729): 79–84.

Thomson, Alex M. 2010. "Neocortical Layer 6, a Review." *Frontiers in Neuroanatomy* 4 (March): 13.

Winnubst, Johan, Erhan Bas, Tiago A. Ferreira, Zhuhao Wu, Michael N. Economo, Patrick Edson, Ben J. Arthur, et al. 2019. "Reconstruction of 1,000 Projection Neurons Reveals New Cell Types and Organization of Long-Range Connectivity in the Mouse Brain." *Cell* 179 (1): 268–81.e13.

Zhang, Z. W., and M. Deschênes. 1997. "Intracortical Axonal Projections of Lamina VI Cells of the Primary Somatosensory Cortex in the Rat: A Single-Cell Labeling Study." *The Journal of Neuroscience: The Official Journal of the Society for Neuroscience* 17 (16): 6365–79.

Author Rebuttals to Initial Comments:

Response to Referees

Referees' comments:

Referee #1:

Summary of the key results

This paper introduces 16 mouse lines for genetically labeling defined subsets of cortical pyramidal neurons using a combination of recombinases (11x CreER, 1xCre, 2xFlpO, 1x FlpER, and a new Dual tTA reporter, Table 1). These lines are designed as knockin mice to specific transcription factor sites, which, along with the temporal control offered by tamoxifen induction of the CreER/FlpER alleles, are

intended to offer superior control of the targeting of transgene expression to genetically defined cell types. In particular, the tools are designed to study different subtypes of excitatory pyramidal neurons in neocortex for developmental or circuit studies. Retrograde viral injections (to target cells based on axonal projection patterns) are added to refine subtypes based on projection pattern.

Genetic access to defined pyramidal cell types is of interest to neuroscientists dissecting cortical circuits. The field begins to understand of the range of cortical cell types based on expression patterns (Zeisel et al., 2015, Tasic et al., 2018, and Hodge et al, 2019, etc.) or projection patterns (Economo et al., 2018). Though some lines exist, these are predominantly inhibitory interneuron lines (such as Taniguchi et al., 2011 from the senior author's group). There are some useful lines that do address this issue (from the GENSAT project, Gerfen et al., 2013, among others – Harris et al., 2014). These are not sufficient for all cell types in all cortical areas, and thus the questions these tools are intended to address are highly relevant.

- We thank Rev1 for his/her appreciation of the significance of our work.

Originality and significance: if not novel, please include reference

The work is original and significant. I would like to see the authors make a more direct comparison of the cell types to which their methods permit access and the existing tools to specify (a) which type/types can we reach that we could not reach before, (b) how do these types correspond to what we think the existing genetically defined or projectionally defined cell types are. My excitement for these tools would be improved if such a comparison were presented in an easy-to-understand way. This is further detailed below.

- We address these important points thoroughly under “two main suggestions for improvement – (a)” below.

Data & methodology: validity of approach, quality of data, quality of presentation

These are generally strong. There are some minor comments on the presentation of the paper's figures (in minor comments). The main data that would ideally be shared is the image libraries of different mouse lines (which of course are not possible to put in the paper). In the movies, these are visible at low resolution. I found the Pittsburgh Supercomputing Site difficult to access/navigate/obtain the images, perhaps due to lack of familiarity. The supplemental tables to provide some ID numbers that might be helpful in getting access to the image stacks. This is further detailed below.

- We have improved the trackability of datasets from IDs in Supplemental Tables (e.g. suppl Table 5) and on the Brain Architecture website, a portal for viewing the data at high resolution with the option of showing Allen CCF3 labels.

- We have made a major effort on data availability. All movie files can be downloaded and whole brain STPT datasets registered to the CCFv3 are readily accessible on the Brain Architecture Portal by Partha Mitra's lab (this portal is currently undergoing some restructuring). We are including:

> 38 brains on Cell Projection (<http://brainarchitecture.org/cell-type/projection>) for this paper. 18 of them were posted and have been available. 20 brains are yet to be posted by March 22.

> 50 brains on Spatial Distribution for this paper. These will be made available by March 22.

Appropriate use of statistics and treatment of uncertainties

Appropriate. There is very little use of statistics. Where the images do not seem to support the point being made, I put a comment but without some quantification of the images, this is a qualitative and not quantitative assessment.

- We have added extensive quantification and performed statistics as appropriate, for Figure 2, 5e-f, Extended Data Fig. 1, 2, 5, 8a,p.

Conclusions: robustness, validity, reliability

See comments.

Suggested improvements: experiments, data for possible revision

There are two main suggestions for improvement:

(A)

-Discussion/Text to address which lines/strategy are comparable or better than existing tools. I did not find this done persuasively. An example of what I mean is this: Suppose I am interested in studying the range of subtypes of “PT-type” L5B projection neurons (Shepherd 2013 Nat Rev Neurosci). The literature has led me to believe that there are likely 2-3 subsets of PT-type neurons (Economo et al 2018), those that project to thalamus (possibly two types, Winnubst et al., 2019) and those that project to medulla. If I want access to these, I can:

Use a Cre-driver like KJ18/Sim1_Cre (Gerfen et al., 2013) which gets some of all of these for many cortical areas, including S1. To limit this to just S1, I must inject some Cre-dependent AAV into S1.

Or

Use a Cre-driver like KJ18/Sim1_Cre (Gerfen et al., 2013) which gets some of all of these for many cortical areas, including S1. Then to further limit this to just medulla or thalamus projecting subset, I use a retrograde injection of some Flp-dependent construct. To then limit this to just S1, I must inject some Cre and Flp dependent AAV into S1 (Fenno et al., 2014), with a few options for Con/Fon, Con/Foff, etc.

Or

For targets in superior colliculus, brainstem, spinal cord, a single retrograde injection in these areas may suffice (or a retrograde injection of Cre in this target and a cortical injection of Cre-dependent virus.)

Your approach may be superior to this in some ways, but I need to be persuaded how and why (because after reading the paper I am still thinking this way). The summary in Fig 4e suggested that the *Fezf2*, *Adcyap1*, and *Tcerg1l*, for example, might include a mix of cell types that project to both thalamus and medulla, and so also require additional manipulation to subdivide.

- Up to here in this opening section, Rev1 raised three overarching questions/critiques. In order to address these effectively, I summarize them in the following order, as this will provide a more logical flow to present our rebuttal that will be easier to follow:

a) “Discussion/Text to address which lines/strategy are comparable or better than existing tools. I did not find this done persuasively”. ... “Your approach may be superior to this in some ways, but I need to be persuaded how and why”.

b) which type/types can we reach that we could not reach before,

c) how do these types correspond to what we think the existing genetically defined or projectionally defined cell types are.

- I will take significant length to address these three important points as they are also raised by Rev2&3, and they impact many of the more specific comments that we address later point-by-point.

a) “... address which lines/strategy are comparable or better than existing tools.”

We are very grateful to Rev1 for urging us to explain more clearly why and how our strategy and resource advance the field, over the existing tools, to experimentally access biologically meaningful pyramidal neuron (PyN) subpopulations toward understanding their organization principles. We now present specific side-by-side comparisons with existing tools in the substantially revised Figure 4, Ext data Fig.4, and a new sppl Table 2. We elaborate these comparisons throughout the Result text as well as their significance in Discussion. Beyond these comparisons in an easy to read format, the severe word limit of this paper does not permit us to further elaborate several conceptual and strategic points, which are perhaps better suited for a review paper later. Here we would like to take the opportunity in this

rebuttal to present these points to the reviewers. Although somewhat lengthy, we believe this will provide an overarching starting point to address many of the subsequent more specific comments.

A major principle of cell type organization in the cortex and across brain regions is their hierarchical taxonomy, consisting of top-level major classes and subclasses that branch into progressively finer subpopulations and “atomic types”. An “ideal” genetic toolkit would have three important features. The first is *specificity* at several levels of the hierarchy or granularity, from broader subclasses to finer types. The second is *completeness* – to capture most if not all cells in each subclass or type across cortical areas so that investigators can use these tools not only to study certain specific areas but further to gain insight into the global network across areas with cell type resolution. The third is *comprehensive coverage* of many PyN subpopulations to enable systematic studies of cortical circuit organization. In this context, the genetic toolkit and strategy we report are considerably superior than existing resources in terms of approach, design, specificity, completeness, and coverage; indeed they represent the establishment of a novel and powerful *experimental paradigm* for dissecting PyN organization.

Currently there are only a handful of driver lines for PyN subpopulations and even fewer for progenitors, and most of these are constitutive transgenic Cre lines. A fundamental distinction between the transgenic and gene targeting/knocking (KI) approach is the nature of regulatory mechanisms that drive tool-gene expression. This contributes to numerous stark contrasts in the resulting tools:

- 1) Transgene expression results from ectopic interactions of the transgenic promoter/enhancers with the surrounding gene regulatory elements at a random genomic integration site, giving rise to an arbitrary spatial and temporal “composite expression pattern” that often comprises an *artificial mixture* of cells, especially when examined at the cell type resolution.
- 2) Desirable transgenic expression patterns need to be identified by screening many transgenic lines, and the “hit rate” is quite low – only a dozen or so Tg lines are now more commonly used by investigators from over 250 lines screened/reported (Gerfen et al., 2013 Neuron).
- 3) A transgene expression pattern often has no relationship to the selected transgene itself. For example, while the *Sim1-KJ18* Cre line labels a set of PT neurons, endogenous *Sim1* is not even expressed in PT neurons; the same is true for the *Tlx3-PL56* line (subset of IT cells), *Ntsr1-GN220* line (subset of CT cells). Therefore, Tg expression patterns often result from “genetic coincidence” and do not inform the molecular identity and mechanism of the labeled cell populations.
- 4) transgene expression often only captures a partial, i.e. *incomplete* set of an endogenous cell population, likely due to unpredictable interactions between the transgene and surrounding genomic regulatory elements. For example, while *Sim1-KJ18* labels a set of PT neurons in numerous frontal cortical areas, it labels much less well in more posterior areas (e.g. Somatosensory areas, auditory, visual cortex). Similarly, while *Ntsr1-GN220* labels CT cells in somatosensory, it does not label the more medial (e.g. cingulate, PL) and more lateral MOp and MOs and SSp (Suppl Table 1; https://connectivity.brain-map.org/static/referencedata/experiment/thumbnails/472986374?TRANSGENE=true&image_type=TW O_PHOTON). Such arbitrary partial patterns preclude the use of these lines in many non-expressing areas and in gaining a global view of the targeted PyN subpopulation across areas.
- 5) Although useful once characterized, each transgenic pattern is unique (e.g. *Sim1_KJ18* line can be very different from *Sim_KJ21* line, depending on genomic integration site) and cannot be precisely reproduced. This property precludes the re-targeting of the same population with Flp or other tool-genes for designing intersection/subtraction strategies to capture more restricted subpopulations.
- 6) Transgenic lines can be *unstable*, likely due to epigenetic silencing. For example, the *Sim1-KJ18* shows variable patterns, in some cases with no expression in sensory areas (https://connectivity.brain-map.org/static/referencedata/experiment/thumbnails/491609390?TRANSGENE=true&image_type=TW

O PHOTON). And many transgenic lines lose their initially characterized and reported expression patterns.

In summary, the design of current PyN driver lines has been largely haphazard, and these are grossly inadequate for a systematic dissection of PyN cell types and progenitors. More importantly, the technical limitations of the Tg approach preclude a rational and coherent strategy towards achieving specificity, completeness, and comprehensiveness in systematically targeting the hierarchical organization of PyN types.

In contrast, our gene targeting based strategy and toolkit overcome most of the above shortcomings.

1) The vast majority of KI lines (over 95% based on at least 60 lines generated in our lab and others in the literature) precisely recapitulate the endogenous spatiotemporal and cellular gene expression patterns that reflect endogenous developmental processes, physiological mechanisms, and tissue organization. Therefore, judicious selection of genes for constructing knockin lines has the potential to capture and reveal inherent biological elements and processes.

2) We designed KI driver lines based on key transcription factors (TF) and effector genes implicated in the specification, differentiation, and maintenance of PyN cell fate and phenotypes. Thus, our strategy is based on the inherent hierarchical organization PyN types and their developmental genetic trajectories. These driver lines enable investigators to parse biologically relevant PyN subpopulations through their intrinsic developmental, anatomical and physiological properties, i.e. “carving nature at its joints”. We begin at the top level of the hierarchy with multiple broad and categorically distinct populations, then progress to more restricted subpopulations, and further design multiple intersectional strategies to capture increasingly finer subsets and cell types.

3) The precision and reliability of the KI approach allows us to achieve high specificity for targeting marker-defined subpopulations and completeness for capturing most if not all cells in each subpopulation across cortical areas.

For example, *Fezf2* is a master TF that specifies subcerebral/extratelencephalic projection PyNs (ET) and can *reprogram* IT neurons into PT fate (Rouaux and Arlotta, 2013). We now demonstrate that our *Fezf2* driver line captures the vast majority of PT neurons (over 95%; Figure 5). This is the first driver line that captures the most complete set of PT cells, setting the stage for studying this important population and for dissecting its numerous subtypes. Similarly, although the *PlxnD1* IT neurons are distributed across L5A, L3, L2, their expression of this key cell adhesion molecule likely defines a specific wiring feature distinct from other IT neurons. Our *PlxnD1* drivers specifically and completely capture this subpopulation. Same is true for *Tle4* and *Tbr1* drivers targeting CT neurons and other drivers for subpopulations. Importantly, the cortex-wide expression of these driver lines with characteristic areal patterns reflects the areal organization of these cell types and have been used by the Allen Institute as major parameters to delineate the mouse brain reference atlas and the common coordinate framework (CCF). These KI lines allow functional imaging and manipulation of target PyN subpopulations across widespread cortical areas (Huang lab unpublished results; e.g. wide-field GCaMp7 imaging, optogenetic activation/inhibition scanning) that are not feasible in many Tg lines).

Together, the specificity and completeness of these drivers establishes a solid foundation for studying multiple categorically distinct components of cortical network scaffold across areas and functional modalities.

4) As each KI driver is a single line at a precisely defined genomic site, the reproducibility of this approach allows us to design intersection/subtraction strategies for dissecting finer subpopulations. For example, although *Fezf2* expression appears broader than the conventional PT definition (L5b and corticofugal projection) and includes additional deep L6 neurons, *Fezf2* expression likely confers certain fundamentally common features of this population that are yet to be dissected and understood. Similarly, *PlxnD1* IT neurons are distributed across L5A, L3, L2 and yet likely share common wiring features.

These laminar patterns suggest the presence of multiple subpopulations within each population. By generating orthogonal Flp and CreER lines with nearly identical expression patterns, we can strategically design intersection and subtraction schemes to dissect PlexD1 and Fezf2 subsets based on other molecular markers, projection pattern (retro-AAV), lineage origin and cell birth order (progenitor cell division, Fig 5, 6, Ext data Fig 7, 8, 9). These strategies achieve by far the highest specificity and finest granularity in targeting PyN subtypes (e.g. Fig 6).

5) The *inducibility* of our driver lines confer temporal control over developmentally dynamic gene expression to achieve desired specificity and further provide spatial flexibility in the density or sparseness of cell type labeling and manipulation. Most endogenous and trans-genes that show restricted expression to PyN subpopulations are often more broadly expressed at earlier developmental stages, thus many constitutive Cre lines are often not suited to take advantage of the highly valuable tool-gene reporter lines because Cre recombination patterns integrate across time and become non-specific to the intended PyN subpopulation in mature cortex. [e.g. Fezf2-Flp]. The inducible control of our driver lines confers major advantages of *specificity and flexibility* in cell targeting and functional manipulation. Furthermore, temporal control allows gene manipulations (e.g. using conditional alleles) in specific subpopulations at different developmental stage to discover the cellular and molecular mechanisms of circuit development and function. Importantly, inducibility also allows investigators to control the density of labeling and manipulation of PyN subpopulations, from dense coverage to *single cell analysis*. Single cell analysis provides the ultimate resolution to examine the variability of individual neurons within driver and anatomy defined PyN subpopulations, and can be better achieved by inducible driver lines. Indeed, our Fezf2-, PlxnD1-, Tle4- CreER drivers constitute the key tools for the recent large scale single cell reconstruction of PyNs (Peng et al., 2020 bioRxiv; biccn companion paper).

6) Existing genetic tools for glutamatergic (GLU) neural progenitors are especially poorly developed. Except the Cux2-CreER line, there are almost no other KI inducible drivers for GLU progenitors. Our inducible Cre and Flp drivers provide the first set of precision tools for targeting of transcription factor-defined progenitor subpopulations. These will enable *fate mapping the developmental trajectories of PyN types*, facilitate studying many aspects of PyN development (migration, axon guidance, synapse formation), and may provide fundamental insight into the developmental basis of the hierarchical organization of PyN types.

In summary, our study represents a decisive progress over the existing driver lines for targeting PyNs. The systematic and strategic design of our driver lines and their intersections provide much increased specificity, completeness, and scope of coverage over existing lines. More importantly, the judicious choice of genes based on neuroscience knowledge for constructing these driver lines provides a road map and a set of key guideposts for dissecting the hierarchical organization of PyN types based on the molecular and development logic. In this sense, our toolkit and strategy are not just an improvement on the existing driver line collection, they represent the establishment of a *novel and powerful experimental paradigm* for exploring and understanding PyN types.

We have added sentences in Introduction, Results, and Discussion to elaborate on these important points in a succinct way.

b) “which type/types can we reach that we could not reach before”

We thank Rev1 for this important suggestion. We now summarize the comparison between existing and our new driver lines in accessing PyN types, cortical areas, single cells, and progenitors in Figs, 3, 4, Suppl Table 2, Ext Figure 4. These are presented in a systematic and easy to understand format, and explained in the Result text. We highlight a few examples below.

The IT class is the most diverse yet with the least developed tools. IT cells reside across layers 2-6 and project to combinations of ipsi- (associational) and contra-lateral (callosal) cortical areas, and striatum. It is unknown how many projection types there are and whether they correspond to distinct molecular types. Among existing lines, Cux2 is quite broad to L2/3/4, Rasgrf1 is weak (de-stabilized Cre), and Sepw1-Cre_{NP39} is not well characterized. Our new driver lines provide access to some of the most specific IT subpopulations in terms of laminar position or projection targets.

In particular, during the revision we found that Cux1-2A-CreER selectivity targets cortex-projecting but not striatum-projecting ITs in layer 2-4. This is the first KI driver showing such important specificity, indicating that Cux1 has distinct (in addition to possible overlapping) expression patterns and likely function compared to Cux2. Notably, Cux1 continues to evolve in primates and is located in human accelerated region (HAR) of the human genome. Thus, this driver line not only is a highly valuable tool but also will facilitate studying molecular and cellular basis of cortical function and evolution.

In addition, Tbr2-CreER provides the first set of tools to target L2 and L3 IT cells through embryonic birth dating. Tbr2-2A-CreER/PlxnD1-Flp intersection further defines molecular subtypes within these laminar subsets. There are no existing driver lines targeting these IT subpopulations. The Cux2-Cre and Cux2-CreER lines target L2/3 IT cells that include mixtures of striatal-, callosal-, and associational-projecting IT cells. The Rasgrf2-dCre line expresses a destabilized Cre, which is weak and not widely used (Carl Petersen, personal communication).

In terms of L5 IT cells, the existing Tlx3 Tg line is a valuable tool, but it targets an unknown molecular subset and unknown fraction of L5 IT cells, and its expression in medial cortical areas is weak. Our PlxnD1 drivers target a molecularly-defined subpopulation, and intersection with Tbr2-CreER at E13.5 further captures a specific subtype.

L5B PT is a highly diverse class with an unknown number of projection types that vary substantially across cortical areas. The existing Sim1 Tg line is a useful tool for some cortical areas, but it targets an unknown molecular subset and unknown fraction of PT cells, and its expression is weak and variable in posterior areas (https://connectivity.brain-map.org/static/referencedata/experiment/thumbnails/491609390?TRANSGENE=true&image_type=TW O_PHOTON). Sim1-Cre is not well expressed in posterior cortical areas. Our Fezf2-CreER and Flp lines capture the most complete set of PT cells (>95% Figure 5a-d) across neocortical areas, and provide a solid starting point for intersectional dissection of subtypes. In addition, Adcyap1-, Tcerg11-, Sema3E- CreER target molecular defined subsets for which their projection pattern can be analyzed both as subpopulations and at the single cell resolution.

Cell types within the L6 CT class are not well understood. The existing Ntsr1 Tg line is a useful tool, but is not expressed in medial cortex (cingulate) and portions of motor areas (anterior/lateral MOp and MOs). Our Tbr1-2A-CreER likely captures the most complete set of CT cells across neocortical areas. In addition, Tle4 and Foxp2 target transcription factor-defined subsets. In particular, Tle4+ cells consist of at least two types, one with apical dendrites ending in L4, the other in L1. Foxp2 CT PyNs project to a broader set of thalamic nuclei. These lines thus provide new tools to dissect more specific CT types.

Single cell analysis is essential to achieve the ultimate resolution for analyzing cell diversity. It is important to point out that none of the existing constitutive Tg lines can be used for this purpose, while all of our inducible KI lines are highly effective for this purpose.

Finally, genetic tools for GLU progenitors are particularly poor. Beside a generic Emx1 Tg line, there is only the Cux2-Cre and Cux2-CreER lines, which label relatively sparse radial glia cells (RGs) (Franco et

al., 2012). Our multiple inducible TF driver lines present the first systematic toolkit for RGs, IPs and neurogenic progenitors. In particular, we provide new data that Lhx2-CreER and Fezf2-Flp intersection/subtraction reveals, for the first time to our knowledge, three distinct sets TF-defined RG subpopulations (Fig 2s-v, Ext data Fig 1-2).

Supplementary Table 2

Types	Layers	Existing	Matho et al
IT	L2	none	Tbr2-2A-CreER TM E17.5, E18.5; Tbr2-2A-CreER;PlxnD1-Flp TM E17.5, E18.5;
Cstr Callosal association	L3	none	Tbr2-2A-CreER TM E16.5
	L2-4	Cux2-Cre, CreER Rasgrf2-dCre* (weak)	Cux1-2ACreER
	Cortex- projecting	none	Cux1-2ACreER; project within cortex and not to striatum
	L5a	Tlx3-Cre (Tg) molecular identity unclear	PlxnD1-2A-Flp;Tbr2-CreER E13.5
	L6	None	likely Tbr1-CreER
ET	L5b	Sim1-Cre (Tg); weak or variable expression in posterior cortex; molecular identity unclear	Fezf2-2A-CreER (TM > P1), Fezf2-2A-Flp Adcyap-2A-CreER TM E17.5 Tcerg11-2A-CreER TM E17.5 Sema3E-CreER
CT	L6	Ntsr1-Cre (Tg); no expression in medial cortex and lateral MOp, MOs; molecular identity unclear	Tle4-2A-CreER, postnatal TM Tbr1-2A-CreER, postnatal TM Foxp2-Cre
Single cells	L2-4	Cux2-CreER	Tbr2-2ACreER TM E15.5-E18.5; PlxnD1-2A-CreER, postnatal TM Cux1-2A-CreER, postnatal TM
	L5/6	none	Fezf2-2A-CreER, postnatal TM Tle4-2A-CreER, postnatal TM Tbr1-2A-CreER, postnatal TM
Progenitors	Radial glia	Emx1-CreER (Tg) Cux2-CreER	Lhx2-2ACreER Fezf2-2A-CreER, Flp Cux1-2A-CreER
	Intermediate progenitor	Tbr2-CreER (inactivates endogenous Tbr2)	Tbr2-2A-CreER Tbr2-2A-FlpER
	Neurogenic	Tis21-CreER (not public available)	Tis21-2A-CreER Tis21-2A-FlpER

* destabilized Cre, weak; # not public available

c) how do these types correspond to what we think the existing genetically or projectionally defined cell types are.

The short answer is that our new driver lines provide the most reliable tools that capture major transcriptomic and anatomic PyN subclasses based on molecular and developmental programs. The finer “types” within these subclasses are currently not well established as there is not yet good correspondence between transcriptomic and anatomic types. Our new driver lines and intersectional strategies provide a much needed experimental system that will facilitate the clarification and discovery of finer PyN types. I elaborate this point further in the next few paragraphs.

There has been tremendous progress in the past three years in large-scale single cell profiling of transcriptomic (Tasic 2018) and anatomical (Gouvens 2019; Peng 2020) features of PyNs, resulting in comprehensive classification schemes and taxonomies. However, our understanding of PyN types and their organization remain quite incomplete, especially at finer granularity. Currently, the correspondence between transcriptomic types and anatomic (morphology, projection) types is coarse or poor beyond major subclasses. We do not yet have a consensus definition of “PyN type” and an understanding of the principles of PyN organization. For example, scRNAseq analyses defined 5 to 6 PyN subclasses and ~50 “atomic types”. Whereas the major transcriptomic subclasses correspond with major projection subclasses (IT, ET, CT, NP, L6b), the biological validity of finer transcriptional clusters are mostly unclear. These “transcriptomic types” are substantially influenced by relatively arbitrary clustering parameters. For example, the SST interneurons may comprise 15 or 30 types depending on the clustering parameter (Tasic 2018, 2019), and the same is true for PyN transcriptomic types. Therefore, the current cortical neuron transcriptomic taxonomy (i.e. “genetic type”) is only a working draft (Tasic 2018); we expect significant modifications and improvement as multi-modal datasets are integrated. Therefore, rather than attempting to “correspond” to current transcriptomic types, we provide the experimental access to clarify, validate, and discover biological PyN types.

From anatomical analysis, there is general consensus on several broad projection subclasses (IT, PT, CT), supported by transcriptomic features and developmental origins. Beyond this top level, there is increasing evidence for substantial diversity within each subclass in terms of laminar and areal position, and projection targets. However, whether and how such diversity may cluster into distinct subtypes is far from clear. Currently there are few clear correspondence between the anatomic and transcriptomic diversity of PyNs. The deep, medullar-projecting and superficial thalamus-projecting L5B cells in ALM (Economo et al., 2018) represent rare examples. We now provide clear evidence that these two subtypes are contained within the *Fezf2* population (Fig 5c-d). Most importantly, we expect that our inducible driver lines will be crucial to facilitate systematic single neuron reconstruction, already underway (Peng 2020 bioRxiv), which will provide the ground truth for anatomical cell typing.

In summary, currently there are rather few well-demonstrated PyN types based on multi-modal features. Much work is needed to discover the biological basis of PyN types and their organization, and to achieve a biology-grounded taxonomy at finer granularity. In this context, the PyN toolkit we establish will be essential to clarify and improve the transcriptomic and anatomic analysis toward discovering biologically relevant PyN types and their relationships. These driver lines cover each of the major transcriptomic and anatomic subclasses and subpopulations by expression of key transcription factors. They further enable orthogonal intersectional dissection by lineage and birth dating. Importantly, these inducible drivers will enable “saturation single cell reconstruction” in selected areas (Peng et al., 2020 demonstrate this feasibility) to truly examine whether one can discover and define projection subtypes.

(B)

- General housekeeping (see Clarity and context). I think the figures would benefit from covering different projectionally defined classes together (e.g. all the figures pertaining to tools for PT-type neurons, then all the figures pertaining to tools for CT-type, then IT-type; currently parts of these are all in Fig 3 and 4, while 5 and 6 are more accessible to me since they go into detail). Clearing up the references to supplemental material will help too.

- We thank the reviewer for this suggestion, which we considered carefully. Given the entirely new design of Figure 4 that prominently features a side-by-side comparison with existing lines, as well as the side-by-side comparison of our driver lines with existing lines in Figure 3, we think the reviewer would agree that our current presentation is a better option. We have improved our reference to the Extended

data Figures and supplemental materials. Due to the complexity of the datasets, it is not possible to arrange the citation of figures in perfect order.

Clarity and context: lucidity of abstract/summary, appropriateness of abstract, introduction and conclusions

The clarity of the paper is (to me) the main problem. The scope of the work is obviously complex, encompassing a range of new mouse tools for addressing several different classes of cell types. Following the text is thus difficult, since the figure panels are discussed somewhat out of order (text moves between figures and then back and forth).

- We have improved the text by substantial editing. Where feasible we try to present figure panels in order in the text. However, as Rev1 appreciated, due to the complexity of the datasets, it is not possible to arrange the citation of figures in perfect order.

The supplemental material was difficult to assess since it was presented in many formats: excel files, pdf table, and movies.

- We now present these in more consistent format - all in excel format.

The numbering of the movies does not seem to always correspond to the text (once this is figured out, they are nice) and several are not even referenced in the main text or the supplement.

- We have corrected the correspondence text and movies, and referred to all the movies.

As such, I think the paper could be improved by editing (even in the absence of more data).

- I have done substantial editing.

Major Comments:

How do lines compare to the cell types we know?

The main obstacle to my understanding the advance here is trying to make the connection between existing resources and approaches and how the new ones move beyond that. Specifically, I want to know: How do the cell types able to be targeted in the present paper compare with the genetically (such as Zeisel et al 2015, Tasic et al 2018, Hodge et al, 2019) or projectionally defined pyramidal cell types (Chen, Helmchen 2015 Nat Neurosci, Economo et al, 2018, etc.) in other studies? This is discussed transiently: Line 354-355: “Together, this new set of driver lines facilitates experimental access to a set of genetically-defined PT PyNs.” This is a great finding and seems likely to be the case.

- This has been addressed extensively in reply to b) and c) in the opening section.

Line 488 “The relationship between current and previously reported driver lines targeting similar PyN classes remains to be further characterized at the cellular resolution.” Achieving an understanding of this is necessary. It is not as useful to the field to present the new tool without characterizing how it is comparable to the previously reported one.

- This has been addressed extensively in reply to b) and c) in the opening section. The comparisons are presented in an easy-to-understand way in Figs 3 & 4, Extended Data fig 4 and Suppl Table 2. This sentence is now removed.

The statement (Line 88: “Among the challenges of establishing genetic access to PyN types, the first and foremost is specificity (with an appropriate granularity) and the second is comprehensiveness.”) gave me the hope that there would be a range of lines for different IT-type neurons (something similar to M1 or S2 projecting L2/3 neurons in S1, or V1 neurons projecting to LM or PM or other higher order visual areas). I was concerned that the results here don’t quite reach that level of granularity. For example, it might be

possible to indicate whether the new lines cover the same projection classes as existing lines and/or any additional specificity achieved? (e.g. a Table similar to Fig 3a or Table 2 which lists all the PT-type together, categorizing as ... ‘PT-type, thalamus projecting’ or ‘PT-type, thalamus and medulla projecting’, and something similar for IT-type and CT-type?).

- This has been addressed thoroughly in sppTable 2 and in reply to a), b) and c) in the opening section. As we envision, an ideal toolkit would access PyN hierarchical organization at multiple levels with increasing granularity. We provide the foundational tools and strategy to achieve this goal. For example, Cux1-CreER is the first line that captures cortex-not-striatum projecting IT cells, and PlxnD1/Tbr2 intersection captures the most fine-grained molecular/anatomical PyN types. Currently, it is not clear whether cortical area specific projections, such as M1 versus S2 projection specificity, can be resolved at the molecular level. The new driver lines provide more opportunities to test this possibility in the future.

How effective do you assess the transcription factor strategy to be? I would like to believe that the hierarchical transcription factor strategy would minimize this, but some of the images suggest that genetically defined cells in a single line might contribute to multiple projectionally-defined populations. I would have hypothesized that a genetically homogenous population would project to a well-defined set of targets. Do you feel that these mouse lines support or reject that? Do some lines have multiple populations mixed together (PT-type and CT-type or PT-type and IT-type in the same line)? If we must reject this hypothesis, it suggests that maybe retrograde injections will be necessary for the foreseeable future to target projectionally-defined types, unless the enhancer-based techniques give an alternative effective method (Blankvoort et al; Allen Institute).

- We thank Rev1 for bringing up this important point to discuss - the relationship between transcription factor expression and “PyN types”. The key is at what levels of granularity do we examine this relationship. It is now well-established that hierarchically organized TF networks underlie hierarchically organized cortical neuron types. This is well demonstrated for GLU PyNs (Woodworth 2012; Greig et al., 2013) and GABA INs (Rubenstein, 2015). While we have better grasp of master TFs at the higher level that specify major classes (Fezf2, Lhx2, Cux, Tbr1 etc), we have less understanding of the “lower level” TFs that may specify/maintain finer grained “types”.

At the higher level, Fezf2 (ET), Tbr1 (CT), Tle4 (CT), Cux1/2 (IT) etc define major subclasses and subpopulations quite well. Cells expressing such a master TF likely share certain basic properties, but they are not genetically or anatomically “homogenous” at the finer granularity. These driver lines do provide a solid foundation for dissecting finer types within each subclass by molecular intersection, target specific retrograde labeling, and their combinations. Our finding that Cux1 defines a subset of cortex-projecting IT cells provides compelling evidence for finer correspondence between TFs and PyN types. Thus this correspondence is worth being examined more systematically by comprehensive profiling of TF expression in projection defined subpopulations in the future (e.g. as the study from Tom Jessell’s lab on TF expression and spinal interneuron types; Bikoff et al., 2017 Cell).

Rev1 also raised another interesting and important point – that some of these TFs do not “perfectly” parse the major ET, IT, CT classes. For example, the Fezf2+ population contains a small fraction of PlxnD1+ cells located at the top of L5, which display IT cell features; it also contains a set of L6 Tle4+ cells that are possibly CT cells. Similarly, Tbr1 is a key TF for CT cells, but some Tbr1+ cell also project to contra-lateral cortex - an IT characteristic. Thus the relationship between TF and projection patterns is more nuanced. One possibility is that TF expressions are a bit “sloppy” and “leak” to other subclasses, an inconvenience we have to live with. Another more interesting and intriguing possibility is that these results may be revealing novel biology. Given that *Fezf2* and *Tbr1* are crucial fate restricting TFs likely regulate a set of associated target genes, and ectopic *Fezf2* expression results in reprogramming of cell fate (Rouaux and Arlotta, 2013), it seems highly unlikely their expression can afford to be sloppy and leaky. These results raise the intriguing possibility that the *Fezf2*- and *Tbr1*- expressing “IT-like” cells

might represent previously unrecognized novel or “intermediate” PyN types. This possibility can be examined by using intersectional strategies to reveal their projection patterns and by single cell reconstruction; the results may revise our understanding of how these master TFs shape PyN fate. Therefore, the true value of these driver lines is that they provide reliable experimental access to dissect biologically relevant PyN subpopulations, even though these may defy the traditional classification scheme. This is essential to understand the biological basis of “PyN types” and their organizational principle. We have added new data on these findings in Ext data Fig 6 (on Tbr1 driver) and Ext data Fig 8, 9 (on Fezf2 driver) with associated text in Results. We elaborate these points with a few succinct sentences in the Discussion.

Beyond these conceptual issues, to achieve more specific cell type targeting, additional viral strategies such as retrograde targeting will be necessary to complement the TF driver lines for targeting more specific cell types defined by molecular programs and projection targets.

A potential problem for this approach [is that] different cell types might be generated at different times with some probability (potentially overlapping developmental periods when multiple cell types are being generated) as well as if cell types are specified not only by whether a given transcription factor is on/off, but if these genes could be expressed at different levels (a low enough level to mediate recombinase activity but not drive high levels of all the genetic targets needed for differentiation into a given cell type). Are there some labeled neurons using these approaches that don’t fit the expected pattern? To what degree is this a problem?

- Rev1 raised three additional important points to discuss.

First, “different cell types might be generated at different times”. This is precisely the reason why we generated mostly inducible CreER drivers. These lines will allow the investigator to systematically discover how TF-defined progenitors and lineage specify PyN types at different developmental times. Thus this is not a “potential problem” but rather a major strength of our tools and approach.

Second, the “level of” gene expression in relation to recombinase activity and cell labeling. In mature cell types, many of the key cell identity TFs express in a largely ON/OFF pattern rather graded pattern (e.g. like transmitter receptors and certain signaling proteins). Our transcriptome analysis demonstrated this pattern clearly (Paul et al., 2017 Cell). In early postmitotic cells, there is good evidence that opposing TFs compete to secure nascent cell fate, thus low level TF expression may not represent the intended cell type, as Rev1 correctly pointed out. But this is not a problem for using these drivers because, for targeting cell types, we use postnatal induction when the relationship between TF expression and cell type is well established. For fate mapping studies in progenitors or early postmitotic cells, one should keep this issue in mind when interpreting the results. It is important and satisfying to note that in both *Fezf2*- and *Lhx2*-CreER drivers, embryonic recombination patterns precisely recapitulate their expression gradient along the anterior-posterior and medial-lateral axes (Fig. 2; Extended Data Fig.1,2), indicating a tight quantitative relationship between TF expression levels and Cre recombination levels.

Third, whether there are cases where our results don’t fit the “expected” pattern. Yes, there are a number of such cases, as mentioned above. For example, by careful intersectional targeting combined with mRNA in situ analysis, we found that *Fezf2* does not perfectly divide PT from IT and CT cells. There is a small set of *Fezf2* cells at the very top of L5b that also express *PlxnD1* and project to the striatum (Ext data Fig. 9; we do not know whether they also project to other subcortical targets). There is also a set of low level *Fezf2*+ cells in L6 that are *Tle4*+ (Rebuttal Figure 1). An interesting and intriguing possibility is that these *Fezf2*+ “IT” and “CT” cells might represent previously unrecognized novel or “intermediate” cell types. We favor this possibility for two reasons. First, “unexpected” *Fezf2* expression is spatially not random among *Fezf2* cells - they are restricted either to the very top L5B or L6, but not

the L5b proper, thus unlikely due to simply sloppy control. Second, as a master transcription regulator with many downstream target genes, there is good reason to think that its expression would have a significant consequence in cell phenotypes. The important point is this: now that we have reliable tools that precisely recapitulate the patterns of these three genes (Fezf2, PlxnD1, Tle4), we can use genetic intersection and subtraction to distinguish the above two possibility, and the results may lead to novel discoveries and conceptual advances. We have added new data on these findings in Ext data Fig 6 (on Tbr1 driver) and Ext data Fig 8, 9 (on Fezf2 driver) with associated text in Results and Discussion.

Rebuttal Figure 1. Adult induced Fezf2CreER;Ai14 sections stained with Tle4 antibody demonstrate Fezf2 expression in a subpopulation of Tle4+ layer 6 neurons (a, a'). The levels of Fezf2 expression were not quantified in this experiment, but situ hybridization for Fezf2 mRNA from our lab, consistent with those from Allen Institute, show lower intensity Fezf2 expression in layer 6 compared to 5. Preliminary double fluorescent in situ hybridization experiments with Fezf2 and Tle4 mRNA were done to address this question, and Tle4+ Fezf2+ layer 6 neurons indeed show lower intensity Fezf2 mRNA signal than Fezf2 single positive neurons from layer 5b (b, b'). Experiment images are above. Thalamus projecting Fezf2+ neurons (Fig 5 and extended data Fig. 8) include layers 5 and 6 and serve as supporting evidence that Fezf2 layer 6 contains corticothalamic cells that are Tle4+.

Over what cortical areas (m/l or a/p or granular/agranular) do these lines work best?

In characterizing these lines, it would be helpful to understand how broadly (different cortical areas from medial to lateral, granular/agranular) these new lines work? Medial-to-lateral, Extended Data Fig 3 is the closest which is very nice for the lines that are presented. But this isn't available for all lines. But it seems like the images exist for all the lines. In other pyramidal cell lines (GENSAT, for example), the number of neurons labeled varies across the tangential surface of cortex. This opens the possibility that either neurons of potentially the same type are missed (for example, pyramidal tract type cells can be ID'd by retrograde approaches in many cortical areas, such as medial wall areas, but seemingly not well by the GENSAT transgenics or the current methods). Thus, there is something about this that suggests that neither approach does a comprehensive job of targeting projectionally defined cell types. Isn't it a concern for the logic of the approach (hierarchical use of transcription factors) that some brain areas don't obey these rules (e.g. the transcriptional control of PT-type neuron generation in medial sites may differ from lateral sites)?

- We thank Rev1 for raising this important point. We consider areal expression a very important feature of cell types and driver lines. The answer is quite simple: because all the KI lines precisely recapitulate endogenous gene expression, their areal recombination patterns are very similar if not identical to the mRNA in situ pattern (e.g. Ext Data Fig 8a), which are well demonstrated in the Allen Mouse Brain Atlas. This is in sharp contrast to the GENSAT Tg lines (e.g. Sim1, Ntsr1), which vary across the

tangential surface of cortex for unknown reasons, as Rev1 rightly pointed out. We discussed this point in the opening section a) and b) and summarized results in Extended Data Fig 4 and Suppl Table 2. Our KI lines are distributed more homogeneously across cortical areas (Extended Data Fig 4b).

In some cases, the CreER drivers induced at a particular time may not cover all areas. For example, postnatal Cux1 expression in medial cortex is low (Extended Data Fig 4 a-b). We interpret this to indicate that either the medial cortex does not contain the “equivalent cell type” or that they are significantly different. Thus in such cases, it is not necessarily that the KI driver line does not do “a comprehensive job of targeting projectionally defined cell types”. Rather, by recapitulating the endogenous expression pattern of the TF, the driver line may reveal differential areal organization of the PyN type.

We also point out that, as gene expression is often developmentally dynamic, the time of tamoxifen induction influences the cellular and areal patterns of recombination in our CreER drivers. In most lines though (e.g. Fezf2, PlxnD1, Tle4, Tbr1, Cux1, etc), postnatal expression is very stable and postnatal induction reliably covers most of the dorsal cortex and is highly consistent with P28 and P56 in situ patterns of the Allen Mouse Brain Atlas. The variation in density across areas likely reflects the endogenous variation of the corresponding cell types. This biological variability/pattern is fundamentally different from “artificial variability” in Tg driver lines, which results from unexplainable arbitrary ectopic interactions between transgene promoter and genome regulatory elements near the transgene.

We have used our driver lines for extensive wide-field Ca imaging, optogenetic scanning, which leverage on the wide range coverage of cortical areas in these drivers. These studies will be presented in separate papers.

What about the fraction of neurons labeled in each of these approaches?

Is there some idea of the fraction of total PT-type cells each line might label? Is the change in % labeled consistent with the hierarchical level? (e.g. successively fewer % labeled for later-acting TFs?) Fig 3c suggests that Fezf2 labels far more neurons in L5B than Adcyap1 and Tcerg11, but what fraction of the overall PT population is this targeting?

- This is another important parameter for driver lines. We are glad to report that Fezf2-Cre/Flp labels > 95% spinal projecting PT cells (Fig. 5a-b), assay by retrograde labeling. For finer PT-types, two factors make such quantifications difficult. First, the lack of well-established pan-ET, PT or type antibodies. Second, some expression levels (e.g. Adcyap1 and Tcerg11) change across different ages and tamoxifen CreER induction has inherent variability. E17 induction Adcyap1 and Tcerg11 labels a significant and more restricted fraction of L5b subset of PT cells, consistent with its endogenous pattern and as a target of Fezf2 regulation (Figure 3, 4, Extended data figure 4). Also see Rebuttal Figure 2 below.

Supplemental Info and Tables need some detailed proofreading/examination.

The text reads “Adeno-associated viruses (AAVs) serotype 8, 9, DJ PHP.eB or retro2 packaged by commercial vector core facilities (UNC Vector Core, ETH Zurich, Biohippo, Penn, Addgene, ...) were used as listed in Supplementary Table 2.” where the ellipsis might be omitted. The table I think is one of the Excel files, which lists all injections for all figures, but contains many blanks and isn’t titled. This is in contrast to other Tables that are titled or are in a different format (S4 versus S1, say). Furthermore, some movies are referred to incorrectly and not all movies are mentioned in the text (Movies 14, 15, 16 never mentioned). It was frustrating to try to be thorough and review the data as the text directs me to, and find it difficult to connect which movie went with which statement.

Here are some examples of where this arose: Movies 6 and 7 are referenced (lines 339-341) but the actual movies are likely to be 7 and 8. The movie files don’t contain text to differentiate in the movie itself, so I went to the titles for the uploads (see below). But even then, the same Movie 7 is referred to in the text as pertaining to Adcyap1 (Line 341) and Tcerg11 (Line 345). The imaging is beautiful and deserves to be

examined, but the way the files are presented and referenced makes it frustrating and puts the onus on the reader to figure out what's going on. This could be improved by potentially including a caption in the video itself to disambiguate. I have not yet figured out where the higher resolution images on which the movies are based are stored, but these are the sort of thing I would want to peruse at leisure if there were a site/viewer with which to do it. (<http://www.brainimagelibrary.org/download.html> or <http://www.brainimagelibrary.org/datasets.html>) from the methods seems to be the site at which to do it, but this is not as user friendly as other sites (I don't see how to browse images) and requires me to figure out some identifying number for the specific brain I want to see. Possibly some of this information is the supplemental tables. To test this, I tried using Image ID from Table 2 to search, but didn't find it and the website for most of the images in that table are blank. Brains 180804 and 180820 are listed in both the PSC website and the Table, but others are not. I imagine others would want to browse these images too!

- We have substantially improved the trackability of datasets from IDs in Supplemental Tables. They are now in consistent format as excel files.
- We have improved movie references and annotations. All movie files can be downloaded and whole brain STPT datasets registered to the CCFv3 are readily accessible on the Brain Architecture Portal by Partha Mitra's lab (this portal is currently undergoing some restructuring). We are including:
 - > 38 brains on Cell Projection (<http://brainarchitecture.org/cell-type/projection>) for this paper. 18 of them were posted and have been available. 20 brains are yet to be posted by March 22.
 - > 50 brains on Spatial Distribution for this paper. These will be made available by March 22.

Line specific comments:

Cux1:

Cux1-CreER seems to label layers 2-4. Issues include: (1) a difference in the connectivity of L2 and L3, and (2) different subtypes with L3 depending on the projection target (Chen, Helmchen et al Nat Neurosci 2015; Sato, Svoboda et al J Neurosci 2010, etc.).

- These are excellent questions. Following Rev1's suggestion, we did anterograde labeling in Cux1-CreER. Amazingly, we found that Cux1+ PyNs project within the cortex but not to the striatum. This is the only molecularly defined population with such projection specificity. Thus this line is distinct from the Cux2-CreER which labels a mixture of PyNs projecting to both cortical and striatal targets. This finding has major implications. Cux1 is closely homologous to Cux2 yet they differ in function as revealed by knockout phenotypes (Cubelos et al., 2014), the cellular basis for these similarities and differences are unknown. The Cux1 driver now provides exciting opportunities to discover the cellular differences between Cux1+ versus Cux2+ IT cells (although some IT cells may express both), which may contribute to the evolution of primate and human IT cell types (Doan et al., 2016). The Cux1 driver also provides a key entry point to further dissect cortex-projecting L2/3 IT subtypes, including those in Chen, Helmchen et al Nat Neurosci 2015; Sato, Svoboda et al J Neurosci 2010, etc.

The line is also not uniform across layers (as few of these lines are), but it suggests that there are some neurons in these layers that aren't accessible under these strategies. This line requires work with an additional strategy (retrograde label or other) in order to achieve some specificity beyond layers 2/3/4.

- Cux1 expression is indeed not uniform across L2-4, and it is certainly possible that there are other L2/3 IT types not covered by Cux1. On the other hand, we suggest that genetic and projection properties are more fundamental than laminar position in defining cell identity, thus Cux1 likely captures a distinct and important subclass of IT cells regardless of their exact laminar position.

How this might be superior to some of the layer 2/3 Cre lines that already exist? (The Sepw1-Cre_NP39 line (Gerfen et al., 2013), the Cux2-CreERT2 (Franco et al., 2012) and Rasgrf2-2A-dCre (Harris et al., 2014)).

- Cux2-CreER, Cux2-Cre, Rasgrf2-2A-dCre include a mixture of striatal and cortical projecting PyNs. Rasgrf2 expresses a destabilized Cre and is weak (personal communication with Carl Petersen, the only

lab that published a paper using this line). Sepw1-Cre_{NP39} is a Tg line with unknown molecular identity. Our precise targeting of Cux1 is significant for the following reasons. As a prominent homeobox transcription factor with major role in PyN development and evolution, we now convincingly demonstrate, for the first time, that Cux1 is distinct from its close family member Cux2. We believe that Cux1-CreER captures a highly biologically significant subpopulation, which restricts their projections within the cortex. Together with the Cux2 driver, they will have a major impact in understanding cortical organization, development, function, and evolution.

PlexinD1

PlexinD1 also not uniform across layers (assumes the images shown are the PlexinD1-CreER instead of the Flp line but the legend could be more explicit).

- PlxnD1 and Sema3E function as a repulsive signaling mechanism to specify synaptic connectivity and the developmental patterning of vasculature at multiple stages. Although not spatially “uniform” (few gene expression patterns are as Rev1 pointed out), PlxnD1 likely mark a distinct molecular and connectivity defined IT subpopulation.

Is the expression pattern the same in PlexinD1-CreER and PlexinD1-Flp? The example images in 3b (Cre) versus 6e (Flp) look slightly different - the Flp version is more specific for L5 or L5A and also the Flp version is brighter in S1 (some lateral area) than other areas, while perhaps this difference is not as pronounced in the CreER.

- We generated both inducible CreER and constitutive Flp to achieve more versatile and specific targeting of PlxnD1+ PyN types. The CreER and Flp expression are very similar as far as we can tell and as expected from the reliability of the KI approach. When using Cre or Flp dependent AAV as reporter in a postnatal age, they are identical. However, when using reporter lines to visualize recombination patterns, we expect some areal and laminar differences. This is because the constitutive the Flp line integrates across time and represents a cumulative pattern, whereas the CreER line is a “snapshot” of the expression at the time of tamoxifen induction. *PlxnD1-Flp;frt-STOP-frt-RFP* (Fig6a) also include more vascular cells, resulting from recombination in early postnatal times; but use of Flp-dependent AAV would only target PlxnD1+ neurons in this driver line. The main value of PlxnD1-Flp is for designing intersectional schemes to capture laminar subtypes of IT cells (e.g. with Tbr2-CreER, Fig 6, and possibly Cux1-, Cux2-CreER etc).

Is there a figure like the Extended Data Fig 3 “flat map” to compare? PlexinD1 in your images (Fig 3b) has seemingly two clear peaks ~L2 and L5, but the histogram across S1 looks flat in Ext Data Fig 3, why the difference? (Also, I don’t think you need both histograms with different binning if you edit that figure.)

- We thank the reviewer for these comments. In terms of laminar pattern, PlxnD1+ PyNs reside in L5a, L3, L2, with some areal differences. More importantly, we have added a much larger set of lines with areal flat maps as well as annotations of their subcortical expression in Ext Data Fig 4. This should be very useful for users to glean to overall patterns. The PlxnD1 cells are enriched in L2/3 and L5A, but do have sharp peaks in these layers, as shown in our revised Ext data Fig 4. This can be appreciated by looking at the corresponding movie of whole cortex PlxnD1 cell distribution.

Adcyap1

The treatment of the Adcyap1 expression needs improvement. In Fig 3c, Adcyap1 expression looks relatively specifically restricted to L5B. It is sparse and not uniform from medial to lateral. (If Video 7 is typical, that brain had much more widespread expression from medial to lateral). But then the text directs me to the Supplementary Info Fig 3 where the histogram seems to suggest expression across many layers (Present in 2/3 in the histogram, a dip but not absent in 4, more in 5 ... maybe the histogram is just mislabeled). But then example images are back to sparse in L5 targeted in the Supplementary Info Fig 4.

- *Adcyap1* is a neuropeptide and a direct transcription target of FEZF2. Its perinatal expression is highly restricted to a subset of ET cells in upper L5b, but the expression then diverges in postnatal times. Our *Adcyap1-CreER* precisely recapitulates these patterns. The key to capture L5b PT cell is a well-timed TM induction at round E17.

We thank the reviewer for noting the mislabeled patterns, which were switched between *Adcyap1* and *Tcerg11* for cortical distributions in Extended Data Fig 4 and now present the correct cortical distribution patterns. Sparseness in Fig3 is due to TM dose, which can be adjusted and especially suited for single cell analysis.

Tcerg11

I didn't understand the comparison of *Tcerg11* to *Fezf2* (Lines 345-347): "It is currently unclear how these PyNs*Tcerg11* relate to PyNs*Fezf2*. Anterograde AAV labeling of PyNs*Tcerg11* revealed overlap with PyNs*Fezf2* but absence of the collateral to first-order thalamic nucleus VPM". I didn't see (or expect per prior work) much collateralization of any PT-type projection in VPM from L5B cells, and this seems consistent with neither of these lines showing much arborization there (Fig 3c, the VPM and POM row). Compare Fig 3c and Fig 4b. They seem to tell the opposite story for *Fezf2* (Fig 3c has POM >> VPM; Fig 4b has VPM >> POM nearly invisible). Why? I hypothesize (but can't tell) that the sparse *Fezf2* population visible in L6 might contribute corticothalamic output to VPM (from the S1 viral injection). It is possible that the relative absence of VPM in the S1 injection (4b) is because these axons visible in 3c originate from other cortical areas.

- *Tcerg11* is a transcription elongation factor. Similar to *Adcyap1*, its expression is restricted to a subset of L5 ET cells at perinatal time but the expression diverges in postnatal times. One of the authors (Paola Arlotta) carried out single cell analysis of molecular genetic trajectories across cortical neurogenesis and revealed that *Tcerg11* defines a distinct sub-branch of PyNs^{*Fezf2*} postmitotic differentiation (DiBella et al., 2020 BioRxiv; a panel of the Suppl Fig 4 in this paper is shown in rebuttal Fig 2). From our new analysis, we now show that *Tcerg11* more restricted to L5B of the *Fezf2* population (Fig 3), which correlate to more specific projection pattern in comparison to *Fezf2* with respect to subcortical projections within thalamus (Fig. 4e. g). Thus, we expect that future studies (e.g. single cell reconstruction) using this line may reveal more specific subtype(s) of PT cells.

Rebuttal Figure 2. *Tcerg11* defines a distinct sub-branch of PyNs^{*Fezf2*} postmitotic differentiation. **a**, URD (a simulated diffusion-based computational approach) trajectory branching tree of the developing cortex. Cells are colored according to their identity (left) or developmental time of collection (right). **b**, branching trees labelled by the expression of marker genes *Sox2*, *Eomes* and *Neurod2*, showing the distribution and sequential developmental progression of apical progenitors, intermediate progenitors and excitatory neurons, respectively.

Line 217-218: “There was no apparent morphological difference between Fezf2+ and Fezf2- nRGs.” What are you intending to convey here? Prior to this, you’re arguing for heterogeneity in this population. But this suggests you don’t see it (in whatever feature was surveyed for morphology). Seems contradictory. I imagine many people will think that some of the heterogeneity hypothesized might come in the projection targets, and this was not easy to assess (in that figure). So either say what the morphological feature measured is (or leave out)?

- we agree that this sentence is not informative. We have deleted it.

It is interesting in the CT-projection population that Tle 4 but not Tbr1 has a more pronounced intracortical axonal projection to L5A (guess of lamination from images in 3d middle row), suggesting these neuron subsets might not be overlapping.

- The neurite in L4 of the Tle4 are dendrites, not axons. See Ext data Fig 8i. We have good evidence that Tle4 is a subset of L6 Tbr1 CT cells (Fig 4d).

PT-type retrograde label:

I found it interesting that there was no laminar structure to the different L5B cells based on their projection targets (Fig 5d). The graph suggests similar laminar pattern for the three groups (except striatal), though the text argues that cSp5 differs. Is this mostly due to inclusion of some very deep layer cells in layer 6?

- we have added new results that thalamus- and medulla- projecting PT neurons occupy different sublaminar positions in L5 (Fig 5d,e), consistent with Economo et al., 2018 Nature. In general though, this question is not resolved in the field. The challenge is that each PT neuron has multiple projection targets, so it may take systematic single cell reconstruction to obtain the right answer. Again, our molecular defined inducible CreER driver lines are necessary for such single cell analysis.

Are there L5A (presumed IT-type) included in the striatal injection?

- Fezf2 expression is restricted to L5B and does not extend to L5A. However, as mentioned above, a small set of Fezf2+ cells at the very top of L5b are also PlxnD1+ and project to the striatum. It is not known yet whether these cells are IT, PT or a novel “hybrid” type with both striatal, callosal, and subcortical targets. We now include these results in Ext data Fig 9 and associated text in Results. This will make it very clear for the investigators using these lines.

The sample image seems sparse, though I would have anticipated the opposite since so many PT-axons to other subcortical targets have collaterals in striatum (MouseLight has many examples, but I think this is also described as far back as Cajal ... and Charles Wilson? See Kita and Kita J Neurosci 2012; more recently the MouseLight papers Economo et al 2018, Morita et al 2019 Front Neural Circuits, Winnubst et al Cell 2019).

- The lower labeling efficiency is in part due to the time and dose tamoxifen CreER induction, which does not capture all PT cells.

Is this retrograde approach susceptible to any problem of tropism (e.g. AAV2retro infecting all PT-types equally)?

- We addressed this issue with two experiments using either CTB (which has no tropism issue) or HSV (Ext data Fig 8p). In both cases, we see the same labeling pattern as in Fig 5h. So at least in these targets, viral tropism does not appear to contribute to the labeling pattern. We note that rAAV thalamus injection also labels upper layer 5 cells in motor cortex (MO) and L6 cells in SSp, this is evidence that rAAV effectively labels projection specific subtypes, not due to tropism.

This finding seems to slightly contrast to the Economo et al (2018) paper (upper vs lower neurons projecting to thalamus and medulla, respectively). This was also shown in rat by Rojas-Piloni (2017 Nat Communications), with different laminar distributions for ~4 targets. The data here are especially different since the one group that differs is the retrograde experiment from striatum.

- we have addressed this question directly. We added new result that, consistent with Economo 2018, thalamus- and medulla- projecting PT neurons occupy different sublaminar positions in L5 (Fig 5d,e).

If all/most PT-type neurons have a striatal collateral, and this would be the broadest group (e.g that the other subsets might all be included here).

- this is an interesting and important point. Currently there is limited evidence in the literature for very broad as well as rather restricted projection patterns of PT cells. We believe systematic single cell reconstruction is the way to resolve this (see below), and our inducible driver lines are essential for this effort.

- in summary, the question of whether and how the field as a whole can discover and enumerate ET and PT subtypes defined by genetic and anatomic (laminar position and projection/connectivity patterns) features is important and yet complex and unsolved. Our current retrograde labeling in S1 cortex may not have enough power to resolve sublaminar patterns of projection defined ET types. We believe that this important issue needs to be resolved by systematic single neuron reconstruction. Our new set of molecular defined and CreER inducible ET/PT drivers greatly facilitate such single cell analysis. Indeed, they are highly featured in the biccn companion single neuron reconstruction paper (Peng et al 2020, bioRxiv)

Complex genetic strategy:

Line 467: “using just three alleles” For many groups this approach will be expensive and complex. The number of mice/transgenes required to pursue some of these approaches (potentially three transgenes plus viral injections) will be a challenge for some smaller labs to adopt due to cost/space. Thus, it is likely that adoption will not be uniform for the field. So, can the authors argue for the cases where a more well-defined subset of cell types is targeted that can't be achieved other ways, so as to justify the cost?

- We appreciate this comment and suggestion. It is now well established that there is no simple relationship between single gene expression and cell type, especially at finer granularity, but combinations of multiple genes do seem define cell types quite well. This presents a fundamental challenge for germline approach to cell types in the mouse, as the combination of multiple alleles is effort intensive. We previously designed a highly effective intersectional strategy to target specific interneuron types by combining molecular marker, developmental program, and anatomy (He et al 2016). Here we show that this strategy also works extremely well for targeting highly specific PyN types. There are no other methods that we currently can envision that would achieve the level of specificity to capture L2 versus L5 P1xnD1⁺ IT cells. This is a significant achievement and will facilitate important discoveries worthy of the effort (think about Drosophila genetics). On the other hand, we agree with Rev1 that combining 3 alleles is still a significant hurdle for labs not well versed with mouse genetics. We and others are developing enhancer based methods and other new technologies to tackle this major challenge.

Minor Points (Including typos and presentation):

Intro Line 56: Some recent evidence for (rare) GABAergic projection neurons include:

<https://www.ncbi.nlm.nih.gov/pubmed/25164650>

<https://pubmed.ncbi.nlm.nih.gov/27159237/>

- We thank Rev1 for pointing out these rare cases. We have modified our statement to be less categorical for the distinction between local GABA interneuron and long range GLU projection neurons.

In many cases, an acronym is used ('IS mice', line 213) prior to being defined in the text (line 403). Other places include Line 95: TF-defined RGs? Transcription factor? It wasn't defined yet. Line 154: Define NEs before using acronym.

- We now make sure to define acronyms before their use.

Items listed as Tables (Table 1, Table 2) are presented to the reviewers in the Supplemental Information pdf and not with the manuscript (makes it harder to find when reviewing). It's not clear if the idea is to present in the paper or in the supplement once formatted. This info (Tables 1+2) is quite useful but does take up space (the list of lines is partially duplicative with the table in Fig 3).

- We now refer to all Tables and Supplemental Tables and make them all in excel format.

Line 141: "In a similar yet contrasting way", probably not ideal phrase.

- we have modified this sentence to "*Fezf2* is similarly expressed from early RGs ~~and~~ but subsequently functions as a master regulator for the specification and differentiation of infragranular corticofugal PyNs^{23,24}.

Fig 1e: Cartoon looks like a more defined trajectory than the inset in 1c (where IT's seemingly continue to be produced at later times).

- Fig 1e cartoon is a highly simplified depiction of a highly complex developmental process, and is a description of the general understanding that corticothalamic and corticofugal PyNs are generated earliest in development.

The order of the panels for presentation/discussion could be improved (e.g. present figures and panels in the order they are discussed in the text? Fig 2q-s discussed after 2t-u).

- we have made an effort to match the order of the text and figure panels. Due to the complexity of figures and space limitations, it is not always possible to have a perfect match.

Fig 2: For panels 2e-u, where are the scale bars? These don't seem to be in the legend. For these same panels, are the authors sure the inset is the same as the boxed area in the left images (2e, 2g 2i, etc.)? This is not obvious and in the case of 2e, as the intense fluorescence of the low magnification image in the top and bottom layers is not as apparent in the magnified image. Why is the RGBow approach used for 2f but not h and j? This is in some panels and not others (also in the *Fezf2* part of the figure). The blue channel is visible in some but not others.

- We have added scale bars

- The high magnification inset for Figure 2 embryonic analysis is not a maximum intensity projection so that the progenitors can be examined for their morphological features. Therefore, the high-mag inset does not contain all the cells seen in the low mag image. We have added a sentence in the legend for Figure 2 and Extended Data Figure 1 stating the same. Please see the max intensity projection in high mag for previous Fig 2e (new Fig. 2d) with a side-by-side comparison of what we present in the figure. The max intensity projection shows the intense fluorescence in top and bottom layers as seen in the low mag image. This does now allow us to analyze morphological features of progenitors.

Rebuttal Figure 3

- The RGBow approach is used at E10.5 (Figure 2e, n) specifically to show the clonal relationship between different PyN progeny. We have fixed the blue channel for Figure 2n (*Fezf2-CreER*; RGBow)

The use of the IS mouse has some minor inconsistency. I thought this mouse is supposed to express either red or green. “Express tdTomato for “Cre NOT Flp” subtraction and GFP for “Cre AND Flp” intersection” (Miao et al 2016). But the double expressing cells (Fig. 2) are not explored in great detail other than the explanation that “The mixed GFP-RFP clones are likely derived from *Fezf2*- nRGs in which *Tis21*-CreER first activated RFP expression followed by postmitotic activation of GFP through *Fezf2*-Flp (Fig. 2u, Extended Data Fig. 2d, d’, g).” Maybe this is a reasonable explanation but is not directly tested. Since the cell was born embryonically, but the dual expression persists at P30, doesn’t this suggest the possibility that some cells in the IS reporter line are capable of stable expression of both red and green?

- Here we need to make a clearer distinction for our description of “cell” versus “clone” (which consists of cell siblings from the same progenitor). At the cell level, each cell expresses either RFP or GFP but not both (i.e. “double-expressing”). At the clone level, a clone can consist of all-RFP, all-GFP, and a mixture of RFP and GFP cells. “The mixed GFP-RFP clones” refers to clones, not cells. We have made the description more clear to “The mixed clones containing both GFP and RFP cells ...”

Line 333: “anteropretectal nucleus” I think means anterior pretectal nucleus (References I looked at did not call it that.)

- we have made the suggested change.

Line 346: Calls the gene *Tcerg11* instead of *Tcerg 11* in one place.

- corrected.

Electrophysiology: Why 2/2 mM Ca and Mg?

- In most slice physiology experiments, 1-3 mM of Ca²⁺ have been used, and usually lower concentration for Mg²⁺. Higher Mg²⁺ (2 mM) have been used to keep the slice from being too active and for a longer time. The impact of relative concentration of Ca/Mg has been discussed in a review (Clements and Silver 2000 TINS), which could affect the release probability of synapse. we have published this recording condition before (e.g. Lu et al., Nat Neuroscience 2017)

Fig 4c: Scale bar typo fraction not ‘faction’.

- corrected.

Fig 5c: What are the inset cartoons in 5c used for? I see a few areas identified but then no corresponding images (or discussion in the figure legend). Maybe this is where the retrograde AAV was injected (and

5b shows only one case)? Why do some images have DAPI, others do not? Why is the level of RFP expression high for most cases, but so low in the cSpV case? Is this region in the same cortical area as the others?

- we thank the review for this comment. We have re-designed the cartoons in Fig.5 and added significant new data. We think the layout is now easier to understand.

Fig 6jk: The cStr and iStr labels not ideally placed (white on white).

- we have improved these labels.

References: Ref #11 missing year

- added.

Referee #2 (Remarks to the Author):

Matho et al. generate and describe new mouse models to dissect cortical circuits, by enabling to target specific subpopulations of pyramidal neurons. The models rely on the combinatorial use of inducible CreER or Flp, knocked in the endogenous locus of about 10 genes displaying specific patterns of expression in the developing or adult cortex.

They first describe two drivers (FexFf2 and Lhx2) labeling radial glial cells that may constitute two distinct but partially overlapping pools, that seemingly generate most pyramidal neurons, and drivers labeling committed progenitors (Tis21 and Tbr2 genes), enabling to target distinct, mostly layer-specific, subtypes using timed induction.

These are complemented by about 10 more specific lines that enable collectively (typically using intersectional strategies) to label in a specific way a specific repertoire of pyramidal neuron subsets with distinct patterns of projections (intratelencephalic, subcerebral, thalamic) and/or layer identity. They provide some examples of how these can be used to identify anatomically some subtypes, characterized by specific axonal or dendritic patterns that would otherwise be indistinguishable by existing genetic or anatomical labeling techniques.

Overall this study provides a comprehensive resource that may be useful for a broad range of applications, from development to functional analyses of defined populations of neurons. It goes beyond the state of the art by the degree of specificity achieved, that results from the genetic targeting strategy as well as combinatorial aspects relying on regional and temporal patterns of expression.

One weakness is that there are no major novel insights on cell diversity of the cortex, as the intriguing observations that suggest unexpected cell diversity among some of the labeled progenitor or neuron subpopulations, are not investigated in depth. This study should thus be considered mostly as the description of a resource to the community for future studies, more than a direct approach to tackle cortical cell diversity in a systematic way.

- We agree with the overall assessment of Rev2. The goal of our work is mainly to provide an essential resource and strategy for studying cortical pyramidal neurons. Importantly, we also establish a compelling experimental paradigm to systematically dissect the hierarchical organization and developmental trajectory of PyN types. This is extensively elaborated in the opening section of the rebuttal (to Rev1).

In this context one limitation is that the targeted neuronal types are not linked in any way to the emerging cortical neuronal identities recently identified by single cell transcriptomic analyses. This would greatly increase the impact and applicability of the models described. It could be done by profiling the labeled

cells in at least a few of the described lines, followed by direct comparison with the existing transcriptome datasets.

- We have addressed this comment extensively in the opening section c). Along the current hierarchical transcriptomic taxonomy (Tasic 2018), the top level major classes have good correspondence with known anatomic classes and developmental lineage, but the bottom level “transcriptomic types” are largely statistical constructs for which the biological validity remains unclear. Because KI lines precisely recapitulate endogenous gene expression, most of our driver lines have clear correspondence to the major transcriptomic subclasses (IT-Cux1, PlxnD1, Tbr2, ET-Fezf2, Adcyap1, Tcerg11, CT-Tbr1, Tle4, Foxp2). They provide important entry points to validate and discover finer types and subtypes within each population, including possible transcriptomic types in Tasic 2018 – the goal of this work. A simple profiling of a few driver lines is unlikely to be sufficient/informative to establish the correspondence with current transcriptomic types, and a full validation is beyond the scope of current paper.

The authors show data suggesting that *fezf2* and *Lhx2*- CreER strains label different populations of progenitors, based on cell distribution and outcome. The authors should provide more evidence, in particular quantification (more cells, different litters,..) to determine if the qualitative differences that they observe are indeed robust and reproducible and not the mere result of variability in labeling efficiency. Transcriptome profiling of each cell population would be another complementary way.

- We thank Rev2 for the suggestion. We have carried out extensive quantification of our fate mapping results in more cells and litters and presented these in Fig. 2 and Extended Data Fig 1,2.

Similarly, the authors describe distinct profiles of *Tbr2*/*Tis21* labeled cells that indicate that “most *Tbr2*+ progenitors are neurogenic instead of transit-amplifying”. It is not clear how the data presented lead unequivocally to this conclusion. Clearly more should be done to support this conclusion, in particular by comparing *Tbr2*/*Tis21* and *Tbr2* cell labelling at shorter time points, and using clonal (low TAM induction) analyses following the induction.

- We agree with Rev2 on this point. The presence of transit-amplifying IPs is minor in the mouse cortex. While *Tbr2*/*Tis21* intersection should capture this population, it is a subtle and complicated issue. We have now tuned down this result and moved it to Extended Data Fig. 3c-e.

The authors state that all the lines described allow expression that is faithful to the endogenous gene, but only reporter gene expression are shown. This is particularly important since the pattern of reporter genes also reflects lineage and not just expression. Importantly, they should also demonstrate that the expression pattern of the endogenous gene is unaltered by the targeting. The authors should provide more data on these important aspects, for instance using *in situ* hybridization of Cre / Flpe compared with endogenous gene expression, in the various knock-in and control lines.

- We thank the reviewer for discussing this important point. As elaborated in the opening section 2), we and the mouse genetic community have overwhelming evidence on the KI approach for recapitulating endogenous gene expression. To provide more direct evidence, we have performed multiple fluorescence mRNA *in situ* on several driver lines Ext data Fig 8. We have also performed anti-LHX2 immunohistochemistry on *Lhx2*-CreER tissue to show direct evidence of recapitulation of endogenous pattern of expression (Extended Data Fig. 2a). In addition, we now include numerous side-by-side comparisons of recombination pattern and *in situ* data from Allen Brain Atlas or the literature (Extended Data Fig 2). As shown, the correspondence between recombination and *in situ* pattern is truly excellent, including the gradient pattern of *Fezf2* and *Lhx2* from E10-E13. These results go a long way to show the high value of these driver lines on precisely leveraging the biological processing and mechanism of key transcription factors.

Regarding the effect of KI cassette on endogenous gene expression, we designed all KI lines by inserting Cre/Flp after the endogenous translation sequence, which should minimize the impact. Indeed, all KI

breed as homozygotes and we do not observe notable phenotypes, as other GABA neuron KI lines we generated before (Taniguchi et al., 2011 Neuron; He et al., 2016 Neuron). However, as any genomic modification is a genomic lesion, we cannot rule out the possibility of more subtle alterations of endogenous gene expression.

The authors provide information on layer distribution of the reporters in various cortical areas. They should also provide a similar information for axonal projection patterns. This would be indeed extremely useful information as these are likely to differ depending on the area considered.

- We thank Rev2 for this suggestion. We have now included this information in Fig. 4a-c, e-h, Fig 5, ext data Fig 7. We have also performed a side-by-side comparison of projection patterns between our newly generated PyN driver lines and existing driver lines (Fig. 4a-c, e-h, suppl Table 2).

In addition, we have performed axon projection mapping of multiple driver lines in many other cortical areas. These results are available through the BCDC (Brain Cell Data Center) and on the Brain Architecture portal (we are also improving the metadata entries so that these resources may become more accessible).

The authors should describe in much more detail the mode of administration of Tamoxifen (timing, method, concentration etc) that is given to the mice for timely inductions to achieve each given profile of cell specificity. This is not trivial given the known variability of this method to achieve reliable and safe induction of gene expression. Similarly, the mouse strain background should be specified and any known strain difference in specificity or efficiency of the induction of the reporters should be explicit.

- We have included these information in the Method section.

The various crossings and reporters used should be schematized more systematically in each figure : It would make the study much easier to follow.

- We thank the review for the suggestion. We have improved the schematics in Figs. 2, 5, 6.

Referee #3 (Remarks to the Author):

Advances in sequencing technologies have greatly improved our understanding of the genetic identity of neuronal subtypes in the mammalian brain. However, in part due to a lack of genetic access, much less is known about the functional and anatomical diversity of pyramidal neurons and their underlying developmental lineage. In this manuscript Matho et al. describe novel genetic knockin lines that make use of developmentally-timed induction to target specific neuronal progenitor pools. By combining these lines with intersectional reporters they were able to target sets of neurons defined by birth time, marker expression, anatomical location and projection targets. The resources and techniques described in this manuscript are innovative and are combined in highly creative ways. This will likely open up new possibilities for researchers studying cell-type specific neuronal functioning, connectivity, and developmental neurogenesis. However, the impact of this resource is undercut by a lack of detailed characterization of the presented lines. Furthermore, the potential of combining this approach with retrograde tracing to characterize PyNs subtypes based on projection targets is only marginally explored. As a result, there is a lack of novel biological insights that would make it valuable to a larger audience.

- We thank Rev3 for his/her overall enthusiasm. Rev3's main critique is similar to that of Rev2. We have addressed the significance and the weakness of our paper in our reply in the opening section and to Rev2. Following Rev3's comments, we have added substantially more characterization throughout the paper in new figure panels and tables as summarized at the beginning.

Major Points

1. How specific, and non-overlapping, are the populations labeled with the described approach? This is not made explicit in several parts of the paper. For example the *Lhx2* and *Fezf2*-CreER driver lines are described as giving ‘partially overlapping as well as possibly distinct RG pools.’ (190-191). Are *Lhx2* and *Fezf2* chosen because they are non-overlapping during development? Could the authors perform stainings for the opposite TF with each driver line to investigate this? Later on it is mentioned that ‘their progeny might differ in terms of projection pattern and connectivity when analyzed at cellular and clonal resolution’ (192-193). Unfortunately, this is not further investigated later in the paper which prevents any conclusive statements on ‘whether different RG subpopulations ..contribute, ultimately, to projection-defined PyN subpopulations’ (line 81-83). In general, it is often ambiguous if parts of this section serve as validation or aim to present novel biological findings on corticogenesis as the authors often state that results capture known endogenous expression patterns.

- We thank the Rev3 for this comment. We have now addressed these two concerns.

> “how specific”: as elaborated earlier and demonstrated with in situ where feasible (Extended Data Fig. 2), our KI driver lines are highly specific and in most cases precisely recapitulated endogenous gene expression. Using anti-LHX2 antibody, we have validated *Lhx2*-CreER (Extended Data Fig. 2a).

> “how non-overlapping”: this is a very important question for distinguishing cell type at single cell resolution, but often cannot be easily achieved due to lack of proper antibodies or ambiguity of in situ. For example, there are no good commercial Abs for FEZF2. We have also described the relationship between *Lhx2* and *Fezf2* using anti-LHX2 immunohistochemistry on *Fezf2*-CreER and *Fezf2*-FlpO embryos, showing that 10% of *Fezf2*⁺ RGs do not express LHX2 (Extended Data Fig. 2). Furthermore, we have used *Lhx2*-CreER, *Fezf2*-Flp intersection/subtraction (*IS* reporter) to convincingly demonstrate the presence of *Lhx2*⁺/*Fezf2*⁺ and *Lhx2*⁺/*Fezf2*⁻ RG subpopulations (Fig. 2s-v; Extended Data Fig. 1j). Therefore, we have convincingly demonstrated three distinct RG populations defined by differential expression of *Fezf2* and *Lhx2*, to our knowledge the first such evidence in the field. We have obtained further evidence that PyN progeny from these two RG subpopulations have distinct projection patterns, but those results will constitute a different paper.

More generally, this result is a compelling demonstration of the power of our approach: to address the issue of “overlapping or not”, we need the precision and reliability of the KI lines for Cre/Flp intersection/subtraction, which can never be achieved by the transgenic approach. In fact, *Fezf2*-Cre Tg lines have been generated but are clearly non-specific and misleading (Guo et al., 2013).

2. It is unclear how the described driver lines for IT, PT, and CT neurons fit with our current understanding of known projection subtypes. A previous study has shown that in the motor cortex the PT neurons can be roughly divided into two anatomical and genetic subtypes: medulla and thalamic projecting PT neurons (Economo et al. 2018). It would be very informative to know what proportion of cells of the *Fezf2*-CreER line are labeled by injections into the thalamus and/or medulla. Single cell reconstruction data has suggested that the actual diversity of PT cells is likely to be even larger (Winnubst et al. 2019). Can the authors provide any evidence that the *Adcyap1*, *Tcerg11*, *Sema3a* lines capture different parts of this diversity?

- These questions also have been raised by Revs 1 & 2, and have been addressed before. Briefly, *Fezf2* captures >95% of corticospinal neurons (now in Fig. 5a-d). To investigate the question of laminar specificity for *Fezf2*⁺ projection types, we examined the distributions of *Fezf2*⁺ PyNs projecting to thalamus versus medulla have been quantified in motor areas, and our results are in line with the results of Economo et al., (2018); sublaminar distinctions that were not present in SSp-bfd were evident in motor cortex (Fig. 5e, f). The proportions of thalamus and medullary projection types within the overall *Fezf2*

population were determined (Fig. 5e, Extended Data Fig. 8c). These proportions were estimated by comparing the number of Fezf2 projection defined cells in a coronal columnar field of view of motor cortex to the overall number of recombined cells in the same region of a Fezf2-CreER;Ai14 (TM induction P21, P28) brain. These results indicated that 27% and 16% of the total PyN^{Fezf2} population were thalamus- and medulla- projecting, respectively. In SSp-bfd, the Fezf2+ thalamus- and medulla- projecting PyN subsets were not clearly distinguishable in terms of sublaminar specificity (Fig. 5g, h).

At E17.5, *Adcyap1* is a molecular target of the Fezf2 TF; we have therefore induced *Adcyap1*-CreER at E17.5 targeting the L5b subset of Fezf2+ neurons (Arlotta et al., 2005; Lodato et al., 2014).

Preliminary dual FISH experiments from our lab show ~40% Fezf2 PyNs express *Sema3E* and ~40% Fezf2 PyNs express *Tcerg1l*. The potential significance of *Tcerg1l* is discussed in response to Rev1 and in Rebuttal Fig. 2.

Overall, the significance of these driver lines is that they provide crucial tools for the large scale single cell reconstruction project by the BRAIN Initiative Cell Census Network (BICCN) (Peng et al., 2020 bioRxiv), which is underway to tackle this important problem of anatomic diversity (in IT, PT, CT neurons) in coming years.

3. Similarly, L6 CT projecting neurons in the barrel cortex are often separated based on their projections to primary or non-primary sensory nuclei. According to this scheme, primary projecting cells are located in upper L6, have dendrites terminating in L4, and project mostly to the VPM and reticular nucleus; non-primary cells are located deeper in L6, have dendrites terminating in L5, and project mostly to POM as well as VPM but not the reticular nucleus (Zhang and Deschênes 1997; Thomson 2010). The authors describe that “Anterograde tracing reveals that PyNsTle4 in S1 project specifically to first order thalamic nucleus VPM”. Because of the previously described literature it’s quite surprising that these neurons are located throughout L6 and have dendrites in L4 and 5. Even more surprising is that in Extended Data Figure 5f it seems that cells in deeper L6 are labeled by the retrograde injection in VPM. Can the authors comment on this? The authors should be better able to identify these subtypes based on a retrograde injection in the reticular nucleus. Do the *Tbr1*, *Tle4*, *Foxp2* label both types equally or is there a bias? Finally, in line 373 and 413-415, why is it specifically the retrograde label that confirms the two populations of cells with dendrites in L4 and L1? Both cell types are labeled and it’s unclear what the retrograde injection adds here. I found this quite confusing because I initially assumed that this meant only L1 neurons projected to VPM.

- We have investigated the relationship between *Tbr1*, *Tle4*, and *Foxp2* more carefully and presented the analysis in Fig. 4h, Extended Data Figs. 6, 7h-j, 8i. Among these three lines, *Tbr1* is the most inclusive of CT cells, followed by *Foxp2*, and then *Tle4*, which appear to represent subsets. Di Bella et al., 2020 has used scRNAseq to characterize the significance of *Tbr1*, *Tle4* and *Foxp2* expression in determining cell identity. They ranked *Foxp2* and *Tbr1* at similar levels in determining CT fate, *Tle4* being predicted to be involved in specifying near-projection L6 as well as CT neurons.

- CT cells with L1 versus L4 dendrite and projection to VPM: *Tle4* PyNs with dendrites in L4 or L1 were both observed from retrograde injections in VPM. Both types were corticothalamic. This result is presented in Ext data Fig 8i. The apparent “inconsistency” with results of (Zhang and Deschênes 1997; Thomson 2010) could in part be due to incomplete sampling and/or species differences (rat versus mouse).

The more detailed characterization of projection patterns in different cortical areas using these lines are beyond the scope of the current paper. Although retrograde labeling from reticular thalamus could be informative, it is unlikely to solve the issue of CT projection types, as these CT cells usually have other

collaterals. A definitive solution of these issues requires large-scale single cell reconstruction. Again, we provided *Tbr1*- and *Tle4*- CreER inducible lines to the BICCN large-scale single cell reconstruction project to tackle this question systematically across cortical areas.

4. An extension of Extended Data Figure 4 to include more of the reported cell lines would be very helpful for getting an insight into the identity of the targeted cell populations. A similar analysis as in (c), but with a calculation of the fraction of antibody labeled cells that are tdTom+ would also be informative.

- We have added multiple knock-in lines for the analysis presented in the new Ext Data Fig 5. These now include *Cux1*-CreER, *Lhx2*-CreER, *PlexinD1*-CreER, *Tbr2*-CreER (TM E16.5, TM E17.5), *Fezf2*-CreER, *Tcerg1*-CreER, *Adcyap1*-CreER, *Tle4*-CreER and *Tbr1*-CreER.

- Please see Rebuttal Figure 4 below with the percentage of marker-labeled cells that are tdTom+ for different lines we have analyzed in Extended Data Fig. 5. The number of cells labeled in each driver line depends on the TM dose administered and the litter size etc.

Rebuttal Figure 4

5. In the *Tis21*-CreER;*Fezf2*-Flp long-duration experiments, isn't it also possible that Gfp-only cells were *Fezf2* negative at one point but the RFP protein has since degraded? Are there therefore three distinct groups or could there be a gradient? Also, why does it say on line 222 that RFP-only cells likely consisted of PT cells? I assume it can only be said that these cells probably originated from nRGs that never expressed *Fezf2*.

- No. In the IS reporter, once RFP is activated by CreER, it is driven by the constitutive and strong *Rosa26* promoter; thus RFP expression will remain on in the cell.

Regarding line 222, yes Rev3 is correct. We have changed this to "PyN" instead of "PT cells".

6. Can the authors say anything about how precise the tamoxifen induction window likely is during the embryonic stage? Are the short-duration pulse-chase experiments (sacrifice after 24 hours) directly relatable to the long-term expression observed in the adult or can the induction of expression take place over a longer timescale?

- This comment was also made by Rev2 and was addressed previously. In addition, we add two more sources of information here:

1. We can see RFP expression after 8 hrs post-induction in many of our lines (*FezF2-CreER*; *Tbr2-CreER*). Birthdating studies using BrdU and EdU shows it doesn't last more than 18 hrs.
2. For a single dose of tamoxifen, 4-24hrs has been shown to be the optimal time window of its action (Jahn et al., 2018: <https://www.nature.com/articles/s41598-018-24085-9>).

Minor Points

1. The figure panels are on many occasions cited out of order with some of the skipped panels being cited till much later in the text. I would suggest the authors consider reordering the figures to be more in line with the text.
 - We have tried to better correlate the order of text and figure, but this is not always perfect given the complexity of the figures and the word limit of the paper.
2. Please note that several figures lack scale bars (for example: Fig. 2, Fig 5f, most of Fig 6.)
 - We have added these.
3. The barrel cortex is sometimes referred to as SSp other times as SSp-bfd, according to the Allen atlas these abbreviations refer to more the general and specific primary sensory areas and should be used consistently.
 - we used SSp in a broader sense at times because we are not doing specific staining for SSp-bfd so it is sometimes hard to say we are in the exact location. We have now used SSp-bf more consistently.
4. Line 179-180. What characteristics are referred to here? the medial-lateral gradient? Because this was already mentioned in the previous sentence.
 - These characteristics refer to the apical process (arrows, Figure 2f,o) in RGs^{FezF2+} and RGs^{Lhx2+} as well as cell soma at the ventricular surface indicating dividing progenitors (arrowheads, Figure 2f,o). This has been explained in the corresponding legend for Figure 2.
5. Line 372. Reference to Extended Data Figure 5h-n does not seem relevant.
 - we have revised this reference according to the new Figures and Ext data figures.
6. Line 455. Contralateral striatum mentioned twice.
 - corrected.
7. Line 74 in extended data. '...'
 - corrected.
8. Figure 1
 - 1a. Order of blue, purple red cells doesn't seem to make sense compared to c.
 - Colors in a refer to broad areas and are therefore distinct from the colors in b-e. The color code in b includes blue, purple and red, and is matched to colors in c.
 - Why does only *Tbr2* have an associated embryonic age (>15)?
 - This was referring to embryonic tamoxifen induction after E15. We realize that it is not clear to readers. We have removed this label.
9. Figure 2
 - In many cases the zoomed-in panels don't seem to match the marked boxes (e.g. 2e. does not appear to have dense labeling near the ventricle wall).

The high magnification inset for Figure 2 embryonic analysis is not a maximum intensity projection so that the progenitors can be examined for their morphological features. Therefore, the high-mag inset does not contain all the cells seen in the low mag image. We have added a sentence in the legend for Figure 2 and Extended Data Figure 1 stating the same. Please see response to Rev1 (Minor Points, Fig 2; rebuttal Fig 3). We show the max intensity projection in high mag for previous Fig 2e (new Fig. 2d) with a side-by-side comparison of what we present in the figure. The max intensity projection shows dense labeling near the ventricular wall as well as the marginal zone as seen in the low mag image.

○ 2b. The dating with tamoxifen induction is not mentioned in this schematic.
- The induction time is indicated in the figure's image panels, as these vary depending on the experiment.

○ 2f,h,j Density of the reporters look very different (RGBow vs Ai14) making them hard to compare. Would it be to add a quantification of the layer distribution?
- We have added layer-wise quantification in Figure 2, including statistics.

○ 2i,o. Can't really see the PyNs clones generated from individual RGs. Separate inset would be useful (similar to Extended Data Figure 1) with an explanation in the legend why these must come from single RGs. Also why is the asterisk yellow?
- We have replaced 'clones' with 'clusters'. We have also removed the asterisk in the current Figure 2h since we elaborate on the clusters in Extended Data Fig 1c, indicating the proliferating potential of the *Lhx2*⁺ RGs.

10. Figure 3c. According to text *Adcyap* PyNs should be in upper L5B but in picture it looks like they are mostly in L5a.
- Our cell depth distribution measurements shows an overlap between *Adcyap1* cells and *Fezf2* cells in L5b, that is distinct from the distribution in L5a seen for *PlexinD1*.

11. Figure 4

○ 4c. Not very helpful since lines are so thin and many areas are zero for all. Better to group by larger anatomical areas. Text states 'faction' should be 'fraction'. Finally, the panel is referred to as 4b in the legend.

○ 4e. Are the cartoons for *Adcyap* and *Tcerg11* switched? Shouldn't *Tcerg11* have no VPM projections?
- Rev3 is correct. We have revised this schematic Figure 4g.

12. Figure 6. Usage of colors for induction days is slightly misleading since they are not linked to the expressed fluorophore.
- We have modified the color tone.

13. Table 2. Unclear what the meaning of +?, +-, and what the difference is between - and no entry?
- we've changed the whole table and included it in the new Ext data Fig 4c.

14. Extended Data Figure 1f. No quantification, going by eye seems to be roughly half/half glia cells and PyNs and more in layer 5? Doesn't seem to match text (170-171).
- We have included a quantification in Ext Fig 1f for PyN/Glia (40:60) and number of PyN distribution across all cortical layers (majority in L5a and L4).

15. Extended Data Figure 2a. Mislabeled insets.

- We have reordered and modified this figure as Extended Data Fig 3 in the current manuscript. We have ensured the insets are labeled appropriately.

16. Extended Data Figure 4.

- It's not clear to me why some apparently double labeled cells do not have arrows or tdTomato cells without labeling have no asterisks, are these supposed to be examples only? Why are there two different kinds of arrows?
- I would add somewhere that *ctip2* and *cux1* are used as markers for L5-6 and L2-4
- We have added arrowheads to indicate double labeled cells, and asterisks to show tdTomato cells without marker labeling.
- The legend for Extended Data Figure 5 (new version of Extended Data Figure 4) refers to *Ctip2*^{HIGH} and *Cux1* as markers for L5b and L2-4 respectively.

17. Extended Data Figure 6. Would be useful to see the quantification of the red cells as well for comparison

- We have added the quantification of the red cells as well for comparison, now in Ext Data Fig 8.

References

Economo, Michael N., Sarada Viswanathan, Bosiljka Tasic, Erhan Bas, Johan Winnubst, Vilas Menon, Lucas T. Graybuck, et al. 2018. "Distinct Descending Motor Cortex Pathways and Their Roles in Movement." *Nature* 563 (7729): 79–84.

Thomson, Alex M. 2010. "Neocortical Layer 6, a Review." *Frontiers in Neuroanatomy* 4 (March): 13.

Winnubst, Johan, Erhan Bas, Tiago A. Ferreira, Zhuhao Wu, Michael N. Economo, Patrick Edson, Ben J. Arthur, et al. 2019. "Reconstruction of 1,000 Projection Neurons Reveals New Cell Types and Organization of Long-Range Connectivity in the Mouse Brain." *Cell* 179 (1): 268–81.e13.

Zhang, Z. W., and M. Deschênes. 1997. "Intracortical Axonal Projections of Lamina VI Cells of the Primary Somatosensory Cortex in the Rat: A Single-Cell Labeling Study." *The Journal of Neuroscience: The Official Journal of the Society for Neuroscience* 17 (16): 6365–79.

Reviewer Reports on the First Revision:

Referee #1 (Remarks to the Author):

Comments on Matho et al.

Overall, I find the tools, approach, and lines presented will be useful. I appreciate the effort that has been put into characterizing them. Regarding the rebuttal letter, I was interested in this paper from the first read and intended my comments to be useful to the authors and not hinder their work. Since the rebuttal is 30 pages long, this is more work than the comments were intended to generate. I'm sure that has been a lot of work.

The main source of disagreement I have with this manuscript is the comparison between the utility of BAC transgenic mice versus the transcription factor based lines. There is a lot of text devoted to the problems with transgenics, which I do not always find equitable both because you are short on space, as you argue in the rebuttal, and because many labs have found the transgenic lines effective. Your approach certainly has utility for many purposes, especially developmental ones, but it is also weaker in some respects when the cell type specific begins to be lost (for cortical circuits labs, the laminar specificity is important). For publication purposes, I would recommend focusing the text on the strengths of your approach. My earlier comment that it would be helpful to "address which lines/strategy are comparable or better than existing tools" I meant mostly in the spirit of aligning these as you do in tables for the new and existing lines by layer and cell type (such as IT, CT, PT, etc.).

Overall, I think the lines are useful and should be published. If not possible here due to space constraints than I would also support it at another high profile place where the full story is told.

Major Comments:

(1) Compare the access to PT-type neurons in Fezf2 to that in Sim1-Cre. By the criteria of specificity, it seems Sim1 is more specific for PT type neurons than Fezf2 in the following ways: (a) the laminar distribution of PT neurons is relatively restricted/crisp edges in Sim1 compared to Fezf2. See the images in Gerfen lab's work from Neuron 2013 (<https://pubmed.ncbi.nlm.nih.gov/24360541/>), Figure 3 and well as in his later Nat Comms 2018 (<https://pubmed.ncbi.nlm.nih.gov/30177709/>), Figure 1. Compare this laminar restriction to that possible with Fezf2 (Extended Data Fig 4d), which is much more broad. (b) The projection targets of PT-type neurons in Sim1 seem much more restricted to what is traditionally PT compared to the mixture of CT/IT/PT in Fezf2. All cortical and striatal projections are ipsilateral in Sim1 in contrast to Fezf2. Fezf2 is given as an example to support the "precision and reliability of the KI approach" that "allows us to achieve high specificity for targeting marker-defined subpopulations and completeness for capturing most if not all cells in each subpopulation across cortical areas" is Fezf2. But this line includes many non-L5B neurons in L6. And it shows contralateral cortical projections (Fig 4g) which are not typically major features of either L5B PT-type neurons or L6 CT type cells. (The extended figure is titled "A subset of PyNs(Fezf2) manifest IT features"). I don't think this fault should be a barrier to publishing the paper. But it does suggest that specificity is hard to achieve even with this transcription factor-based strategy. This problem with specificity is present in PlexinD1, but dealt with effectively by timing of TM injection: "In the neocortex, L5A and L2/3 IT PyNs were labeled (Fig. 3b, Extended Data Fig. 4a, d)". Since I could not find a library of images published of the PT projections targets throughout the brain to clarify what I mean by "All cortical and striatal projections are ipsilateral in Sim1", perhaps see the images in Gerfen lab database at (<http://gerfenc.biolumida.net/images/?page=images&selectionType=collection&selectionId=32>) After exploring this in more detail and looking at the images provided for Sim1 in Extended Data Fig. 4, it seems as though we are discussing different transgenic mice – the published work shows robust label throughout L5B in MOp and SSp; the Ed Fig 4. images suggest there are no neurons in

SSp and very sparse label in MOp! This near absence of label might be why it is not possible to provide a map of Sim1 coverage as done in the other lines in the same figure. You assert that "For example, while [Sim1-Cre] labels a set of PT neurons in numerous frontal cortical areas, it labels much less well in more posterior areas (e.g. Somatosensory areas, auditory, visual cortex)." This is different than my experience. Something is wrong between the lines used in this paper and what is already in the literature, which might account for the disagreement. I would recommend assessing what is the difference between the published results and these new ones before devoting a lot of text to the weakness of this tool that other groups find useful.

(2) Contrasting PlexD1 with other lines, I do think there will be interest in this approach as well as Cux 1. But I don't think there is a need to downplay the strengths of the existing transgenic lines in order to play up the strengths of your approach. In looking at Fig 3, comparing the IT type lines, you have some classes that label cells in layers 2-5A, but the Tlx3-Cre line gets many of the IT type cells across both L5A and L5B (which the coverage of IT-type cells here doesn't seem to include). (Gordon Shepherd – the one at Northwestern – has made great progress studying the differences between IT and PT across all of L5 so it is worth appreciating that there are ways to genetically access both of those cells.)

(3) In the specifics of the comparison you make, I am not persuaded some of the points made regarding the general weaknesses of the transgenic approach.

(a) When discussing the advantages of these novel lines with respect to existing lines (such as BAC transgenics), one disadvantage claimed is "A transgene expression pattern often has no relationship to the selected transgene itself." Is this necessarily a disadvantage? If any line expresses in the right cell type, and this is due to (intended or unintended) placement near an enhancer that is specific for that cell type, I don't find this an obstacle to its use. Many people are doing circuit tracing or silencing of given laminae and are using mouse lines that give them optogenetic or chemogenetic access to these. If these cells are accessible via differences in enhancers used by certain cell types and not others (assuming this is how transgenics that pass screening achieve such specificity) ... I don't see the error in exploiting that, and there is evidence to suggest different cell types use different enhancers.

(b) The heterogeneity between regions and layers seems present in both transgenic and transcription factor approaches. When I look at the images of the lines here, I mentally assess whether – in a coronal section – the same laminae are labelled moving from lateral to medial. For Fig 3A, look at Cux1 or Tbr2 or Tcerg11, for example; these hardly look uniform across cortical areas, and the gaps or absence in some areas suggests this is something more than just relative thickness of a given layer in a given region. I get this impression from the flat maps of Extended Data Fig 4d, in which almost all these approaches have problems in medial areas (cingulate) or lateral (entorhinal) ones. Maybe the problem is that there are molecular differences we don't fully appreciate between pyramidal neurons of the same (IT, PT, CT, etc) type moving from primary sensory cortices to medial wall areas. The heterogeneity between layers (or lines that label a mix of CT/PT/IT) suggests to me that defining some cell types by transcription factors is not so simple and that we still don't fully understand it.

(c) This is one reason alternative methods, perhaps combined with your strategy here may be needed and I am hopeful they will be fruitful. ("Recent advances in enhancer-based usage of AAV vectors show significant promise in achieving subpopulation restriction. The intersection of enhancer AAVs with strategically designed driver lines may substantially increase the specificity, ease, and throughput of neuronal cell type access.")

(d) Given the emphasis placed on developmental experiments in your rebuttal, I imagine these categories of questions are more aligned to your interests. Perhaps our difference here is interest as someone cracking circuits trying to label cell types versus understanding the developmental biology/regulation of expression of each cell type and you are certainly right to emphasize this difference.

(4) Some of the arguments explaining the lack of specificity I did not find persuasive. How can we distinguish whether this means the marker is imperfect, or that this represents some interesting

shared developmental trait (the asserted “likely confers certain fundamentally common features of this population that are yet to be dissected and understood”)? Further, “One possibility is that TF expressions are a bit “sloppy” and “leak” to other subclasses, an inconvenience we have to live with. Another more interesting and intriguing possibility is that these results may be revealing novel biology. Given that Fezf2 and Tbr1 are crucial fate restricting TFs likely regulate a set of associated target genes, and ectopic Fezf2 expression results in reprogramming of cell fate (Rouaux and Arlotta, 2013), it seems highly unlikely their expression can afford to be sloppy and leaky. These results raise the intriguing possibility that the Fezf2- and Tbr1- expressing “IT-like” cells might represent previously unrecognized novel or “intermediate” PyN types”. Perhaps this will become clear with time (and I am in no way requesting additional data). But this argument to me is problematic because it suggests if the expression seems to lack specificity ... then it’s not a problem because it’s actually meaningful (without yet understanding how it is meaningful!). This line of reasoning has a heads-I-win, tails-you-lose feel to me. The way I would put a positive spin on the IT/CT/PT character of Fezf2 would be as follows: “An interesting and intriguing possibility is that these Fezf2+ “IT” and “CT” cells might represent previously unrecognized novel or “intermediate” cell types.” I think what you are arguing is that: the existing grouping of pyramidal neurons in IT, PT, and CT classes does NOT necessarily mean that each of these three is its own developmental lineage, and it is possible for cells in a given lineage to become something different (e.g. there is an IT subset in a majority PT lineage and so forth). Detailing this is beyond the scope of this paper, could make an interesting future paper, but doesn’t yet undermine the general idea that IT type are IT type (project to a subset of ipsi and contra cortex and striatum but NOT thalamus and brainstem) and PT type are PT type (project only to a subset of ipsi cortex and striatum, as well as thalamus and brainstem). Similarly, “The variation in density across areas likely reflects the endogenous variation of the corresponding cell types. This biological variability/pattern is fundamentally different from “artificial variability” in Tg driver lines, which results from unexplainable arbitrary ectopic interactions between transgene promoter and genome regulatory elements near the transgene.” I guess I just don’t get this argument. There are differences across cortex (corticocortical cells project to different targets) and likely the differences in expression of both approaches are capturing some aspect of that.

(5) I understand why you might be tempted to make the point: “the severe word limit of this paper does not permit us to further elaborate several conceptual and strategic points, which are perhaps better suited for a review paper later” (as all of us write papers and then need to cut). But the main points need to be conveyed concisely (which are constraints we all operate with), and this argument is not useful and begs the response that the manuscript would be more suited to a longer format (especially if you think it is needed to convey the ideas).

(6) For the trackability of datasets and viewing individual brains, I provide this info for feedback on your site, not as a review of the paper. I viewed some of the brains at the Cell Projection site (<http://brainarchitecture.org/cell-type/projection>). These images are great. When I use the Image ID from Extended Data Table 5, it is possible to search and find brains. Depending on the goal (e.g. do you want users to be able to come and search for injections/visualize genetically defined neurons labeled from injection in a given site; I think it is unlikely many people will search by brain ID number), it would be nice to be able to search by injection site, though perhaps this is a wish list since it is not yet populated with huge numbers of injections. The contrast control is not easy to use (I followed the instructions attached) and, while 180406 and 170519 looked great, 180807 looked like it had substantial autofluorescence that made interpreting the injection difficult. 170506 did not have images posterior to olfactory bulb. These were PlexinD1-CreER mice. For Fezf2-CreER mice, the images for 180806 have some contralateral striatal and contralateral cortical projections (clearly not just PT). As for the other site, <http://brainarchitecture.org/cell-type/density>, there are not many cases populating it yet and those that I found are interneuron cases. Again, the images are great (and if I neglect to mention elsewhere, as are the images and other graphics in this manuscript).

Overall, it is not as user-friendly as some other sites, and that unfortunately is a big hindrance.

Your anatomy is beautiful as are other prior image datasets at brainarchitecture.org, and they are begging to be seen.

Minor points:

*Fig 3b1 illustrating the laminar position of different IT cells. I think this panel is meant to emphasize the L5a concentration in PlexinD1, but because of differences in scale/alignment of the image, the L5a in this panel appears much closer to the pia than in the adjacent Fig 3c1 and d1? Similar point for 3a1.

*Thanks for the figure addressing Areal Expression. It is pretty and a great way to convey this point. "Our KI lines are distributed more homogeneously across cortical areas (Extended Data Fig 4b)." The arrowheads in it aren't defined in the legend (I assume pointing out where some of the lines fall short). But not clear why the problems in midline/cingulate expression in all the mice where this is imperfect are not pointed out (Cux1, etc.). Also, having Sim1-Cre included here (it is in the images of ED Fig 4a but not quantified in 4b) would be nice for comparison.

*What's left of the T2A sequence on the endogenous transcription factor after cleavage and does it do anything different than the endogenous protein?

*In most figures and text you use PT, but ET is used in Fig. 2a

*The arrows in Fig 3e2, f2, g2, ... j2 are labeling axons I imagine but in context seem confusing (e.g. the axons comprise so much of the image, what's it pointing out in 3h2 for example).

*Line 301 Re: Cux1 and Cux2, "most upper layer neurons are thought to co-express these two proteins" but then later it seems your work suggests these can't perfectly overlap since Cux1 cells differ in some ways, including never projecting to striatum (so why suggest this here?).

*Line 319 This is nice, and inducing is easy enough for me to understand and use. "In our Tbr2-CreER driver targeting IPs (Fig. 2q), TM induction at E16.5 and E17.5 specifically labeled PyNs L2/3 and L2, respectively (Fig. 3c, d). Combined with the CreER→Flp conversion strategy that converts transient lineage and birth timing signals to permanent Flp expression, this approach enables specific AAV manipulation of L2 and L3 IT neurons."

*I'm assuming but not sure that the strategy in Fig 4a-4c is described in 4d ("Cell type specific anterograde tracing using a CreER→Flp conversion strategy (Fig. 4d) and Flp340 activated AAV-fDIO-EGFP from whisker somatosensory barrel cortex (SSp-bfd) reveals that 341 PyNsPlexinD1 project to ipsi- and contra-lateral cortical and striatal regions (Fig. 4a ... "). Assuming yes, since the age/date of tamoxifen administration is relevant to the Flp+ population, would it be possible to append "TM, PXX" or some label at the bottom of the injection site row to make it easier to ID when to give TM if desired to replicate the pattern shown.

*I am used to the plots of Fig 4e from reading some similar presentations. The cartoons in 4f-h are more accessible in part because they label fewer areas and these are labeled (as opposed to 4e where space prohibits). They make this a really attractive figure! Because it is complicated to assess where "all" the axons go, I am curious what threshold or method was used to determine what goes in the cartoon (which projection is big enough or bright enough). I looked in methods and supplemental methods but maybe I missed it. (This is meant as a minor comment – I am just curious how it was done, not asking for any changes.)

*Do Fezf2+ axons target VPM, or just POM? The axons shown in 3e2 seem to be relatively strong in POM and perhaps absent in VPM. This seems most consistent with the L5B collaterals mainly targeting higher order thalamic relays (the idea I get from Sherman and Guillery review). But (although the image is smaller, so harder to assess), in 4b for Fezf2 they do seem to target VPM and POM.

*Why is primary motor MOp most places but only "MO" in 5e and 5f?

*Re: electrophysiology and sag ratio measurement (Fig 5l-5m), would it be better to measure sag starting from the same resting membrane potential? It looks like the cells have different V_{rest} (5m) but just not injecting current and letting the I_h be measured from the cell's own resting potential means the I_h channels may start at different states. (This isn't really needed for the paper.)

* For the PlexD1 mice, "constitutive PlxnD1-Flp allele marks the whole population (Fig. 6e)" but the images in Fig 6e seem to show many more cells in L4/L5a than in upper layers. In contrast, label is dense the TM experiments in L2/3 in 6c and 6d, but relatively less in the TM experiment in 6b. Why the difference?

* "Further, as each driver line captures most if not all cells in each marker-defined subpopulation across cortical areas largely identical to mRNA in situ patterns, they enable investigators to study the global cortical areal network with cell type resolution." Won't global studies be hindered in part by the non-specific components of the expression (e.g. CT and IT cells in Fezf2, or mixture of L2-5 in PlexD1)? Because each of these lines has a complex expression pattern that seems to label multiple cells (not the atomic cell type specificity that might be desired), is does not seem consistent with the "carving nature at its joints" language used – seems more to me that we still don't have the full picture of how these cell types come to be in that expression of the same transcription factor might lead to IT, CT, or PT type cells depending on other factors (and so the organization of cell types might not be as perfectly hierarchical as I infer from this language – this is emphasized by the difficulty in mapping the cells defined by these lines to the transcriptomic cell types of other studies). Certainly the temporal control enabled by the CreER strategy does help, especially when the same factor is active at multiple points developmentally.

Referee #2 (Remarks to the Author):

The authors have addressed most of the major points raised in the first round. This is an exhaustive piece of work that will provide new and useful tools to the neuroscience community.

Referee #3 (Remarks to the Author):

The authors have made substantial changes to the manuscript in order to address the comments made by me and the other reviewers. Especially the description of the projection patterns of the driver lines has significantly improved and, together with the additional quantification and analysis throughout the manuscript, now give a clearer overview of the specificity and overlap of the toolkit. However, I fear the lack of new biological insights means that its appeal to a wider neuroscience community is still missing. I have laid out my broader reasons for this concern below. Finally, I conclude with a few specific comments to some of the newly presented data.

First, is this really a systematic framework? The authors mention that unlike transgenes "KI lines precisely recapitulate the endogenous spatiotemporal and cellular gene expression patterns'. However, they also say that it now appears that cell types are more fluid and cant be linked to single genes, and I would agree. Why then are these few gene driver lines specifically so crucial for a systematic investigation of cell types? Without these driver lines being linked to available gene expression data or some overarching developmental-based strategy it seems unlikely to serve as a systematic experimental framework, especially for labs that do not have the expertise in the utilized genetic tools and techniques.

It seems that a main benefit of these lines is their 'comprehensive coverage' i.e. they target cells in large portions of the brain. Labeling more specific projection classes though (or the 'specificity' of the toolkit) still requires additional retrograde/anterograde injections which induces its own experimental variability and tropisms on top of the (as the authors mention) inherent variability of the tamoxifen induction. I would disagree with the authors that this toolkit is therefore "essential" for further single cell analysis. There are existing ways to target the described cell populations in the manuscript which allow for the joined analysis of both anatomical and gene expression properties.

While the authors claim that constructing knock in lines "has the potential to capture and reveal inherent biological elements and processes." proof of this in the form of new biological insights are missing. Regarding this point, the first fate-mapping section of the paper goes in depth on the neurodevelopmental profile of Lhx2^{+/-} and Fezf2^{+/-} RGs. However, they don't report any difference in, for example, the projection patterns of these three populations in the later section (though the authors apparently have results on this?). The extensive RG profiling section therefore seems more likely to be of interest to neurodevelopmental researchers and not to the wider neuroscience/biology community. I would like to echo here also the point brought up by the other reviewers: constructing intersection/subtraction lines to reveal such interactions and effects can only be performed by highly specialized labs and is therefore likely not of interest to a large audience.

Specific comments:

The authors describe a novel PyNs(Fezf2/PlxnD1) population at the border of L5A/L5B, which is a very interesting finding. However, they then go on to speculate that these cells represent "intermediate PT-IT hybrids" and that "single cell reconstruction may reveal whether PyNs(Fezf2/PlxnD1) are typical IT cells or also project subcortically" (later also mentioned in the discussion). I think this is a fairly remarkable claim and such significant speculation requires more evidence. Considering what we know now, it seems more likely that this is a result of gene expression gradients along cortical layers combined with "fuzzy" borders between cell types. In this light it can also be seen as evidence of a lack of 'specificity' for the driver line.

The authors claim that anterograde tracing revealed that the Cux1-CreER is the first cortex-restricted KI driver line. First, in Supp. movie 2 it seems there are in fact some axons in the striatum. Taking a closer look at the online publicly available data revealed this as well: <http://braincircuits.org/viewer4/mouse/map/29117F>. Are these minimal axons ignored in the analysis? Previous single cell research has shown that IT cells likely form a continuum in terms of their abundance of projections to the striatum. I therefore wonder if these merely fall at the lower end of this spectrum. Second, since systemic induction of the Cux1-CreER line causes labeling of medium spiny neurons in the striatum can the authors claim that the Cux1-CreER line will label cortex-restricted cells throughout the cortex? Or only in SSP? These points challenge the claim that Cux1-CreER is the first cortex-restricted KI driver line.

The interpretation of the new description of three RG populations (Lhx2⁺/Fezf2⁺, Lhx2⁺/Fezf2⁻, and Lhx2⁻/Fezf2⁺) is unclear to me. Is this the best description of the data or could this be the result of taking a 'snapshot' during one time-point in development while looking at the time-dependent gradient expression of two interacting genes. In other words, are these discrete groups or is there a decreasing expression of Lhx2 over time that normally suppresses Fezf2 so that the majority of RG identities progress as: Lhx2⁺/Fezf2⁻ -> Lhx2⁺/Fezf2⁺ -> Lhx2⁻/Fezf2⁺. Considering their difference in developmental onset isn't it expected then that Lhx2⁺/Fezf2⁻ is twice as abundant? Could repeating these measurements at multiple developmental time points answer this question? I admit that I have the feeling I am missing something important here while reading the text.

The description of the Lhx2 pulse-chase experiments seem at some places to be inconsistent with

the new quantification. "E12.5→E13.5 pulse-chase revealed a prominent medial-high to lateral-low gradient of RGs Lhx2 suggesting significant differentiation of the earlier, E10.5 RG pool." (Line 162). Looking at the data in figure 2J and 2K it seems that this gradient is already present at E10.5 so I don't understand how this can be used to support this claim. It is then stated that a similar gradient is present at E13.5 but at lower overall density (line 163). Yet the quantification shows no quantitative difference in the number of progenitors. It would be better to show these numbers as normalized by the investigated area.

Minor points:

Line 309-311: In Supp. movie 2 it seems there are in fact some axons in the striatum of (see also <http://braincircuits.org/viewer4/mouse/map/29117F> and others). Is this low quantity ignored in the analysis?

Line 846: "Tbr2-CreER induction at E16 and E17 label L2/3 and L3 PyNs, respectively." Shouldn't this be L2?

Line 272: "Tbr2-creER" should be "Tbr2-CreER"

Line 849: "cp, cerebral peduncle; P_{Om}, posterior medial nucleus of thalamus; VPM, ventral posteromedial nucleus of thalamus.". These don't seem to be used in the figure.

Figure 4B: what are these strange black circles in the striatum?

Figure 4E: Too much details in the areas being shown (321 areas!) so that the main message is lost. Authors should be more selective to highlight the important differences.

Extended data figure 3: "Mixed RFP/GFP clones are most prominent and likely result from Cre activation of RFP in nRGsFezf2- and subsequent Flp activation of GFP in Fezf2+ L5/6 postmitotic PyNs (i,j')." From these pictures it seems all cells (except maybe 2?) are either GFP or RFP only.

Author Rebuttals to First Revision:

Referees' comments:

Referee #1 (Remarks to the Author):

Overall, I find the tools, approach, and lines presented will be useful. I appreciate the effort that has been put into characterizing them. Regarding the rebuttal letter, I was interested in this paper from the first read and intended my comments to be useful to the authors and not hinder their work. Since the rebuttal is 30 pages long, this is more work than the comments were intended to generate. I'm sure that has been a lot of work.

The main source of disagreement I have with this manuscript is the comparison between the utility of BAC transgenic mice versus the transcription factor based lines. There is a lot of text devoted to the problems with transgenics, which I do not always find equitable both because you are short on space, as you argue in the rebuttal, and because many labs have found the transgenic lines effective. Your approach certainly has utility for many purposes, especially developmental ones, but it is also weaker in some respects when the cell type specific begins to be lost (for cortical circuits labs, the laminar specificity is important). For publication purposes, I would recommend focusing the text on the strengths of your approach. My earlier comment that it would be helpful to "address which lines/strategy are comparable or better than existing tools" I meant mostly in the spirit of aligning these as you do in tables for the new and existing lines by layer and cell type (such as IT, CT, PT, etc.).

Overall, I think the lines are useful and should be published. If not possible here due to space constraints than I would also support it at another high profile place where the full story is told.

- We thank Rev1 for raising this good point and for the excellent suggestions. Overall we agree with Rev1 on the practical value of validated transgenic lines for cell type access. We have removed most if not all of the critiques on transgenic lines. Following Rev1's suggestion, we focus instead on the strength of our approach. Indeed streamlining the text and the word limit also make these changes a necessity.

Major Comments:

(1) Compare the access to PT-type neurons in Fezf2 to that in Sim1-Cre. By the criteria of specificity, it seems Sim1 is more specific for PT type neurons than Fezf2 in the following ways: (a) the laminar distribution of PT neurons is relatively restricted/crisp edges in Sim1 compared to Fezf2. See the images in Gerfen lab's work from Neuron 2013 (<https://pubmed.ncbi.nlm.nih.gov/24360541/>), Figure 3 and well as in his later Nat Comms 2018 (<https://pubmed.ncbi.nlm.nih.gov/30177709/>), Figure 1. Compare this laminar restriction to that possible with Fezf2 (Extended Data Fig 4d), which is much more broad. (b) The projection targets of PT-type neurons in Sim1 seem much more restricted to what is traditionally PT compared to the mixture of CT/IT/PT in Fezf2. All cortical and striatal projections are ipsilateral in Sim1 in

contrast to Fezf2. Fezf2 is given as an example to support the “precision and reliability of the KI approach” that “allows us to achieve high specificity for targeting marker-defined subpopulations and completeness for capturing most if not all cells in each subpopulation across cortical areas” is Fezf2. But this line includes many non-L5B neurons in L6. And it shows contralateral cortical projections (Fig 4g) which are not typically major features of either L5B PT-type neurons or L6 CT type cells. (The extended figure is titled “A subset of PyNs(Fezf2) manifest IT features”). I don’t think this fault should be a barrier to publishing the paper. But it does suggest that specificity is hard to achieve even with this transcription factor-based strategy. This problem with specificity is present in PlexinD1, but dealt with effectively by timing of TM injection: “In the neocortex, L5A and L2/3 IT PyNs were labeled (Fig. 3b, Extended Data Fig. 4a, d)”.

Since I could not find a library of images published of the PT projections targets throughout the brain to clarify what I mean by “All cortical and striatal projections are ipsilateral in Sim1”, perhaps see the images in Gerfen lab database at (<http://gerfenc.biolumida.net/images/?page=images&selectionType=collection&selectionId=32>)

After exploring this in more detail and looking at the images provided for Sim1 in Extended Data Fig. 4, it seems as though we are discussing different transgenic mice – the published work shows robust label throughout L5B in MOp and SSp; the Ed Fig 4. images suggest there are no neurons in SSp and very sparse label in MOp! This near absence of label might be why it is not possible to provide a map of Sim1 coverage as done in the other lines in the same figure. You assert that “For example, while [Sim1-Cre] labels a set of PT neurons in numerous frontal cortical areas, it labels much less well in more posterior areas (e.g. Somatosensory areas, auditory, visual cortex).” This is different than my experience. Something is wrong between the lines used in this paper and what is already in the literature, which might account for the disagreement. I would recommend assessing what is the difference between the published results and these new ones before devoting a lot of text to the weakness of this tool that other groups find useful.

- We agree with Rev1 on the complementary aspects of transgenic versus knockin approaches; indeed the comparison between Tg Sim1-Cre and KI Fezf2-2A-CreER is a good example. We have removed the critique on Sim1 and focused on the strength of our approach. We also state the complementary nature of existing and new tools (first sentence in Discussion).

We do hope to clarify for Rev1 on the following points:

- We have removed the phrases: “... allows us to achieve high specificity for targeting marker-defined subpopulations and completeness for capturing most if not all cells in each subpopulation across cortical areas”. What we meant was that the precision and reliability of the KI approach allows us to *specifically and completely* capture cells expressing the *endogenous gene*. This is critical because even if the captured cells may still comprise multiple types (e.g. Fezf2), it provides a solid entry point to dissect those more specific laminar and projection types. The other important point is that for major transcription factors such as Fezf2, the multiple types that it expresses in are likely to have certain mechanistic or even functional relationships. This statement is based on our substantial results from optogenetic motor mapping using our Fezf2 (and other) driver lines, which

generate surprisingly complex and smooth (i.e. natural looking) forelimb and orofacial movements compared to similar studies using other Tg driver lines (e.g. Thy1). Thus dissecting these TF-related “composite cell types” may reveal important biological insights. Although the “composite cell types” captured by some KI lines may seem, for the first pass, less convenient and not immediately satisfying for circuit neuroscientists, we believe these provide a logical and clear path toward a biology-based systematic dissection of the hierarchical organization of neuron types.

- In a complementary way, certain Tg lines such as Sim1-Cre may hit quite a specific cell type, and we fully embrace it and should take full advantage of these by all means. The main problem is that the chances are rather low and this is why useful Tg lines are so few despite the large-scale GENSAT project. Most labs no longer use this approach to our knowledge. More specifically, the Sim1 image in our Extended Data Fig. 4 is provided by Julie Harris at the Allen Institute and is the same line as published in Gerfen 2013 Neuron paper. Georg Keller from FMI Basel also told us that he saw reduced expression in more posterior cortical areas compared to frontal areas in this line. It is not clear what is the source of these discrepancies, and whether the genetic background might play a role (which is more common for Tg lines).

In summary, to achieve more cell type targeting, intersection based on KI lines is a logical path forward, although it does require more work than a “simple” driver line. We are also developing entirely new technologies to overcome some fundamental limitations of germline and DNA engineering.

(2) Contrasting PlexD1 with other lines, I do think there will be interest in this approach as well as Cux 1. But I don't think there is a need to downplay the strengths of the existing transgenic lines in order to play up the strengths of your approach. In looking at Fig 3, comparing the IT type lines, you have some classes that label cells in layers 2-5A, but the Tlx3-Cre line gets many of the IT type cells across both L5A and L5B (which the coverage of IT-type cells here doesn't seem to include). (Gordon Shepherd – the one at Northwestern – has made great progress studying the differences between IT and PT across all of L5 so it is worth appreciating that there are ways to genetically access both of those cells.)

- We agree with Rev1 and have removed our critique of other transgenic IT driver lines.

(3) In the specifics of the comparison you make, I am not persuaded some of the points made regarding the general weaknesses of the transgenic approach.

(a) When discussing the advantages of these novel lines with respect to existing lines (such as BAC transgenics), one disadvantage claimed is “A transgene expression pattern often has no relationship to the selected transgene itself.” Is this necessarily a disadvantage? If any line expresses in the right cell type, and this is due to (intended or unintended) placement near an enhancer that is specific for that cell type, I don't find this an obstacle to its use. Many people are doing circuit tracing or silencing of given laminae and are using mouse lines that give them optogenetic or chemogenetic access to these. If these cells are accessible via differences in enhancers used by certain cell types and not others (assuming this is how transgenics that pass

screening achieve such specificity) ... I don't see the error in exploiting that, and there is evidence to suggest different cell types use different enhancers.

- We agree. We have removed almost all critiques on transgenic lines in Results and Discussion.

(b) The heterogeneity between regions and layers seems present in both transgenic and transcription factor approaches. When I look at the images of the lines here, I mentally assess whether – in a coronal section – the same laminae are labelled moving from lateral to medial. For Fig 3A, look at *Cux1* or *Tbr2* or *Tcerg11*, for example; these hardly look uniform across cortical areas, and the gaps or absence in some areas suggests this is something more than just relative thickness of a given layer in a given region. I get this impression from the flat maps of Extended Data Fig 4d, in which almost all these approaches have problems in medial areas (cingulate) or lateral (entorhinal) ones. Maybe the problem is that there are molecular differences we don't fully appreciate between pyramidal neurons of the same (IT, PT, CT, etc) type moving from primary sensory cortices to medial wall areas. The heterogeneity between layers (or lines that label a mix of CT/PT/IT) suggests to me that defining some cell types by transcription factors is not so simple and that we still don't fully understand it.

- Rev1 is correct in noting that there are areal and laminar variations (“heterogeneity”) of cell labeling in both transgenic and knockin drivers. Due to word limit, we have to pretty much cut the statement and discussion on this issue in our revised manuscript.

To clarify, the main difference between Tg vs KI line in terms of their areal and laminar variations is that the former is largely unexplainable while the latter precisely reflects the expression pattern of the endogenous gene. Therefore, the areal differences revealed by *Fezf2*, *PlexinD1*, *Tle4*, *Tbr1*, *Cux1* drivers may reflect biologically relevant areal differences and indeed have been used by the Allen Institute to delineate cortical areas. Thus the absence or reduction of cells in certain areas in these KI lines might suggest a biological difference in cell type composition, a testable hypothesis given the reliability of these lines. *Tcerg11* is not a good example here as it is meant to capture a currently less well defined rare PT population.

(c) This is one reason alternative methods, perhaps combined with your strategy here may be needed and I am hopeful they will be fruitful. (“Recent advances in enhancer-based usage of AAV vectors show significant promise in achieving subpopulation restriction. The intersection of enhancer AAVs with strategically designed driver lines may substantially increase the specificity, ease, and throughput of neuronal cell type access.”)

- We are working on the enhancer approach and on other technologies.

(d) Given the emphasis placed on developmental experiments in your rebuttal, I imagine these categories of questions are more aligned to your interests. Perhaps our difference here is interest as someone cracking circuits trying to label cell types versus understanding the developmental biology/regulation of expression of each cell type and you are certainly right to emphasize this difference.

- Indeed there are differences between systems and developmental genetic studies in terms of goals and practical needs. We hope that methods based on inherent development and gene regulation will provide a more rational approach and systematic tools for circuit-cracking systems neuroscientists, but any useful tools obtained from other approaches should be embraced and taken full advantage of.

(4) Some of the arguments explaining the lack of specificity I did not find persuasive. How can we distinguish whether this means the marker is imperfect, or that this represents some interesting shared developmental trait (the asserted “likely confers certain fundamentally common features of this population that are yet to be dissected and understood”)? Further, “One possibility is that TF expressions are a bit “sloppy” and “leak” to other subclasses, an inconvenience we have to live with. Another more interesting and intriguing possibility is that these results may be revealing novel biology. Given that *Fezf2* and *Tbr1* are crucial fate restricting TFs likely regulate a set of associated target genes, and ectopic *Fezf2* expression results in reprogramming of cell fate (Rouaux and Arlotta, 2013), it seems highly unlikely their expression can afford to be sloppy and leaky. These results raise the intriguing possibility that the *Fezf2*- and *Tbr1*- expressing “IT-like” cells might represent previously unrecognized novel or “intermediate” PyN types”.

Perhaps this will become clear with time (and I am in no way requesting additional data). But this argument to me is problematic because it suggests if the expression seems to lack specificity ... then it’s not a problem because it’s actually meaningful (without yet understanding how it is meaningful!). This line of reasoning has a heads-I-win, tails-you-lose feel to me.

The way I would put a positive spin on the IT/CT/PT character of *Fezf2* would be as follows: “An interesting and intriguing possibility is that these *Fezf2*+ “IT” and “CT” cells might represent previously unrecognized novel or “intermediate” cell types.” I think what you are arguing is that: the existing grouping of pyramidal neurons in IT, PT, and CT classes does NOT necessarily mean that each of these three is its own developmental lineage, and it is possible for cells in a given lineage to become something different (e.g. there is an IT subset in a majority PT lineage and so forth). Detailing this is beyond the scope of this paper, could make an interesting future paper, but doesn’t yet undermine the general idea that IT type are IT type (project to a subset of ipsi and contra cortex and striatum but NOT thalamus and brainstem) and PT type are PT type (project only to a subset of ipsi cortex and striatum, as well as thalamus and brainstem).

Similarly, “The variation in density across areas likely reflects the endogenous variation of the corresponding cell types. This biological variability/pattern is fundamentally different from “artificial variability” in Tg driver lines, which results from unexplainable arbitrary ectopic interactions between transgene promoter and genome regulatory elements near the transgene.” I guess I just don’t get this argument. There are differences across cortex (corticocortical cells project to different targets) and likely the differences in expression of both approaches are capturing some aspect of that.

- Overall we agree with Rev1 on all the above discussions. We have removed most of the statements mentioned above. Indeed, our argument/ideas on the “composite cell types” captured

by KI lines in the previous rebuttal is more of a speculation at this point. This will be tested by future studies.

We do not need to make those statements in the current paper.

(5) I understand why you might be tempted to make the point: “the severe word limit of this paper does not permit us to further elaborate several conceptual and strategic points, which are perhaps better suited for a review paper later” (as all of us write papers and then need to cut). But the main points need to be conveyed concisely (which are constraints we all operate with), and this argument is not useful and begs the response that the manuscript would be more suited to a longer format (especially if you think it is needed to convey the ideas).

- We fully agree with Rev1. We have removed all interpretational/subjective statements to convey the key findings concisely in the second revision.

(6) For the trackability of datasets and viewing individual brains, I provide this info for feedback on your site, not as a review of the paper. I viewed some of the brains at the Cell Projection site (<http://brainarchitecture.org/cell-type/projection>). These images are great. When I use the Image ID from Extended Data Table 5, it is possible to search and find brains. Depending on the goal (e.g. do you want users to be able to come and search for injections/visualize genetically defined neurons labeled from injection in a given site; I think it is unlikely many people will search by brain ID number), it would be nice to be able to search by injection site, though perhaps this is a wish list since it is not yet populated with huge numbers of injections. The contrast control is not easy to use (I followed the instructions attached) and, while 180406 and 170519 looked great, 180807 looked like it had substantial autofluorescence that made interpreting the injection difficult. 170506

did not have images posterior to olfactory bulb. These were PlexinD1-CreER mice. For Fezf2-CreER mice, the images for 180806 have some contralateral striatal and contralateral cortical projections (clearly not just PT). As for the other site, <http://brainarchitecture.org/cell-type/density>, there are not many cases populating it yet and those that I found are interneuron cases. Again, the images are great (and if I neglect to mention elsewhere, as are the images and other graphics in this manuscript).

Overall, it is not as user-friendly as some other sites, and that unfortunately is a big hindrance. Your anatomy is beautiful as are other prior image datasets at brainarchitecture.org, and they are begging to be seen.

- We thank the Rev1 for supporting the usefulness of our data sharing portal, and have resolved the issues to the extent that the data is searchable and with more ease of viewing. Metadata is being updated to provide more user-friendly search features on the data portal. The data-viewing features have been corrected to ensure that all sections appear (beyond the olfactory bulb for 170506 and with a broader dynamic range to avoid the contrast problem for datasets like 180807). Partha Mitra’s team is continuing to improve this portal in the coming weeks.

Minor points:

*Fig 3b1 illustrating the laminar position of different IT cells. I think this panel is meant to emphasize the L5a concentration in PlexinD1, but because of differences in scale/alignment of the image, the L5a in this panel appears much closer to the pia than in the adjacent Fig 3c1 and d1? Similar point for 3a1.

- We have improved the alignment of the cortical panels in the new Figure 2 (formerly Fig. 3).

*Thanks for the figure addressing Areal Expression. It is pretty and a great way to convey this point. “Our KI lines are distributed more homogeneously across cortical areas (Extended Data Fig 4b).” The arrowheads in it aren’t defined in the legend (I assume pointing out where some of the lines fall short). But not clear why the problems in midline/cingulate expression in all the mice where this is imperfect are not pointed out (Cux1, etc.). Also, having Sim1-Cre included here (it is in the images of ED Fig 4a but not quantified in 4b) would be nice for comparison.

- Arrowheads are now defined in the legend.

Regarding the Sim1-Cre whole brain STP dataset for areal flat-mapping, the only dataset from the Allen Institute is what we showed in Ext Data Fig4. As Julie Harris at Allen Institute saw suboptimal areal expression pattern, they did not continue the imaging and analysis of this Tg driver line at Allen.

*What’s left of the T2A sequence on the endogenous transcription factor after cleavage and does it do anything different than the endogenous protein?

- Typically, addition of a few amino acids at the C-terminus of a protein, such as epitope tagging that is quite routinely done, seems to have a rather unnoticeable effect on its function. However, this is not rigorously tested in every case, and we cannot rule out minor effects. We can only say that most KI lines appear indistinguishable from wild-type littermates in many of the phenotypes we measured.

*In most figures and text you use PT, but ET is used in Fig. 2a

- We now use PT in the new Fig.1a (formerly Fig. 2a)

*The arrows in Fig3e2,f2,g2, ... j2 are labeling axons I imagine but in context seem confusing (e.g. the axons comprise so much of the image, what’s it pointing out in 3h2 for example).

- These arrows have been removed.

*Line 301 Re: Cux1 and Cux2, ” most upper layer neurons are thought to co-express these two proteins” but then later it seems your work suggests these can’t perfectly overlap since Cux1 cells differ in some ways, including never projecting to striatum (so why suggest this here?).

- We removed the Cux1 and Cux2 co-expression sentence.

*Line 319 This is nice, and inducing is easy enough for me to understand and use. “In our Tbr2-CreER driver targeting IPs (Fig. 2q), TM induction at E16.5 and E17.5 specifically labeled PyNs L2/3 and L2, respectively (Fig. 3c, d). Combined with the CreER→Flp conversion strategy that converts transient lineage and birth timing signals to permanent Flp expression, this approach enables specific AAV manipulation of L2 and L3 IT neurons.”

*I’m assuming but not sure that the strategy in Fig 4a-4c is described in 4d (“Cell type specific anterograde tracing using a CreER→ Flp conversion strategy (Fig. 4d) and Flp340 activated AAV-fDIO-EGFP from whisker somatosensory barrel cortex (SSp-bfd) reveals that 341 PyNsPlxnD1 project to ipsi- and contra-lateral cortical and striatal regions (Fig. 4a ... “). Assuming yes, since the age/date of tamoxifen administration is relevant to the Flp+ population, would it be possible to append “TM, PXX” or some label at the bottom of the injection site row to make it easier to ID when to give TM if desired to replicate the pattern shown.

- TM induction timepoints have been added to Figure 3 under each column (formerly Figure 4).

*I am used to the plots of Fig 4e from reading some similar presentations. The cartoons in 4f-h are more accessible in part because they label fewer areas and these are labeled (as opposed to 4e where space prohibits). They make this a really attractive figure! Because it is complicated to assess where “all” the axons go, I am curious what threshold or method was used to determine what goes in the cartoon (which projection is big enough or bright enough). I looked in methods and supplemental methods but maybe I missed it. (This is meant as a minor comment – I am just curious how it was done, not asking for any changes.)

- To generate cartoons of axon projections for a given driver line, axon detection outputs from all individual experiments were compared (sorting the values from high to low), and analyzed side-by-side with low resolution image stacks (and the CCFv3 registered to the low resolution dataset for brain area definition) to get a general picture of the injection, as well as high resolution images for specific brain areas. We have now included this description in the Methods under Whole-brain Serial Two Photon Tomography and Image Analysis.

*Do Fezf2+ axons target VPM, or just POM? The axons shown in 3e2 seem to be relatively strong in POM and perhaps absent in VPM. This seems most consistent with the L5B collaterals mainly targeting higher order thalamic relays (the idea I get from Sherman and Guillery review). But (although the image is smaller, so harder to assess), in 4b for Fezf2 they do seem to target VPM and POM.

-The seemingly weak labeling in VPM compared to POM in panel Fig.2e2 (formerly 3e2) is due to adjustments made to the dynamic range based on the strongest labeling: we recommend visiting our Brain Architecture data portal to view the images at high resolution with the option of adjusting

the range to a brain region of interest, such as VPM/POm (see for instance: <http://brainarchitecture.org/viewer4/mouse/map/29080F>).

-We think Fezf2 from SSp-bfd targets both POm and VPM. It could be that the VPM projection comes from the minor L6 Fezf2+PyN subset, as opposed to the L5B population, which would agree with the idea from Sherman and Guillery (eg 2011 J Neurophysiol). We have yet to determine whether the VPM-projecting Fezf2+PyNs constitute a subset of CT PyNs.

*Why is primary motor MOp most places but only “MO” in 5e and 5f?

This is a broader region that includes MOs as well as MOp, comparable to the anterior lateral motor cortex (ALM) examined in Economo et al, 2018. We provide the images from the cortical region in Extended Data Figure 9.

*Re: electrophysiology and sag ratio measurement (Fig 5l-5m), would it be better to measure sag starting from the same resting membrane potential? It looks like the cells have different V_{rest} (5m) but just not injecting current and letting the Ih be measured from the cell’s own resting potential means the Ih channels may start at different states. (This isn’t really needed for the paper.)

- We agree with the reviewer on this technical point. We have condensed the text and removed the reference to sag ratio and Ih measurement.

“Compared to PyNsFezf2+/PV-, PyNsFezf2+/PV+ exhibited more depolarized resting membrane potentials and a larger rebound activity after depolarization in cortical brain slices” (Fig. 4m).”

* For the PlexD1 mice, “constitutive PlxnD1-Flp allele marks the whole population (Fig. 6e)” but the images in Fig 6e seem to show many more cells in L4/L5a than in upper layers. In contrast, label is dense the TM experiments in L2/3 in 6c and 6d, but relatively less in the TM experiment in 6b. Why the difference?

- We changed this sentence to “constitutive PlxnD1-Flp allele marks the whole population as well as some blood vessel cells”

The PlexinD1 population captured by the constitutively expressed PlexinD1-Flp marks the whole population from L2 to L5a.

We would like to point out that the images acquired in Fig.5b-d (formerly Fig.6b-d) are maximum projections in the z-axis from confocal image stacks, whereas the PlexinD1-Flp;FSF-tdTomato image used for comparison (Fig. 5e, formerly Fig. 6e) is a single imaging plane from serial two photon tomography. This could explain the difference in density for upper layer PyNs labeled. We provide the image from PlexinD1-Flp;FSF-tdTomato for comparison, to support the idea that this driver provides coverage across lamina and to demonstrate that the intersection approach with Tbr2-CreER allows restriction from the broader population. We expect that imaging from sections with confocal microscopy would provide upper lamina density more consistent with the intersection images.

The variability of labeling density among Fig.5b-d (previous Fig.6b-d) likely results from some variability of TM induction time and dose.

* “Further, as each driver line captures most if not all cells in each marker-defined subpopulation across cortical areas largely identical to mRNA in situ patterns, they enable investigators to study the global cortical areal network with cell type resolution.” Won’t global studies be hindered in part by the non-specific components of the expression (e.g. CT and IT cells in Fezf2, or mixture of L2-5 in PlexD1)? Because each of these lines has a complex expression pattern that seems to label multiple cells (not the atomic cell type specificity that might be desired), is does not seem consistent with the “carving nature at its joints” language used – seems more to me that we still don’t have the full picture of how these cell types come to be in that expression of the same transcription factor might lead to IT, CT, or PT type cells depending on other factors (and so the organization of cell types might not be as perfectly hierarchical as I infer from this language –this is emphasized by the difficulty in mapping the cells defined by these lines to the transcriptomic cell types of other studies). Certainly the temporal control enabled by the CreER strategy does help, especially when the same factor is active at multiple points developmentally.

- We have removed the quoted sentence above.

Overall we agree with the assessment of Rev1 on the more complex relationship between single gene expression and projection/connectivity-defined neuron types. We do believe that transcription factor based KI drivers provide a framework for dissecting the hierarchical organization of neuron types, and in this sense allows “carving nature at its joints”, even though each individual driver line by itself does not reach the granularity of specific cell types. Intersectional strategies based on this framework are critical to access finer types.

Referee #2 (Remarks to the Author):

The authors have addressed most of the major points raised in the first round. This is an exhaustive piece of work that will provide new and useful tools to the neuroscience community.

Referee #3 (Remarks to the Author):

The authors have made substantial changes to the manuscript in order to address the comments made by me and the other reviewers. Especially the description of the projection patterns of the driver lines has significantly improved and, together with the additional quantification and analysis throughout the manuscript, now give a clearer overview of the specificity and overlap of the toolkit. However, I fear the lack of new biological insights means that its appeal to a wider

neuroscience community is still missing. I have laid out my broader reasons for this concern below. Finally, I conclude with a few specific comments to some of the newly presented data.

- The key advance of our study is to provide a toolkit, a strategic framework and a road map for genetic dissection of one of the most complex brain systems. These tools and strategies for cell type targeting and fate mapping will be of broad interest and impact to systems, developmental, and molecular neuroscientists working on the broadly defined cerebral hemisphere (cerebral cortex, hippocampus, basal lateral amygdala). This will enable many subsequent studies to achieve new biological insights.

First, is this really a systematic framework? The authors mention that unlike transgenes “KI lines precisely recapitulate the endogenous spatiotemporal and cellular gene expression patterns’. However, they also say that it now appears that cell types are more fluid and can’t be linked to single genes, and I would agree. Why then are these few gene driver lines specifically so crucial for a systematic investigation of cell types? Without these driver lines being linked to available gene expression data or some overarching developmental-based strategy it seems unlikely to serve as a systematic experimental framework, especially for labs that do not have the expertise in the utilized genetic tools and techniques.

- We respectfully disagree with the reviewer on this point. We believe our toolkit and strategy provide a framework for dissecting the hierarchical organization and developmental logic of pyramidal neurons for the following reasons. We selected key transcription factors and effector genes which are main players of the specification and differentiation programs of PyNs, from major progenitor types to major projection types. Although each individual driver line does not capture a “specific projection type”, because there is no simple relationship between single gene expression and “cell types”, we provide a clear strategy and a road map to achieve finer targeting based on these reliable driver lines. We further present intersectional strategies based on lineage, cell birth order, and anatomy. To our knowledge, there has been no other study that provides such a systematic toolkit and rational strategy. From our extensive characterization and usage, these lines in and of themselves (i.e. without intersection) are already extremely useful in studying cortical circuits by wide-field GCaMP imaging, optogenetic manipulations, and optogenetic tagging in single unit recordings, lineage tracing and fate mapping, targeted molecular genetic profiling etc. Multiple papers are shaping up in our lab using these tools from development to systems studies. It is very clear to us that these tools and strategies will have a wide impact.

As we expressed in our response to Rev1, these tools further provide a reliable and rational path to dissect and access finer-grained cell types by various combinatorial strategies that we presented. Although this is less convenient, it is what it takes to move forward (also see similar need for intersectional methods from the field of *Drosophila* neurobiology). We and others are developing enhancer tools and other new cell type technologies, but we envision that these foundational driver lines cannot be replaced, especially for developmental studies, and linking developmental to functional studies.

It seems that a main benefit of these lines is their ‘comprehensive coverage’ i.e. they target cells in large portions of the brain. Labeling more specific projection classes though (or the ‘specificity’ of the toolkit) still requires additional retrograde/anterograde injections which induces its own experimental variability and tropisms on top of the (as the authors mention) inherent variability of the tamoxifen induction. I would disagree with the authors that this toolkit is therefore “essential” for further single cell analysis. There are existing ways to target the described cell populations in the manuscript which allow for the joined analysis of both anatomical and gene expression properties.

- Again we respectfully disagree with the reviewer on these points. In terms of achieving specificity for cell type access, there is no doubt that we need intersectional strategies, as discussed above and in response to Rev1. Combining driver lines with retro- and antero-grade AAVs is readily feasible. In terms of single cell reconstruction, our driver lines have proven to be essential. This is based on a large effort study using several of our driver lines (and other lines) by the Allen Institute, Southeast University in China and their collaborators, which resulted in the most extensive single cell construction in the mouse brain (Peng et al, bioRxiv 2020; doi: <https://doi.org/10.1101/675280>). The key advantage of our inducible driver lines is that they enable reliable and “saturation screen” of genetically defined populations, with the potential to achieve ground truth of anatomical types and subtypes (e.g. Fezf2-PT, PlexinD1-IT, Tle4-CT etc). We believe this is critical to discover what are the stable/stereotypical versus variable features in neuronal morphology within each population, and clarify discrete anatomical types versus continuous variations. We do not believe this would be possible from purely viral-titer based random labeling (Winnubst et al., 2019), which has no molecular information and constraints.

While the authors claim that constructing knock in lines “has the potential to capture and reveal inherent biological elements and processes.” proof of this in the form of new biological insights are missing. Regarding this point, the first fate-mapping section of the paper goes in depth on the neurodevelopmental profile of Lhx2^{+/-} and Fezf2^{+/-} RGs. However, they don’t report any difference in, for example, the projection patterns of these three populations in the later section (though the authors apparently have results on this?). The extensive RG profiling section therefore seems more likely to be of interest to neurodevelopmental researchers and not to the wider neuroscience/biology community. I would like to echo here also the point brought up by the other reviewers: constructing intersection/subtraction lines to reveal such interactions and effects can only be performed by highly specialized labs and is therefore likely not of interest to a large audience.

- We agree with Rev3 that we have not provided proof for our quoted statement. We do have significant evidence from our studies, which are suited for a separate paper. Specifically for Lhx2⁺/Fezf2⁻ and Lhx2⁺/Fezf2⁺ RGs that we identified at embryonic stages (Fig. 1i and Extended Data Fig.2s in current revision), we have tracked their progeny to mature cortex. Strikingly, we found that these two lineages generated categorically distinct projection classes (Rebuttal Figure 1 below) . Lhx2⁺/Fezf2⁻ RGs generated PyNs that extended callosal axons but no subcortical

axons - the classic IT type. In sharp contrast, *Lhx2*⁺/*Fezf2*⁺ RGs generated PyNs that extended subcortical projections but not callosal axons - the classic PT type. To our knowledge, this result provides the first compelling evidence for fate-restricted radial glial lineages that generated distinct glutamatergic projection types in the cerebral cortex. This finding deserves to be presented in a separate paper with other related results. For the current discussion here, this result highlights the unique power of our genetic tools and strategies.

In summary, we have increasing evidence that, although intersection/subtraction methods are more demanding, they are key to link lineage and developmental trajectory to circuit connectivity and function. This is one of the most fundamental problems in neuroscience and is worth all the tool-building effort. Currently we do not see an obvious and easy alternative approach.

Specific comments:

The authors describe a novel PyNs(Fezf2/PlxnD1) population at the border of L5A/L5B, which is a very interesting finding. However, they then go on to speculate that these cells represent “intermediate PT-IT hybrids” and that “single cell reconstruction may reveal whether PyNs(Fezf2/PlxnD1) are typical IT cells or also project subcortically” (later also mentioned in the discussion). I think this is a fairly remarkable claim and such significant speculation requires more evidence. Considering what we know now, it seems more likely that this is a result of gene expression gradients along cortical layers combined with “fuzzy” borders between cell types. In this light it can also be seen as evidence of a lack of ‘specificity’ for the driver line.

- We agree with Rev2’s comment. We have tuned down the speculation and removed the use of “hybrid type”. We refrain from the use of “gene expression gradient” as we do not see such evidence for Fezf2 and PlxnD1 expression by mRNA in situ. We believe that the most useful experiment is to discover the projection pattern of Fezf2+/PlxnD1+ PyNs in relation to Fezf2+ and PlxnD1+ PyNs. The reliability of the driver lines make this quite feasible, and the result will provide major insight on the definition of PyN type. The fact that this issue can actually be solved using our driver lines also supports the strength of our genetic tools.

The authors claim that anterograde tracing revealed that the Cux1-CreER is the first cortex-restricted KI driver line. First, in Supp. movie 2 it seems there are in fact some axons in the striatum. Taking a closer look at the online publicly available data revealed this as well: <http://braincircuits.org/viewer4/mouse/map/29117F>. Are these minimal axons ignored in the analysis? Previous single cell research has shown that IT cells likely form a continuum in terms of their abundance of projections to the striatum. I therefore wonder if these merely fall at the lower end of this spectrum. Second, since systemic induction of the Cux1-CreER line causes labeling of medium spiny neurons in the striatum can the authors claim that the Cux1-CreER line will label cortex-restricted cells throughout the cortex? Or only in SSp? These points challenge the claim that Cux1-CreER is the first cortex-restricted KI driver line.

- Rev3 is correct in noting the presence of minor axon terminals in the striatum. We have modified our description to: “Anterograde tracing revealed that PyNs^{Cux1} in somatosensory barrel cortex (SSp-bfd) projected predominantly to the ipsi- and contra-lateral cortex, with only minor branches in the striatum (Fig.3a,d,e, Supplementary Movie 2).” We have removed the statement of “first cortex-restricted KI driver line”.

Even though Cux1-CreER is not absolutely specific to intra-cortex projecting PyNs, to our knowledge it is the most cortex-biased population and is thus at the intra-cortex end of the IT cells in terms of their cortex versus striatum targets. As Cux1 is a key TF, it may define a biologically significant PyN subpopulation. This can now be tested using the driver line. Thus Cux1-CreER is unique among the current IT driver lines and is likely quite useful.

The interpretation of the new description of three RG populations (Lhx2+/Fezf2+, Lhx2+/Fezf2-, and Lhx2-/Fezf2+) is unclear to me. Is this the best description of the data or could this be the result of taking a ‘snapshot’ during one time-point in development while looking at the time-dependent gradient expression of two interacting genes. In other words, are these discrete groups or is there a decreasing expression of Lhx2 over time that normally suppresses Fezf2 so that the majority of RG identities progress as: Lhx2+/Fezf2- → Lhx2+/Fezf2+ → Lhx2-/Fezf2+. Considering their difference in developmental onset isn't it expected then that Lhx2+/Fezf2- is twice as abundant? Could repeating these measurements at multiple developmental time points answer this question? I admit that I have the feeling I am missing something important here while reading the text.

- Rev3 has raised a good discussion point here. There is in fact very good evidence that Lhx2 represses Fezf2 transcription in RGs during neurogenesis (Muralidharan et al., 2017). This may contribute to the generation of Lhx2+/Fezf2-, Lhx2+/Fezf2+, Lhx2-/Fezf2+ RGs, and possibly their ratio. We provide the description of the relationship between RGs^{Lhx2+} and RGs^{Fezf2+} at multiple developmental times, not a single “snapshot” (Fig.1i-k, Extended Data Fig.2s-t). LHX2 labels a minor subset (~10%) of the RGs^{Fezf2+} population at multiple developmental stages (Extended Data Fig. 3e,i), whereas RGs^{Lhx2+Fezf2+} labels roughly 25% of the total RGs^{Lhx2+} (Fig.1l, Extended Data Fig.2t). Since the number of RGs^{Fezf2+} is extremely low at TM E10.5 (Extended Data Fig. 2g,m) and TM E13.5 (Extended Data Fig. 2k,o), we have performed experiments related to RGs^{Lhx2+Fezf2-} and RGs^{Lhx2+Fezf2+} at TM E11.5 (Extended Data Fig. 2s-t) and TM E12.5 (Fig. 1j-l). Quantifications from induction at both ages show a similar proportion of RGs^{Lhx2+Fezf2-} (red) versus RGs^{Lhx2+Fezf2+} (green). Finally, and as mentioned above, we have compelling evidence that RGs^{Lhx2+Fezf2-} and RGs^{Lhx2+Fezf2+} give rise to distinct PyN projection classes, suggesting distinct lineages and fate potentials. These results raise many questions and open the opportunities for fate mapping and molecular genetic profiling of these RG pools in subsequent studies by many other investigators.

The description of the Lhx2 pulse-chase experiments seem at some places to be inconsistent with the new quantification. “E12.5→E13.5 pulse-chase revealed a prominent medial-high to lateral-low gradient of RGs Lhx2 suggesting significant differentiation of the earlier, E10.5 RG pool.” (Line 162). Looking at the data in figure 2J and 2K it seems that this gradient is already present at E10.5 so I don't understand how this can be used to support this claim. It is then stated that a similar gradient is present at E13.5 but at lower overall density (line 163). Yet the quantification shows no quantitative difference in the number of progenitors. It would be better to show these numbers as normalized by the investigated area.

- We have modified the quoted sentence. At TM E10.5, although RGs^{Lhx2+} populate the medial cortical neuroepithelium more densely, this gradient is not prominent. There is a significant difference in RGs^{Lhx2+} between the Medial and Dorsal bins but not between Dorsal and Lateral bins. This gradient is much sharpened and more prominent by TM E12.5.

- We thank the reviewer for pointing this out. The Y-axis was mislabeled and therefore didn't show the difference in density between TM E12.5 and TM E13.5. We have corrected the axis label for both TM E12.5 and TM E13.5 (now, Ext Data Fig 2n,o).

We divided the cortical neuroepithelium in bins (medial, dorsal and lateral) of equal length (200µm) and area. Numbers demonstrate the gradient properly, therefore not normalized.

Minor points:

Line 309-311: In Supp. movie 2 it seems there are in fact some axons in the striatum of (see also <http://braincircuits.org/viewer4/mouse/map/29117F> and others). Is this low quantity ignored in the analysis?

This refers to the following example, Cux1 anterograde tracing dataset: <http://brainarchitecture.org/viewer4/mouse/map/29117F>

We agree with Rev3 that there are a minority of axons in the striatum. We have modified the text accordingly. We would like to point out that the striatal signal in Cux1-CreER is not comparable to those from similar anterograde labeling in all other IT lines that have been characterized to date.

Line 846: "Tbr2-CreER induction at E16 and E17 label L2/3 and L3 PyNs, respectively." Shouldn't this be L2?

- We have corrected this.

Line 272: "Tbr2-creER" should be "Tbr2-CreER"

- corrected

Line 849: "cp, cerebral peduncle; POm, posterior medial nucleus of thalamus; VPM, ventral posteromedial nucleus of thalamus.". These don't seem to be used in the figure.

- removed

Figure 4B: what are these strange black circles in the striatum?

- We now include white asterisks in **b**, **c**, **d** and **f** of new Figure 3 (formerly Fig.4) to indicate presence of passing fibers.

Figure 4E: Too much details in the areas being shown (321 areas!) so that the main message is lost.

- We have selected these areas to align with another BICCN paper which includes anterograde tracing from the forelimb motor cortex-- maintaining the 321 areas allows for a direct comparison. We have highlighted certain areas of interest to guide the reader.

Authors should be more selective to highlight the important differences.

Extended data figure 3: "Mixed RFP/GFP clones are most prominent and likely result from Cre activation of RFP in nRGsFezf2- and subsequent Flp activation of GFP in Fezf2+ L5/6 postmitotic PyNs (i,j')." From these pictures it seems all cells (except maybe 2?) are either GFP or RFP only.

- This sentence describes clusters or clones of cells that contain both RFP+ and EGFP+ PyNs (i.e. "mixed"). The way the IS reporter functions is that Cre alone removes a STOP cassette activating RFP expression. In the presence of both Cre and Flp, the RFP cassette is also excised and EGFP is expressed in cells. Therefore, each PyN or cell is expected to express either RFP or EGFP.

Reviewer Reports on the Second Revision:

Referee #1 (Remarks to the Author):

Comments on Matho et al Rev 2 "Genetic dissection of the glutamatergic neuron system in cerebral cortex"

Earlier I wrote "Overall, I find the tools, approach, and lines presented will be useful. I appreciate the effort that has been put into characterizing them." I think this article will be well-cited by those who make use of the new lines developed and characterized here.

Earlier I wrote that "The main source of disagreement I have with this manuscript is the comparison between the utility of BAC transgenic mice versus the transcription factor based lines." This is now minimized in the current revision, while maintaining for me the clarity of which existing types of cell the new lines will be able to access. My main objection that remains here is about the discrepancy between the images in the literature and the images in your Extended Data Figure 5 for the Sim1-Cre line. But the image there doesn't match the images in Gerfen, Heintz, 2013 (Their Fig. 3 and Your Extended Data Fig 5 are in similar planes near primary motor/MOp and the neuron density is quite different. The expression in these images is strongest in the medial wall areas (AP 0 row) while in Gerfen, Heintz, 2013, expression is strong laterally and tapers at the midline. This matches my experience as well (good expression in areas where these images show none). Using the Allen site for connectivity (<https://connectivity.brain-map.org/>) and filtering for Sim1-Cre examples, there are multiple AAV injection cases in these areas with substantial expression (I pick bright injections as examples):

Barrel Cortex: 298760754 (anterior) and 297710633 (posterior)

Motor Cortex: 297711505 and 297951944

Imagine using the set of cortical injections in the Allen data base to form a composite of cell density and compare this to the cell density composite for the brain used as the Sim1-Cre example image. I don't think they will be consistent. Although your rebuttal says "Most labs no longer use this approach to our knowledge" – I use these Tg lines. Other labs ask me to share the ones that I have. I think there is interest in the older approach AND there will be interest in the new one. That is why I would suggest resolving what is going on.

Minor points:

*Fig 1e, what time(s) was TM given? The text (Line 107) says "Long pulse-chase from E10.5, E12.5 and E14.5 labeled PyN progeny across cortical layers (Fig.1e, Extended Data Fig.2b-f, p)," but the figure legend says "e, E12.5 RGsLhx2+ generated PyNs across cortical lamina at P30." And the panel itself is labeled with an arrow up to P30.

*Line 125: is this a typo: "we differentially labeled RGsLhx2+/Fezf2 and RGsLhx2+/Fezf2+" for the first cross (should be "-" as in Fig. 1j)?

*Arrows in Fig. 2c1-d1 not mentioned.

*Tbr2 has this unusual pattern (Fig. 2c-d) where there is a dip in labeled cells between medial and dorsal areas, as well as a complete absence in lateral and temporal cortical areas. Is this present in all planes? This spatial restriction suggests that the rules governing generation of cortical laminae vary across cortex.

*The text did not explain why the Snap25 reporter was used. "The reporter allele was Ai14, except for PlxnD1-CreER (Snap25-LSL-EGFP)".

*Fig 3f, sp.cord is listed as Spd in the Abbreviations.

*PO (Fig. 4, Ext. Data Fig. 9, etc.) and P_{Om} are both used, but only P_{Om} is used in the list of abbreviations.

*Lines 272,274 use FezF2 instead of Fezf2 (Line 208) nomenclature. Maybe this is a typo, but maybe I misunderstand the nomenclature (the referred figure Extended Data Fig. 7 calls it Fezf2).

*Fig 4f (motor cortex) and Fig 4h (somatosensory cortex) quantify similar things, but one is presented in micrometers and the other uses normalized distances.

*Fig 4g-h look at output to various subcortical structures, with nearly overlapping distribution of L5 somata projecting to Sp5, P_{Om}, SC. Are you arguing for or against a variety of cell types or a single branching axon projecting to these different targets? This is consistent with some recent authors suggesting a small number of PT type neurons projecting to subcortical targets (say the Economo and Winnubst papers), but contrasts to the range of laminar depths for different subtypes (Oberlaender group). This is not commented on in depth in the text.

*The retrogradely labeled striatal fraction in Fig. 4g-h is quite substantial and shifted upwards/towards the pia from the bulk of the other targets (Sp5, P_{Om}, SC). Does this data suggest that the Fezf2 population is in the upper part of L5 (L5A) and does not project to subcortical targets other than striatum? I don't have the data to quantify, but that distribution shown is consistent with substantial IT-type neuron label in L5A. (The population itself might be relatively small (4g), but the normalization (4h) gives the impression all these populations are similar size.

*On Fig 4i+, I was trying to verify that TM is delivered at P21-P28 (as in some of the earlier figures), since these seem to be one of the conditions that some users of these lines would like to implement.

*On the sag ratio, the Fig. 4l-m still seems to suggest that the data is gathered in the same way as before, so not fixed. But the rebuttal letter agrees with the technical points (that it should be measured from a comparable V_{rest} to compare voltage gated currents). As said before, "This isn't really needed for the paper." If you think it is the correct way to do the experiment, can you offer a rationale? If not needed to support the overall conclusions of the project, why include the data?

*Line 339: "Strikingly, TM induction at E13.5, 15.5 and 17.5 selectively labeled L5A, L3, and L2 PyNsPlxnD1, respectively (Fig. 5b-d)." I am wondering if any L4 neurons were labeled in these experiments. Fig 5b'-e' appear to be aligned from pia to white matter, and the brightness in Fig. 5c' seems to line up with the label for L4 at left (or at least is situated immediately adjacent to where the L5A neurons of the TM E13.5 induction end), but this is not commented on in the text.

*An effective survey is done of the different induction time and its effect on projection pattern in the PlxnD1 line. This suggests that perhaps the different Fezf2 projection patterns show (PT, CT, IT) might also emerge with neurons born at different times (though a different strategy seems needed to target them to show this).

*This is not a comment on the data in the paper, but on what it all means. From the rebuttal: "Although the "composite cell types" captured by some KI lines may seem, for the first pass, less convenient and not immediately satisfying for circuit neuroscientists, we believe these provide a logical and clear path toward a biology-based systematic dissection of the hierarchical organization of neuron types." Along these lines, Extended Data Fig. 10 is titled "A subset of PyNs(Fezf2) manifest IT features." And these features are discussed in the main text (Fig. 3 as well). I am not convinced that the only logical way forward is by single-genetic-marker (or intersection of two) labels for cell types in cases where cells in different layers of with different projection patterns are lumped together. Perhaps this is because I am stuck in the way of thinking that 'cell types' will explain it all. Perhaps enhanced based methods will surmount some of the obstacles here (and it will be neat to see if they similarly span laminar boundaries). But

(a) Your letter suggests some phenomena (“optogenetic motor mapping using our Fezf2 (and other) driver lines, which generate surprisingly complex and smooth (i.e. natural looking) forelimb and orofacial movements compared to similar studies using other Tg driver lines”) that might be convincing. (I wonder, are the cells with similar expression related in some way and form networks as in the Song-hai Shi series of papers?) So maybe this is an argument that the seeming disparate cell types labeled by a single TF but having apparent differences in axonal targeting or laminar position do form functional units. (***)Absolutely NOT requesting the scope of this paper include this data.***)

(b) Maybe this will become more clear in subsequent papers about the biology (e.g. if there is some literature to emerge that we shouldn’t study cells in a certain layer projecting to a certain place as a type, but instead study according to your TF-based approach even when the cell types differ from the categories we currently use.

I just bet that cell types will be useful and will be interested to see if a new literature emerges to undermine the centrality of laminar identity and projection pattern which is so essential to how I would (for now) define an excitatory cell type in cortex.

Referee #3 (Remarks to the Author):

The authors have made significant efforts to streamline the overall presentation of the result in the paper and I welcome the changes they made based on my and the other reviewers feedback. I also appreciated the author's thoughtful discussion points raised in their response to my previous comments. To be clear, I fully agree that intersectional approaches will always be needed to study fine-grained “cell-types” and that this does not take away from the power of their approach. I think my original point remains, however, that a systematic framework for studying cellular diversity based on this toolkit is not fully demonstrated in this paper. This being said, I feel that with the adjusted texts, that tones down on some of the comparisons on transgenic lines, the advantages of the toolkit are presented in a more equitable way that I agree with.

I enjoyed seeing the rebuttal figure on the distinct projection classes arising from the Lhx2+/Fezf2- and Lhx2+/Fezf2+ RG populations as it is exactly the kind of novel biological insight that I feel is lacking in this section. Given the sheer size and scope of the current paper I can fully appreciate the author’s predicament that they feel it is more appropriate to fully delve into such results in a separate paper. (Indeed, I would be very curious to read such a paper!). However, I feel that including at least a part of this data is important as an example of the utility of this progenitor fate mapping strategy (which is a key point of the paper) since these results could clearly not be achieved with other methods. Doing so would not preclude further classification in a later manuscript and the manuscript could include a statement that doing so falls outside the scope of the current paper. I strongly feel that this would significantly enhance the impact of the paper and make its importance clear to a wider audience.